# A Lagrangian Approach Towards Extracting Signals of Urban CO₂ Emissions from Satellite Observations of Atmospheric Column CO₂ (XCO₂): X-Stochastic Time-Inverted Lagrangian Transport model ("X-STILT v1")

Dien Wu[1], John C. Lin[1], Benjamin Fasoli[1], Tomohiro Oda[2], Xinxin Ye[3], Thomas Lauvaux[3], Emily G. Yang[4], Eric A. Kort[4]

[1]Department of Atmospheric Sciences, University of Utah, Salt Lake City, USA
[2]Goddard Earth Sciences Technology and Research, Universities Space Research Association, Columbia, Maryland/Global Modeling and Assimilation Office, NASA Goddard Space Flight Center, Greenbelt, Maryland, USA
[3]Department of Meteorology and Atmospheric Science, Pennsylvania State University, USA
[4]Climate and Space Sciences and Engineering, University of Michigan, Ann Arbor, USA

*Correspondence to*: Dien Wu (Dien.Wu@utah.edu)

**Abstract.** Urban regions are responsible for emitting significant amounts of fossil fuel carbon dioxide (FFCO₂), for which emissions at finer, city scales are more uncertain than those aggregated at the global scale. Carbon-observing satellites may provide independent *top-down* emission evaluations and compensate for the sparseness of surface CO₂ observing networks, especially in urban areas. Although some previous studies have attempted to derive urban CO₂ signals from satellite column-averaged CO₂ data (XCO₂) using simple statistical measures, less work has been carried out to link upwind emission sources to downwind atmospheric columns using atmospheric models. In addition to Eulerian atmospheric models that have been customized for emission estimates over specific cities, the Lagrangian modeling approach—in particular, the Lagrangian Particle Dispersion Model (LPDM) approach—has the potential to efficiently determine the sensitivity of downwind concentration changes to upwind sources. However, when applying LPDMs to interpret satellite XCO₂, several issues have yet to be addressed including quantifying uncertainties in urban XCO₂ signals due to receptor configurations and errors in atmospheric transport and background XCO₂.

In this study, we present a modified version of the Stochastic Time-Inverted Lagrangian Transport (STILT) model, "X-STILT", for extracting urban XCO₂ signals from NASA's Orbiting Carbon Observatory 2 (OCO-2) XCO₂ data. X-STILT incorporates satellite profiles and provides comprehensive uncertainty estimates towards urban XCO₂ enhancements on a per sounding basis. Several methods to initialize receptors/particle setups and determine background XCO₂ are presented and discussed via sensitivity analyses and comparisons. To illustrate X-STILT's utilities and applications, we examined five OCO-2 overpasses over Riyadh, Saudi Arabia, during a two-year time period and performed a simple scaling factor-based inverse analysis. As a result, the model is able to reproduce most observed XCO₂ enhancements. Error estimates show that the 68 % confidence limit of XCO₂ uncertainties due to transport (horizontal wind plus vertical mixing) and emission uncertainties contribute to ~33 % and ~20 % of the mean latitudinally-integrated urban signals, respectively, over the five overpasses, using meteorological fields from the Global Data Assimilation System (GDAS). In addition, a sizeable mean difference of -0.55 ppm in background derived from a previous study employing simple statistics (regional daily median) leads to a higher mean observed urban signal by ~39 % and a larger posterior scaling factor. Based on our signal estimates and associated error impacts, we foresee X-STILT serving as a tool for interpreting column measurements, estimating urban enhancement signals, and carrying out inverse modeling to improve quantification of urban emissions.

# 1 Introduction

Carbon dioxide ($CO_2$) is a major greenhouse gas in the atmosphere in terms of radiative forcing, with its concentration increasing significantly over the past century (Dlugokencky and Tans, 2015). The largest contemporary net source of $CO_2$ to the atmosphere over the decadal time scales is anthropogenic emissions, namely from fossil fuel burning and net land-use change (Ciais et al., 2013). Urban areas play significant roles in the global carbon cycle and are responsible for over 70 % of the global energy-related $CO_2$ emissions (Rosenzweig et al., 2010). Global fossil fuel $CO_2$ (FFCO$_2$) emission uncertainty (8.4%, $2\sigma$, Andres et al., 2014) may be smaller than other less-constrained emissions such as of wildfire (Brasseur and Jacob, 2017). Still, uncertainties associated with national FFCO$_2$ emissions derived from *bottom-up* inventories typically range from 5–20 % per year (Andres et al., 2014). These estimated emission uncertainties result primarily from differences in emission inventories, such as the emission factors and energy consumptions data used. Moreover, heightened interests in regional- and urban-scale emissions require modelers to investigate FFCO$_2$ emissions at finer spatiotemporal resolutions (Lauvaux et al., 2016; Mitchell et al., 2018) as well as uncertainties in gridded emissions (Andres et al., 2016; Gately and Hutyra, 2017; Hogue et al., 2016; Oda et al., 2018). Dramatic increases in emission uncertainties are associated with finer scales, with these uncertainties being mostly biases due to different methods disaggregating national-level emissions (Marland, 2008; Oda and Maksyutov, 2011). For instance, emission uncertainties of 20 % at regional scales increased to 50–250 % at city scales even for the northeastern United States (Gately and Hutyra, 2017), an area that is considered relatively "data-rich".

Given the large differences/discrepancies in emission inventories at urban scales, the use of atmospheric *top-down* constraints could be helpful for quantifying urban emissions and possibly providing a monitoring support (Pacala et al., 2010). Observed concentrations used in the top-down approach can often be obtained from ground-based instruments (Kim et al., 2013; Mallia et al., 2015; Wunch et al., 2011) and aircraft observations (Gerbig et al., 2003; Lin et al., 2006). Each type of measurement offers valuable information and has both advantages and disadvantages. Most ground-based measurements provide reliable, continuous $CO_2$ concentrations from fixed locations/heights. Unfortunately, current ground-based observing sites are too sparse to constrain urban emissions around the globe. Most National Oceanic and Atmospheric Administration (NOAA) sites are designed to measure background concentrations and few others aim at measuring concentration changes from few vertical levels within the planetary boundary layer (PBL). Other than a few notable examples (Feng et al., 2016; Lauvaux et al., 2016; Mitchell et al., 2018; Verhulst et al., 2017; Wong et al., 2015; Wunch et al., 2009), near-surface $CO_2$ measurements may not be available over many other cities around the world. Alternatively, airborne measurements from field campaigns provide better vertical and regional coverages (Cambaliza et al., 2014). Yet, continuous airborne operations over months to years are often impractical due to limited resources, which limits researchers' capability to track temporal variability of anthropogenic carbon emissions (Sweeney et al., 2015).

The carbon cycle community has entered a new era with advanced carbon-observing satellites—i.e., Greenhouse gases Observing SATellite (GOSAT; Yokota et al., 2009), TanSat (Liu et al., 2013) and Orbiting Carbon Observatory (OCO-2) satellite (Crisp et al., 2012)—routinely in orbit to measure variations of atmospheric column-averaged $CO_2$ mole fraction (XCO$_2$). Although most carbon-observing satellites have revisit times of multiple days (e.g., 3 days for GOSAT and 16 days for OCO-2), their global coverage, large number of retrievals and multi-year observations may further complement the current surface observing networks. Space-borne $CO_2$ measurements, in combination with surface $CO_2$ networks, may help reduce emission uncertainties and benefit urban emissions analysis, especially over regions with no surface observations (Duren and Miller, 2012; Houweling et al., 2004; Rayner and O'Brien, 2001).

Previous studies have demonstrated the potential for detecting and deriving urban $CO_2$ emission signals from satellite $CO_2$ observations, in the form of XCO$_2$ enhancements above the background, without making use of much atmospheric transport

information (Hakkarainen et al., 2016; Kort et al., 2012; Schneising et al., 2013; Silva and Arellano, 2017; Silva et al., 2013). However, the linkage between their derived urban $CO_2$ emission signals and upstream sources is tenuous, as downwind $XCO_2$ can be enhanced by not only near-field upwind urban activities (e.g., traffic, houses, and power plants/industries), but regional-scale advection of upwind sources/sinks as well. Simulations using transport models are able to isolate the portion of satellite observations influenced by urban regions from the portion affected by natural fluxes or long-range transport (e.g., Ye et al., 2017). Therefore, accurate knowledge of atmospheric transport is essential in top-down assessment. As importantly, transport modeling is a necessary step within inverse modeling, which can help improve fossil fuel emission estimates and shed light on $CO_2$ emission monitoring networks (Kort et al., 2013; Lauvaux et al., 2009). Uncertainties in transport modeling have been identified as a significant error source that affects inferred surface fluxes (Peylin et al., 2011; Stephens et al., 2007; Ye et al., 2017). Yet, by analyzing an increased number of satellite overpasses, uncertainties from atmospheric inversions due to non-systematic transport errors in emission estimates can be reduced (Ye et al., 2017).

Two main approaches can be considered for atmospheric transport modeling. Eulerian models, in which fixed grid cells are adopted and $CO_2$ concentrations within the grid cells are calculated by forward numerical integrations, have been widely utilized and customized to understand urban emissions and quantify model uncertainties over specific metropolitan regions worldwide (Deng et al., 2017; Lauvaux et al., 2013; Palmer, 2008; Ye et al., 2017). The Lagrangian approach, especially the time-reversed approach in which atmospheric transport is represented by air parcels moving backward in time from the measurement location ("receptor"), is efficient in locating upwind sources and facilitating the construction and calculation of the "footprint" (e.g., Lin et al., 2003) or "source-receptor matrix" (Seibert and Frank, 2004)—i.e., the sensitivity of downwind $CO_2$ variations to upwind fluxes.

In particular, the receptor-oriented Stochastic Time-Inverted Lagrangian Transport (STILT) model, one of the Lagrangian Particle Dispersion Models (LPDM), has the ability to more realistically resolve the sub-grid scale transport and near-field influences (Lin et al., 2003). STILT has been used to interpret $CO_2$ observations within the PBL (Gerbig et al., 2006; Kim et al., 2013; Lin et al., 2017) and, in recent years, to analyze column observations, i.e., $XCO_2$ (Fischer et al., 2017; Heymann et al., 2017; Macatangay et al., 2008; Reuter et al., 2014). Among STILT-based column studies, most aim at either natural $CO_2$ sources and sinks like wildfire emissions and biospheric fluxes, or anthropogenic emissions at regional or state scales. Very few studies focus on city-scale $FFCO_2$ using column data and LPDMs. Moreover, when applying LPDMs to interpret column $CO_2$ data, three key issues have yet to be carefully examined and will be addressed in this paper:

**I. Uncertainty of modeled $XCO_2$ enhancements due to model configurations.** Very few studies have examined model uncertainties resulting from model configurations—i.e., receptors and particles in LPDMs. Previous studies reported negligible to ~20 % of the modeled enhancements are reported as the error impact due to STILT particle number (released from a fixed level), depending on adopted particle numbers, examined species and their components/sources (Zhao et al., 2009; Gerbig et al., 2003; Mallia et al., 2015). When it comes to representing an atmospheric column using particle ensembles, many studies depicted their setups for receptors/particles without further explaining why they chose those setups or the error impact (due to model configurations) on modeling $XCO_2$. Although this error impact may be small, we still perform a set of sensitivity tests to provide more guidance on placing column receptors.

**II. Horizontal and vertical transport error impact on $XCO_2$ simulations.** Flux inversions, e.g., Bayesian Inversion (Rodgers, 2000) involving LPDMs have been widely adopted to constrain emissions. Approaches to quantify errors in horizontal wind fields and vertical mixing have been proposed followed by comprehensive error characterizations on atmospheric simulations (Gerbig et al., 2008; Jeong et al., 2013; Lauvaux et al., 2016; Lin and Gerbig, 2005; Zhao et al., 2009). Recent efforts (e.g., Lauvaux and Davis, 2014; Ye et al., 2017) have been made to rigorously examine the column transport

errors. The uncertainties in horizontal wind fields and vertical mixing within X-STILT will be propagated into column $CO_2$ space in this study.

**III. Determining background $XCO_2$ and characterizing its uncertainties.** Here we define background value as the $CO_2$ "uncontaminated" by fossil fuel emissions from the city of interest. As urban emission signals are defined as the enhancements of $XCO_2$ over the background, errors in the background value introduce first-order errors into the derived urban $XCO_2$ signal from total $XCO_2$, with such errors propagating directly into fluxes calculated from atmospheric inversions (e.g., Göckede et al., 2010). Consequently, background determination is another critical task.

One commonly used method in determining model boundary conditions of various species in LPDMs is the "trajectory-endpoint" method that establishes the background based on $CO_2$ extracted at endpoints of back trajectories from modeled regional/global concentration fields (Lin et al., 2017; Macatangay et al., 2008; Mallia et al., 2015). The aforementioned studies (adopting the trajectory-endpoint method) aim at extracting relatively large $CO_2$ anomalies (e.g., at a fixed level within the PBL or due to large emissions such as of wildfire) out of the total measured $CO_2$. However, for studying $XCO_2$ that is less variable than near-surface $CO_2$ (Olsen and Randerson, 2004), potential errors in modeled concentration fields and atmospheric transport may pose more significant adverse impact on derived urban signals. Other ways of defining background include geographic definitions (Kort et al., 2012; Schneising et al., 2013) and simple statistical estimates (Hakkarainen et al., 2016; Silva and Arellano, 2017). These simple statistical methods often neglect the atmospheric transport and may use a less accurate upwind region to select measurements for deriving background values. Lastly but more importantly, recent column studies (Nassar et al., 2017; Fischer et al., 2017) studied the impact of potential errors/biases in background values on their emission or fluxes estimates. In this work, we introduce a new background determination that combines OCO-2 observations and the STILT-based atmospheric transport and account for errors in our background estimates.

In general, we attempt to address the aforementioned issues by extending STILT with column features and comprehensive error analyses, referred to as the column-STILT, "X-STILT". We illustrate the model's applications in extracting urban $XCO_2$ signals from OCO-2 retrievals (Fig. 1) and evaluate model performances via a case study focusing on Riyadh, Saudi Arabia. Riyadh, with a population of over 6 million by 2014 (WUP 2014), is chosen as the city of interest because of its low cloud interference, limited vegetation coverage, and isolated location in a barren area, which leads to higher data recovery rates and facilitates the background determination. Saudi Arabia has the largest $CO_2$ emissions among Middle Eastern countries and ranks eighth globally in 2016 (Boden et al., 2017; BP, 2017; UNFCCC, 2017). We examine several satellite overpasses and focus on a small spatial domain adjacent to Riyadh for each overpass.

## 2 Data and methodology

Before demonstrating model details, Fig. 1 highlights several X-STILT characteristics, e.g., column transport errors quantifications, background $XCO_2$ approximations, and the identification of upwind emitters using backward-time runs from column-receptors. Our goal is to evaluate the model by comparing both the latitude-dependent model-data $XCO_2$ urban enhancements (Sect. 3.4) and the overall latitude-integrated urban signals within a small latitudinal range (Sect. 3.5). We selected and examined five OCO-2 overpasses during the time period of Sept 2014–Dec 2016, based on four stringent criteria (Appendix A).

**2.1 STILT-based approach for XCO₂ simulation ("X-STILT")**

The OCO-2's column averaging kernel is the product of normalized averaging kernel ($AK_{norm}$) and pressure weighting ($PW$) function and represents the sensitivity of the change in retrieved XCO₂ due to CO₂ anomaly at each retrieved grid. Column $AK_{norm}$ peaks near the surface and exhibits values near unity throughout most of the troposphere (Boesch et al., 2011). Lower $AK_{norm}$ values are mainly found aloft, which requires more information in the a priori CO₂ profiles ($CO_{2,prior}$; Fig. 2a). For direct comparisons against OCO-2 retrieved XCO₂, CO₂ anomalies at model grids should be properly weighted using the satellite's column averaging kernels (Basu et al., 2013; Lin et al., 2004). Thus, the final $AK$-weighted simulated XCO₂ ($XCO_{2.sim.ak}$) are weighted between model-derived CO₂ profiles and OCO-2 a priori profiles (O'Dell et al., 2012):

$$XCO_{2.sim.ak} = \sum_{n=1}^{nlevel}(AK_{norm,n}PW_nCO_{2.sim.n} + (I - AK_{norm,n})PW_nCO_{2.prior,n}) \tag{1}$$

$I$ is the identity vector and $n$ stands for the combined vertical levels of STILT plus OCO-2. Specifically, we replaced OCO-2 levels with denser model release levels for the lower part of the troposphere (red circles from the surface in Fig. 2), while kept OCO-2 levels for upper part (blue circles in Fig. 2). To reduce computational cost, the air column is only simulated up to the maximum release height (MAXAGL in meters above ground level, mAGL; Fig. 2).

Interpolations are further needed to resolve the mismatch between prescribed OCO-2 retrieval grids and model levels for the lower part of the troposphere. Our intention is to preserve the finer modeled CO₂ variations by performing interpolations of satellite profiles from retrieval grids to model levels. Vertical profiles of $AK_{norm}$, $PW$ and $CO_{2,prior}$ are treated as continuous functions and interpolated linearly to model grids (red circles in Fig. 2). Note that the initial OCO-2 $PW$ functions have steady value of ~0.052 (except for the very bottom and top levels; black dots in Fig. 2b), which results from constant pressure spacings ($dp\_oco2$) between two adjacent OCO-2 levels. However, X-STILT levels are much denser with smaller pressure spacings ($dp\_stilt$) or less airmass between their two adjacent levels. Therefore, the linearly interpolated $PW$ (red circles in Fig. 2b) needs an additional scaling via a set of "scaling factors" representing the ratios of pressure spacings in STILT versus OCO-2 retrieval ($dp\_stilt/dp\_oco2$), to arrive at the correct $PW$ for each finer model grid (orange circles in Fig. 2b).

Eq. (1) can further be rewritten as Eq. (2), since the simulated CO₂ profiles in Eq. (1) is comprised of CO₂ boundary condition plus CO₂ anomalies due to sources/sinks (FFCO₂, biospheric and oceanic fluxes):

$$XCO_{2.sim.ak} = XCO_{2.sim.ff} + XCO_{2.sim.bio} + XCO_{2.sim.ocean} + XCO_{2.sim.bound} + XCO_{2.prior} = XCO_{2.sim.ff} + XCO_{2.bg}. \tag{2}$$

Given our focus, we defined background value as the XCO₂ portion not "contaminated" by urban emissions. Thus, $XCO_{2.sim.ak}$ is the sum of the XCO₂ enhancement due to FFCO₂ ($XCO_{2.sim.ff}$) and estimated background value ($XCO_{2.bg}$). Estimates of XCO₂ anomalies are further explained in Sect. 2.2.2 and four ways to estimate background values ($XCO_{2.bg}$) are proposed in Sect. 2.3.

**2.1.1 X-STILT setup ("column receptors")**

The linkage between the observed XCO₂ concentration by a given OCO-2 sounding and upwind carbon sources and sinks is determined by atmospheric transport. We adopt the STILT model to describe this connection. Fictitious particles, representing air parcels, are released from a "receptor" (location of interest) and are dispersed backward in time. The Lagrangian air parcels within STILT are transported along with the mean wind ($\bar{u}$), turbulent wind component ($u'$), and other meteorological variables, which are derived from Eulerian meteorological fields. In this study, we used meteorological fields simulated by the Weather Research and Forecasting (WRF; Skamarock and Klemp, 2008) and the 0.5° × 0.5° Global Data Assimilation System (GDAS; Rolph et al., 2017; Stein et al., 2015). Hourly WRF fields contain 51 vertical levels with boundary conditions from 6-hourly 0.5°×0.5° NCEP

FNL (Final) Operational Global Analysis data (Ye et al., 2017) are customized and utilized for the first 2 of the total 5 overpasses over Riyadh. We note that the primary focus is to assess the resulting errors given the choice of a particular wind field (i.e., GDAS 0.5°), rather than to carry out analyses of differences between WRF and GDAS.

To represent the air arriving at the atmospheric column of each OCO-2 sounding, we release air parcels from multiple vertical levels, "column receptors" (Fig. 3e), using the same lat/lon as the satellite sounding at the same time and allow those parcels to disperse backward for 72 hours (see Appendix D2 for model impact from backward durations). About 10–20 satellite soundings are selected for simulations over every 0.5° latitude with data filtering using criteria explained in Sect. 2.2. Sensitivity tests are conducted regarding different configurations—the maximum release level (MAXAGL), the vertical spacing of release levels ($dh$), and the particle number per level ($dpar$), when placing column receptors (Sect. 2.5).

### 2.1.2 Modeling XCO$_2$ anomalies

Air parcels traveling back in time provide valuable information about how upwind sources and sinks impact the air arriving at a receptor. However, since particles within the ensemble are subject to stochastic motion, the surface fluxes observed by any single particle caries limited information. The influence of upstream surface fluxes on a receptor is given by summing the sensitivities of all particles in the ensemble over a surface grid $f(x, y, t)$, which is referred to as the "footprint" (Lin et al., 2003; Fasoli et al., 2018) or the "source-receptor matrix" (Seibert and Frank, 2004). Formally, the sensitivity of the receptor located at $\boldsymbol{x}_r$ at time $t_r$ to surface fluxes originating from $x_i, y_j$ is given by summing $\Delta t_{p,i,j,z \leq h}$, the time spent by particle $p$ over grid position $i, j$ within the surface layer of height $h$ for each discrete time step $m$:

$$f(\boldsymbol{x}_r, t_r | x_i, y_j, t_m) = \frac{m_{air}}{h \bar{\rho}(x_i, y_j, t_m)} \frac{1}{N_{tot}} \sum_{p=1}^{N_{tot}} \Delta t_{p,i,j,z \leq h} \tag{3}$$

where $N_{tot}$ is the total number of particles in the ensemble, $m_{air}$ is the molar mass of dry air, $\bar{\rho}$ is the average air density below $h$. The dilution of surface fluxes to half of the PBL height $h = 0.5 z_{pbl}$ is often used. In general, $f$ increases if particles travel at heights $z \leq h$ and if $h$ is low, concentrating surface fluxes within a shallower atmospheric column.

To reduce grid noise caused by aggregation of a finite number of dispersed particles, a kernel density estimator is used to variably smooth $f$ as a function of elapsed time and particle location uncertainty. Refer to Fasoli et al. (2018) for additional details pertaining to the formulation of $f$.

We introduce the weighted column footprint $f_w$ that describes the sensitivity of changes in column concentration due to potential upstream sources/sinks and incorporates satellite profiles. The formulation of $f_w$ is similar to (3) but scales the sensitivity with $AK_{norm}(n, r)$ and $PW(n, r)$:

$$f_w(\boldsymbol{x}_{n,r}, t_{n,r} | x_i, y_j, t_m) = \frac{m_{air}}{h \bar{\rho}(x_i, y_j, t_m)} \frac{1}{N_{tot}} \sum_{p=1}^{N_{tot}} \Delta t_{p,i,j,z \leq h} \, AK_{norm}(n, r) \, PW(n, r) \tag{4}$$

where $\boldsymbol{x}_{n,r}, t_{n,r}$ denotes a column receptor. Multiplying $f_w$ by gridded flux estimates yields a change in CO$_2$ at the downwind column receptor. Thus, surface fluxes $F(x_i, y_j, t_m)$ cause a change in column integrated mole fraction $\Delta XCO_2$ as

$$\Delta XCO_2(\boldsymbol{x}_{n,r}, t_{n,r} | x_i, y_j, t_m) = F(x_i, y_j, t_m) \, f_w(\boldsymbol{x}_r, t_r | x_i, y_j, t_m). \tag{5}$$

For our OCO-2 case study, modeled XCO$_2$ enhancements due to FFCO$_2$ emissions are derived from the convolution of spatially-varying $f_w$ and ODIAC emissions (Sect. 2.4.1). Also, we account for modeled uncertainties that includes errors in prior FFCO$_2$ emissions (Sect. 2.4.1), receptor configurations (Sect. 2.5), and atmospheric transport (Sect. 2.6).

**2.2 OCO-2 retrieved XCO$_2$ and data pre-processing**

The OCO-2 algorithm of retrieving XCO$_2$ from radiances employs an optimal estimation approach (Rodgers, 2000) involving a forward model, an inverse model, and prior information regarding the vertical CO$_2$ profiles (O'Dell et al., 2012). We used the bias-corrected XCO$_2$ values from OCO-2 Lite files (version 7R; OCO-2 Science Team/Michael Gunson, Annmarie Eldering, 2015). The impacts of different versions of the OCO-2 datasets on our results are briefly discussed in Sect. 4.4. Satellite measurements over Riyadh were carried out in Land Nadir and Glint modes. Soundings with quality flags equal zero (QF = 0) are selected, which implies selected observations have passed the cloud and aerosol screening (with removal of albedo > 0.4) and their retrievals have converged (Mandrake et al., 2013; Patra et al., 2017). For smoothing noisy observations, we binned the screened XCO$_2$ data according to the lat/lon of model receptors (served as the midpoints of each bin) and calculated the mean and standard deviation of screened measurements within each bin. Next, background values were defined (Sect. 2.4) and subtracted from the bin-averaged observed XCO$_2$ to estimate the increase in observed XCO$_2$ (step 3 in Fig. 1). The impacts of different bin-widths on bin-averaged observed signals are shown in Appendix E1. Total observed errors contain the spatial and natural variation of observed XCO$_2$ in each bin, background uncertainties (Sect. 2.3.3), and retrieval errors provided by Lite files. Retrieval error variances per sounding are then averaged within each observed bin to obtain bin-averaged retrieval error variances.

**2.3 Estimates of background XCO$_2$**

Definitions of "background" vary among studies with different applications. Here, we define background values as atmospheric XCO$_2$ that is not "contaminated" by the urban emissions around our study site. Determination of background XCO$_2$ is crucial, as it can significantly affect the magnitude of inferred observed anthropogenic signals. If the background is underestimated, then the detected signal may be overestimated, and vice versa. In this study, we seek to develop best-estimated background values given five tracks, where 3 methods are proposed and investigated as follows,

M1. A "trajectory-endpoint" method by assigning CO$_2$ values extracted from global models (i.e., CT-NRT) to trajectory endpoints plus simulated biospheric, oceanic, prior components (Sect. 2.3.1);

M2. Statistical methods estimated solely from XCO$_2$ observations based on two previous studies (Sect. 2.3.2);

M3. An "overpass-specific" background requires model-defined urban plume and measurements outside the plume (Sect. 2.3.3).

We devote considerable efforts to compare the aforementioned three ways (Sect. 3.3) and investigate the background impact on model-data comparisons and emission estimates (Sect. 4.2). We choose M3-based background for the following analysis as it is designed specifically for examining a particular city and specific overpasses downwind of the city.

**2.3.1 Trajectory-endpoint method (M1)**

Modeled background XCO$_2$ comprises modeled boundary condition confined by four-dimensional CO$_2$ fields from CT-NRT and contributions from biospheric fluxes, oceanic fluxes, and OCO-2 prior profiles (M1 in Fig. 1 and Eq. (2)). Specific for modeling CO$_2$ boundary condition, CO$_2$ values for upper levels above MAXAGL are estimated based on CT CO$_2$ at those OCO-2 pressure levels (purple circles in Fig. 2c). And, averaged CT CO$_2$ values at trajectory endpoints are used for boundary conditions at model release levels (orange circles in Fig. 2c). Then, modeled boundary conditions at vertical levels are weighted accordingly via OCO-2's column averaging kernel (red and blue circles in Fig. 2c). Model trajectories are properly subsetted according to the boundary of the footprint domain (i.e., 20° × 20°) used for simulating XCO$_2$ anomalies.

However, potential uncertainties in transport may strongly influence the distribution of Lagrangian parcels as backward duration time increases and may lead to potential spatial mismatch of the background region. Furthermore, potential biases and relatively coarse resolution of 2° × 3° of the global CarbonTracker may add inaccuracies to $CO_2$ values at trajectory-endpoints.

### 2.3.2 Statistic method (M2)

Hakkarainen et al. (2016) (referred to as M2H) extracted local $XCO_2$ anomalies from the daily median of screened measured $XCO_2$ within a relatively broad region (0° N–60° N, 15° W–60° E over the Middle East; Fig. S7). Their detected anomalies vary from 1–2 ppm over 0.5° × 0.5° gridcells near Riyadh. Silva and Arellano (2017) (referred to as M2S) used measurements within a 4° × 4° combustion region centered around the "urban and dense settlements" inferred from the anthropogenic biomes dataset ("anthromes", Ellis and Ramankutty, 2008). Then, they derived background as the mean minus one standard deviation of available observations within their studied urban extents.

Both statistical methods are highly efficient in estimating background values but can be limited to certain applications. For instance, M2H may be not suitable for determining background values when zooming into specific cities. Measurements within their broad spatial domain are lumped together, regardless of their locations (whether over rural or urban areas) and atmospheric transport. Silva and Arellano (2017) have pointed out that their defined 4° × 4° combustion region is suitable for studying the "bulk" characteristics and may be too coarse for studying urban emissions. Also, the Gaussian statistics assumed in M2S may be less applicable when multiple observed peaks are entangled together caused by a cluster of cities. Therefore, without incorporating much atmospheric transport information, accuracies in the transport from an urban center to the downwind satellite overpass cannot be guaranteed. It may be difficult for either statistical method to locate the exact $XCO_2$ peak elevated by target city or background region. These difficulties motivate us to introduce a new approach in the next subsection.

### 2.3.3 Overpass-specific background (M3)

A few space-based studies defined the background values as the averaged observed $XCO_2$ values over a "clean" upwind region. For instance, Kort et al. (2012) and Schneising et al. (2013) defined the "clean" region based on geographic information (e.g., rural area to the north of LA basin). Although OCO-2 has relatively narrow swaths, transport models can be used to differentiate the enhanced versus background portions along an overpass. For example, Janardanan et al. (2016) calculated background $XCO_2$ as the averaged GOSAT observations among gridcells with modeled anthropogenic signals < 0.1 ppm. This 0.1 ppm threshold is determined from the average simulated fossil fuel abundance over desert areas worldwide using the FLEXible PARTicle dispersion model (FLEXPART; Stohl et al., 2005), a model similar to STILT in that both are time-reversed LPDMs. Nassar et al. (2017) derived overpass-dependent background and its uncertainty based on the averaged OCO-2 observations within four different tested background latitudinal ranges.

We present an alternative method using a forward-time run from an urban box to reveal the urban influence on satellite soundings, which are more straightforward and efficient than solely relying on backward-time runs. Fictitious particles are released from a box around the city center (pink dots in Fig. 1) as a feature implemented with STILT (T. Nehrkorn, personal communication) to track air parcels over a city and the transport of the urban plume. Specifically, the model continuously releases air parcels over a 30-minute window from a 0.4° × 0.4° box around the city center, with multiple 30-minute releases of 1000-particle ensembles over the 10-hours ahead of the satellite overpass hour (00–10UTC). Then, an urban plume can be derived from the parcels' distribution during the ~3 minutes OCO-2 passing window (purple dots in Fig. 5a). Note that air parcels are tracked forward in time for 12 hours, allowing for equal contributions from parcels released initially from different

time intervals (every 30 mins) onto defined urban plume. We are aware of potential model errors and their adverse impacts on defined urban plume. Therefore, a wind error component (details in Sect. 2.6) is further added in the forward run to broaden the polluted range (solid black line in Fig. 5a). Next, two-dimensional kernel density estimation (Venables and Ripley, 2002) is applied to determine the boundary of the city plume based on the air parcels' distributions. We normalized the two-dimensional

5 kernel density by its maximum value and "sketched" the boundary of the city plume based on a threshold of 0.05, which is sufficient to include most air parcels. No dramatic change in the shape/size of resultant urban plume was found as testing other thresholds < 0.05. The urban-influenced latitude range is defined as the intersection of the urban plume and OCO-2 overpass (Fig. 5a). Overall, the urban-influenced latitudinal band represented by 5 % of the maximum kernel density covers from 23.5° N–26° N, given multiple overpasses for Riyadh (Fig. S2). The background latitudinal range unaffected by Riyadh's urban plume

10 for estimating background then extend ~100 km from the north-most and/or south-most of derived urban plume (Fig. 5b). We abandon observations with latitudes > 26° N and < 23° N, because those retrievals are too scattered (black triangles in Fig. 5b) and indicate a second peak during few other overpasses. If the near-field wind vectors point more towards the north, screened measurements over the southern background latitudinal range is utilized, vice versa. Eventually, the background value is calculated as the mean value of the screened observations over background region (dashed green line in Fig. 5b). Two error

15 sources are incorporated into the background error— i.e., the measured (standard deviation, SD) and the retrieval errors of background observations.

 In addition to random errors (that are resolved by the inclusion of the aforementioned wind error component and broadening of the city plume), potential large bias in near-field wind direction may lead to mismatch in modeled and observed background regions and may bring relatively higher $XCO_2$ values into background $XCO_2$. However, we do not explicitly account for the

20 potential near-field wind bias's impact on forward-trajectories defined urban plume with following considerations. Firstly, we attempted to propagate a near-field wind bias into the modeled plume by rotating forward trajectories, whereas the robustness of this near-field bias can be affected by the very few wind measurements near Riyadh (further explained in Sect. 2.6.1). Secondly, the background latitude range defined by M3 with the broadening effect (blue lines in Fig. 5b) in general matches well with that observed from OCO-2 for most overpasses, which implies that the overall wind bias around our study site is not significant.

25 Lastly, even if potential wind bias may result in less accurate background range and bring elevated $XCO_2$ into the background, the background uncertainty implicitly contains information about the spatial variation in background measurements (green ribbon in Fig. 5b). In addition, the M3-derived background is the mean value of mostly hundreds of background observations (numbers in Fig. 6e), which may not be greatly affected by a few potential urban-enhanced measurements.

## 2.4 Sources of information for $CO_2$ fluxes

30 ### 2.4.1 Fossil fuel emission (ODIAC) and prior emission uncertainties

To calculate modeled $XCO_2$ enhancements, we used the latest (year 2017) version of the Open-Data Inventory for Anthropogenic Carbon dioxide (ODIAC2017 dataset, Oda et al., 2018; Oda and Maksyutov, 2011, 2015) with monthly fossil fuel $CO_2$ emissions at 1×1 km resolution (Fig. 4). ODIAC starts with annual national emission estimates, separated by fuel type, from the Carbon Dioxide Information Analysis Center (CDIAC, Andres et al., 2011), which are then re-categorized into specific ODIAC emission

35 categories on a monthly basis, i.e., point source, non-point source, cement production, international aviation and marine bunker (Oda et al., 2018). Because CDIAC only covers years up to 2013, ODIAC extrapolates emissions in 2013 for emissions in 2014 and 2015 based on BP (i.e., the British Petroleum Company) global fuel statistical data (BP, 2017). Also, ODIAC estimates point sources emissions according to a global power plants database—the Carbon Monitoring and Action (CARMA) Database (Wheeler and Ummel, 2008), and collects and distributes non-point sources using an advanced nighttime lights dataset from the Defense

Meteorological Satellite Program Operational Line Scanner (DMSP/OLS). The use of the nightlight dataset allows ODIAC to characterize the spatial patterns of the anthropogenic sources such as point sources, line sources, and diffuse sources.

To estimate emission uncertainties, we followed a method similar to those reported in Oda et al. (2015) and Fischer et al. (2017). Three emission inventories derived from different methods are inter-compared: ODIAC, the Fossil Fuel Data Assimilation System (FFDASv2; Asefi-Najafabad et al., 2014; Rayner et al., 2010) and the Emission Database for Global Atmospheric Research (EDGARv4.2; http://edgar.jrc.ec.europa.eu; Janssens-Maenhout et al., 2017). To resolve different spatial grid spacing among three inventories, we aggregated ODIAC emissions from 1 km to 0.1° gridcells. The fractional uncertainty for gridded emissions is characterized by the emission spread (1-$\sigma$, among three inventories) and mean values ($\mu$) of estimated emissions for each gridcell within a given region (10° N–40° N, 25° E–60° E; Fig. S3). In general, fractional uncertainties in gridded emissions mostly range from 60–130 % (Fig. S3) around Riyadh. Ultimately, these fractional emission uncertainties and ODIAC emissions are convolved with X-STILT's weighted column footprints to provide the XCO$_2$ uncertainties due to prior emission uncertainties.

### 2.4.2 Natural fluxes (CarbonTracker)

The trajectory-endpoint method (M1 in Sect. 2.3.1) requires the oceanic and terrestrial biospheric fluxes that come from the 3-hourly 1° × 1° product—CarbonTracker-NearRealTime (CT-NRTv2016 and v2017, http://carbontracker.noaa.gov). CT-NRT, an extension of the standard CarbonTracker (Peters et al., 2007), is designed for the OCO-2 program and uses different prior flux models and "real-time" ERA-Interim reanalysis in its transport model than regular CT, which allows for more timely model results. To calculate oceanic and biospheric XCO$_2$ changes, we multiplied these natural fluxes with column weighted footprint according to Eq. (5). Although wildfire emissions can greatly modify atmospheric XCO$_2$ (e.g., Heymann et al., 2017), we expected relatively small XCO$_2$ contributions from wildfire and hence excluded wildfire-elevated XCO$_2$ estimations, considering the studied times (wintertime overpasses) and study region (the Middle East).

### 2.5 Sensitivity analyses for X-STILT column receptors

The goal of carrying out sensitivity tests is to understand any systematic/random errors towards STILT simulations brought by receptor configurations. Under the premise of limited computational resources, proper column receptors are set up with allowable random errors. The total number of particles (NUMPAR) released from column receptors are decomposed into three parameters, i.e., the maximum release level (MAXAGL), the vertical spacing of release levels (*dh*), and the particle number per level (*dpar*).

Instead of regenerating model trajectories (Jeong et al., 2013; Mallia et al., 2015), we adopted the bootstrap method to resample model ensembles. The bootstrap approach helps construct hypothesis tests and infer error statistics (Efron and Tibshirani, 1986). The initial sample size before the bootstrap should be sufficiently large to ensure the performance of the bootstrap method and its related statistics. Thus, a "base run" of trajectories starting from the surface to 10 km with a vertical spacing of 25 m and 200 particles per level are generated and stored. For testing each parameter (MAXAGL, *dh*, or *dpar*), we fixed the other two parameters and randomly selected/resampled model particles from the base run for 100 times (allowing for repetitions). 100 urban enhancements are calculated from 100 new sets of trajectories for each test. Basic statistics—i.e., mean values and standard deviations (or fractional uncertainty, i.e., SD/mean) among these 100 enhancements—are used to infer systematic and random uncertainties in each test, respectively (with results showed in Sect. 3.1).

## 2.6 X-STILT column transport errors

Uncertainty in atmospheric transport modeling has been identified to significantly affect emission constraints (Cui et al., 2017; Lauvaux et al., 2016; Stephens et al., 2007). Here we quantify uncertainties in modeled $XCO_2$ due to transport errors caused by uncertainties in both horizontal wind fields (Sect. 2.6.1) and vertical mixing (Sect. 2.6.2).

### 2.6.1 Horizontal transport errors

Previous studies (Lin and Gerbig, 2005; Mallia et al., 2017) aimed at estimating transport error at one particular level, whereas for $XCO_2$ we account for transport error in a column sense (i.e., column transport error). Macatangay et al. (2008) briefly explained the column transport error as the weighting of transport error variances with respect to pressures. Similarly, we treated each model level separately and calculated one $CO_2$ transport error per level, denoted as $\sigma_\varepsilon^2 (CO_{2.sim.ak.n})$, following Lin and Gerbig (2005). In short, an additional wind error component ($\boldsymbol{u_\varepsilon}$) is added to the mean wind ($\boldsymbol{\bar{u}}$) and turbulent wind component ($\boldsymbol{u'}$) that are embedded in normal STILT runs (Lin et al., 2003), to randomly perturb air parcels for each level. RMS errors of u- and v-component modeled wind, error correlation timescales and length scales describe the $\boldsymbol{u_\varepsilon}$ in space and time. Details about the wind error calculations are explained in Appendix B.

For each model level ($n$), we obtained two sets of parcel distributions—i.e., one without and one with the wind error component ($\boldsymbol{u_\varepsilon}$). Then, the difference in the spread of these two distributions, or mathematically the difference in the variances of derived $CO_2$ distributions among air parcels (Lin and Gerbig, 2005), serve as the $XCO_2$ uncertainty (in ppm) due to transport error. We tested both the normal or log-normal statistics for describing this $XCO_2$ transport uncertainty. Since transport error using log-normal statistics did not show very distinct improvement from that using normal statistics, we ended up adopting normal statistics for the consideration of benefiting inverse modeling. Because the parcel distribution with $\boldsymbol{u_\varepsilon}$ (orange dots in Fig. S4) is more dispersed than the parcel distribution without $\boldsymbol{u_\varepsilon}$ (blue dots in Fig. S4), the increase in $CO_2$ variance with $\boldsymbol{u_\varepsilon}$ from that without $\boldsymbol{u_\varepsilon}$ describes the transport error for each level. However, negative values of transport error can occasionally occur, due to statistical sampling from insufficient model parcel trajectories. To resolve this technical issue, we modified Lin and Gerbig (2005) by using a regression-based approach. A weighted linear regression slope is used to describe the increase in $CO_2$ variances and then estimate transport error. More descriptions about this regression-based method are demonstrated in Appendix B. Overall, transport errors at levels within the PBL are expected to be larger than those at higher levels that approach zero.

Lastly, vertical profiles of transport errors are weighted against OCO-2's weighting functions. Following the definition of modeled $AK$-weighted $XCO_2$ in Eq. (1), the weighted column transport error follows Eq. (6),

$$\sigma_\varepsilon^2(XCO_{2.sim.ak}) = \sum_{n=1}^{nlevel} w_n^2 \, \sigma_\varepsilon^2 (CO_{2.sim.ak.n}) + 2\sum_{1 \le n < m \le nlevel} w_n \, w_m \, cov_\varepsilon(CO_{2.sim.ak.n}, CO_{2.sim.ak.m}), \tag{6}$$

where $w_n$ denotes the product of $AK_{norm}$ and $PW$ at level $n$; and $cov_\varepsilon$ represents the correlation of transport errors between every two levels $n$ and $m$ ($1 \le n < m \le nlevel$). To calculate a typical vertical error correlation length scale, we fit an exponential variogram according to transport errors and their separation distances between levels. Results of transport error at each sounding and its latitudinal integration for each track are shown in Sect. 3.4 and Sect. 3.5.

In addition to above random error component, we are aware of potential systematic wind errors in certain areas, e.g., positive wind speed bias reported over Los Angeles (Ye et al., 2017), and their impacts on both the forward- and backward- time simulations. As an attempt to resolve these obstacles, X-STILT can incorporate a near-field wind bias correction (to both backward- and forward-time simulations). By rotating model trajectories, this bias correction aims at "correcting" air parcel distributions and resultant footprints, given knowledge that the near-field wind bias can be properly interpolated. Details about this wind bias

correction are described in Appendix C. Unfortunately, only 2 radiosonde stations around Riyadh with 3 vertical pressure levels within the PBL (and sometimes with missing data) may be insufficient to correctly interpolate the near-field vertical wind biases. However, cities with meteorological profiles sampling more levels within the PBL and higher temporal frequency in reporting observed vertical winds will be more suitable sites to retrieve the near-field wind errors. Other methods include rotation and stretching of urban plumes derived from WRF-Chem (Ye et al., 2017), similar to the rotation of X-STILT air parcels, to quantify errors in wind directions and speeds. Deng et al. (2017) sought correction of wind biases in a sophisticated manner via data assimilation. Yet, the near-field correction within X-STILT can be potentially utilized in the future as a quick bias correction to the near-field wind in LPDMs, given denser wind observations and relatively flat terrains. Therefore, we decided to reduce the potential impact of wind bias on model-data comparisons using a latitudinal integration (further in Sect. 3.5).

**2.6.2 Vertical transport errors**

Vertical turbulent mixing dominants the vertical transport of air parcels and control the dilution of surface emissions within the PBL (e.g., Gerbig et al., 2008). Uncertainties in the vertical mixing or PBL height can affect both the footprint magnitude and the its spatial distribution via different horizontal advections at each altitude. Although column-integrated measurements may be less sensitive to vertical distribution of air particles than in situ measurements, vertical transport errors can have some impacts on column simulations nonetheless, due to wind shear and its interaction with vertical redistribution of air parcels (Lauvaux and Davis, 2014). Comprehensive quantifications of the vertical transport errors in a column sense are performed in Lauvaux and Davis (2014) using ensemble of surface and planetary BL parameterizations involving a regional inverse modeling framework.

Instead, we made use of the stochastic nature of STILT and propagated typical PBL height errors in the model. Changes in STILT-modeled mixed layer height modify the vertical profiles of turbulent statistics that directly control the stochastic motions of the Lagrangian air parcels (Lin et al., 2003). Thus, we obtained different air parcel trajectories with rescaled PBL heights. The resultant vertical transport error in $XCO_2$ space is calculated as the root-mean-squared errors (RMSEs) between two sets of $XCO_2$ enhancements among different receptors for each overpass. Due to this calculation, vertical transport errors are only provided at the overpass level (results in Sect. 3.5). Gerbig et al. (2008) reported typical relative PBL errors in the range of ± 20 %. Thus, we rescaled the PBL heights higher and lower by 20 % and evaluated the scaling's impact on $XCO_2$ enhancements. Because of our focus on the urban emissions and potential small $XCO_2$ enhancements contributions beyond one day backwards in time, we only rescaled PBL within the first 24 hours of transport before arrival of the air parcels at the column receptors.

**3 Results**

**3.1 X-STILT sensitivity tests with column receptors**

Fig. 6 shows test results given a sounding on 12/29/2014 around Riyadh. In general, urban enhancements increase as MAXAGL increases from 1–2 km and then stabilizes (Fig. 6a). When MAXAGL is small (< 2 km), the model fails to fully capture the $CO_2$ enhancements within the mixing height and causes underestimations on the elevated $XCO_2$. Random errors reflect the stochastic nature of the model particles leading to small fluctuations in parcel distributions and resultant signals. In this experiment, *dpar* and *dh* are fixed to 100 particles and 100 m. For testing particle number per level *(dpar),* MAXAGL is set to 6 km (well above the top of the PBL; see Appendix D for the choice of 6 km). No obvious bias is associated with mean $XCO_2$ enhancements. The random error reduces as particle numbers increases (error bars in Fig. 6b). We ended up placing 100 particles for each level, as random errors do not change dramatically from 100 to 200 particles.

In addition, we conducted two experiments using constant and uneven vertical spacings with the fixed MAXAGL of 6 km and

*dpar* of 100. Vertical spacing in the constant *dh* experiment ranges from 50 m to 1 km. Mean enhancements generally decrease as vertical spacing increases (red dots in Fig. 6c), likely because fewer release levels are insufficient to represent air parcels in a column and their interactions with surface emissions, especially under strong wind shear. We further performed two cases with uneven vertical spacing below and above a "cutoff level". Both tested three different lower spacings (of 50, 100 or 150 m) with a fixed upper spacing of 500 m. Two cases differ only in their cutoff levels (2 or 3 km). The comparison of the uneven *dh* against the constant *dh* experiment shows that their results in XCO$_2$ enhancements are fairly similar, suggesting that the lower spacing below the cutoff level matters mostly to model results, because most anthropogenic XCO$_2$ enhancements are confined within the PBL. Also, results for uneven *dh* case with the cutoff level of 3 km (blue triangles in Fig. 6c) are more closed to the "truth" implied by the constant *dh* case (red dots in Fig. 6c). To be safe, column receptors are placed from 0–3 km with a spacing of 100 m and from 3–6 km with a spacing of 500 m. See Appendix D for the derivation of the cutoff level.

To summarize, the fractional uncertainties in modeled XCO$_2$ enhancements reduce rapidly as total particle number increases (blue triangles in Fig. 6d). Our choice of column receptors and particle numbers has no noticeable bias and a fractional uncertainty of ~4 % per simulation (dashed green line). Overall changes in X-STILT column receptors have a fairly small impact on modeled anthropogenic signals, which is consistent with the finding (for biospheric signals) in Reuter et al. (2014).

**3.2 X-STILT column footprints and upwind emission contributions**

Upstream source regions and their contributions to downwind air column can be identified as the "footprint" using backward-time simulations. Here we illustrate the differences in parcel distributions and footprint patterns derived from 500 m, 3 km, and multiple levels, for one sounding at 24.4961° N on 12/29/2014 (when southwestern winds dominated). Air parcels released at 500 m are associated with large footprints in the adjacent area of Riyadh (Fig. 3b). While parcels released from a higher level of 3 km travel much faster to their upwind regions, where most parcels barely get entrained back into the PBL (Fig. 3c) and make minimal contact with the surface implied from zero footprint values in Fig. 3d. When air parcels are released from multiple levels, the column footprints (Fig. 3f) cover a broader spatial domain with relatively smaller values than footprints derived from 500 m (Fig. 3b). The intention here is to illustrate the difference in upwind influences from a PBL-based tower-like measurement versus a column-integrated measurement (e.g., satellite). As expected, surface influence arriving at an air column can be one or a few orders of magnitude smaller than that arriving at a given location. Consequently, CO$_2$ changes within the PBL are expected to be larger than column changes. If zooming into the near-field land surface, westerly winds dominated during the 12/29/2014 event. XCO$_2$ contribution maps indicate large contributions due to urban emissions of Riyadh (Fig. 7b, 7f) and small contributions from regions to the south of Riyadh (Fig. 7a, 7e), regardless of the adopted meteorological fields.

**3.3 Comparisons between methods to calculate background XCO$_2$**

As Silva and Arellano (2017) have pointed out their 4° × 4° urban extent may be too coarse for studying urban emissions we only borrowed their statistical assumption ($\mu$-$\sigma$) and used a 2° × 2° domain for computing M2S' background. M2S and M3 calculate background values from local observations. Therefore, M2S may agree better with M3 in their derived background regarding both the temporal variations and their magnitudes (black diamonds and green squares in Fig. 6e). M1 modeled background differs significantly from the other three and exhibits positive biases spanning roughly from 0.5–1.5 ppm (orange dots in Fig. 6e). Reasons to this large bias may be the accumulated transport errors as backward duration increases together with potential errors in the global concentration fields with its coarse resolution (2° × 3°).

We now focus on the comparison between M3 and M2H with objectively analyzing their advantages and limitations. On

average, M2H derived background is lower than our localized "overpass-specific" background by 0.55 ppm (Fig. 6e), which can primarily be attributed to different defined background regions. M3 defined the background region from the same track as the one over Riyadh, which guarantees that the background air contains variations due to long-term atmospheric transport, natural sources/sinks and $FFCO_2$ emissions except for local emissions (e.g., from Riyadh). Whereas the enhanced air contains the enhancements due to local emissions on top of all the information included in the background air. Therefore, the subtraction between M3-defined background and enhanced air correctly represent the $XCO_2$ portion enhanced by the local emissions. On the contrary, M2H use a fairly broad background region (0° N–60° N, 15° W–60° E in Fig. S7) to estimate gridded anomalies over all places in Europe, Middle East and North Africa. Although may yield more data, this broad spatial region may misrepresent the correct upwind region, because the wind regime can be quite different among different overpass dates or seasons.

We admit M3-defined background range and background value can be affect by potential large wind bias over cities other than Riyadh. However, the impact on background may be small and is implicitly considered in the background uncertainty (previously discussed in the last paragraph of Sect. 2.3.3). As for M2H, all regional OCO-2 measurements are lumped into its background calculation. For example, some measurements on the east-most overpass in Fig. S7 are affected by Riyadh's emissions, whereas atmospheric columns at soundings along the west two overpasses in Fig. S7 may not necessarily be the background air that eventually arrives at region around Riyadh. Thus, the regional median of $XCO_2$ may not physically indicate the accurate background that is supposed to isolate local-scale fluxes. Therefore, our localized overpass-specific background is designed and more suitable for extracting local-scale $XCO_2$ anomalies. Given relatively small urban enhancements around our study site, this 0.55 ppm difference may lead to large differences in estimated observed urban signals and emission evaluations (Sect. 4.2).

### 3.4 Latitude-dependent urban enhancements and associated uncertainties

We compare both the magnitude and shape of modeled and observed anthropogenic enhancements along the track. Models using GDAS and WRF report fairly similar $XCO_2$ peaks as bin-averaged observations for the 12/29/2014 overpass (Fig. 8). Although $XCO_2$ contributions using GDAS and WRF can differ in their spatial distributions for some receptors (Fig. 7b vs. 7f), the overall $XCO_2$ contributions integrated from all receptors along the overpass share fairly similar spatial distributions and magnitudes (Fig. 7d vs. 7h). Regarding the shape of latitude-dependent $XCO_2$ enhancements, large enhancements inferred from bin-averaged observations (solid black line in Fig. 8) cover a wider range compared to narrower modeled enhancements (dashed blue or purple lines in Fig. 8). Also, modeled versus observed enhancements exhibit a 0.1° latitudinal shift for event on 12/29/2014 (Fig. 8) and vary from 0.1°–0.4° for other events (Fig. S8). Column simulations with strong near-field influences can be sensitive to potential errors in the near-field wind speeds and directions along with errors in the gridded emissions. And the limited wind observations within the near-field land surface around Riyadh make it even harder to estimate representative wind statistics that can be directly linked with model-data mismatch in $XCO_2$ shapes and magnitudes. All these challenges lead us to perform a latitude integration on the urban $XCO_2$ enhancements over a certain latitudinal band to reduce near-field sensitivity on model-data comparisons and emission evaluations (further discussed in Sect. 3.5).

Based on available radiosonde sites over the Middle East with relatively flat terrain (white crosses in Fig, 4), regional RMS errors associated with the GDAS u- and v-component winds are mainly < 2 m s$^{-1}$ (Fig. S1) and generally smaller than those from previous studies over regions with relatively more complex terrains (Henderson et al., 2015; Lin et al., 2017). Even though positive/negative biases may exist per overpass, the averaged wind bias over a dozen tracks is fairly small, with absolute values close to zero. That is, no obvious systematic error over times is found in GDAS wind field around Riyadh. Similarly, Ye et al. (2017) reported no bias in the transport for Riyadh using WRF-Chem. Because of the spatial inhomogeneity in urban emissions,

wider parcel distributions after randomization may have higher possibilities in making contact with more emission sources than those before. Take the 12/29/2014 track as an example. Small transport errors can often be found over less polluted latitudinal range (< 24.3° N and > 24.9° N in Fig. 8). Transport errors then start to increase as few randomized parcels tend to "hit" some emission sources, even though simulated enhancements are still small (24° N–24.5° N and 24.7° N–24.8° N in Fig. 8). Although air parcels at higher altitudes are also under perturbations, the change in parcel distribution may hardly impact the column transport errors due to minimal contact of those parcels with surface emissions. As a result, the transport error per sounding for this overpass ranges from 0.07–2.87 ppm (Fig. 8). For the other tracks with more intense urban enhancements, maximum transport error per sounding can reach > 5 ppm, e.g., 2016011510 in Fig. S8. In addition, $XCO_2$ errors due to vertical mixing error are not provided at the sounding level given our method described in Sect. 2.6.2 but are reported on a per overpass basis later in Sect. 3.5.

Spatial fractional uncertainties in gridded emissions over the Middle East (Fig. S3) can be comparable to few prior studies. For instance, several commonly-used inventories differ by > 100 % over half of examined 0.1° gridcells (Gately and Hutyra, 2017) in the northeastern U.S. Resultant $XCO_2$ uncertainties due to prior emission errors range from 0.1–1.48 ppm per sounding for the overpass on 12/29/2014 (Fig. 8) and 0.04 – 2.82 ppm for all 5 overpasses (orange ribbon in Fig. S8).

Retrieval errors are reported for each sounding by the OCO-2 Lite file and exhibit a Gaussian-like distribution with the most frequent values of 0.45–0.5 ppm. Background uncertainty (e.g., green ribbon in Fig. 5b) varies from 0.77–1.00 ppm among tracks. Overall observed uncertainty per sounding varies from 0.8–1.27 ppm. Worden et al. (2017) accounted for the natural variability in observed $XCO_2$, the measurement noise errors with error covariance within spatial domain of 100 km × 10.5 km. They found the overall precision of a measurement (WL < 10) over the land is ~0.75 ppm. Our larger observed uncertainties per sounding may be attributed to no filter of observations using WL, different examined regions and time periods and inclusion of background uncertainty (for the purpose of inverse analysis).

On a per sounding basis, $XCO_2$ resulted from horizontal wind errors are comparable to or higher than $XCO_2$ emission errors. Both errors are higher than observed uncertainties. Yet, uncertainty reductions are expected as sounding-level uncertainties are aggregated along the track (Sect. 3.5).

## 3.5 Latitudinally-integrated urban signals and uncertainties

Because shapes and locations for $XCO_2$ peaks between models and observations did not line up perfectly (Sect. 3.4; Fig. 8), direct model-data comparison may lead to significant deviations for each sounding. Thus, we compare urban signals with their associated errors integrated over a latitude band for each overpass.

Firstly, we integrated bin-averaged observed or modeled anthropogenic enhancements (i.e., differences between total $XCO_2$ and overpass-specific background) along their latitudes. While multiple degrees of freedom are sacrificed by this integration, this calculation gains a larger benefit of potentially reducing the impact of near-field wind bias on emission evaluations, as long as the latitude band for aggregation is representative. Secondly, a representative latitudinal range for integration (e.g., ~24° N–25.2° N in Fig. 8) is required. Note that negative observed urban enhancements may occur when the bin-averaged total observed $XCO_2$ is slightly lower than background value. The occurrence of these negative values is partially caused by the natural variations in measured $XCO_2$ and have been included as the background uncertainty. To minimize the inclusion of those negative values, we start with the enhanced latitudinal range (e.g., 24.2° N–24.9° N in Fig. 5b) and further account for latitudinal mismatch in model-data $XCO_2$ peaks. To further include urban enhancements over the "tails" outside the distinct $XCO_2$ peaks, we then extend the previous latitudinal range by 20 % on both sides. We tested percentages other than 20 % and found no dramatic changes in estimated signals due to small enhancements outside the plume (Appendix E).

Overall, the latitude-integrated modeled $XCO_2$ signals range from 0.64–3.04 ppm-degree with a mean signal of 1.57 ppm-degree, whereas the observed signals detected by OCO-2 vary from 1.09–2.92 ppm-degree with a mean value of 1.65 ppm-degree (Table 1, Fig. 9a). The magnitudes of observed signals can be slightly affected by how observations are selected and binned up (Appendix E1).

To arrive at integrated errors per overpass, error variance-covariance matrices can be built. For example, diagonal elements comprise transport error variance per sounding/receptor with off-diagonal elements filled with error covariance between each two soundings/receptors (Fig. S9a). A correlation length scale of transport errors (~25 km) among receptor locations is estimated by fitting exponential variograms (Fig. S9b), given the transport errors (further driven by plume structures) and our choice of grid spacing between receptors. And, similar calculations are performed to integrate sounding-level errors to overpass-level errors due to various error sources. Moreover, assuming errors are independent given the multiple days to months between overpasses, overpass-level $XCO_2$ errors (Table 1) are further aggregated to arrive at an overall error for all 5 overpasses.

$XCO_2$ errors solely resulted from vertical mixing errors are in general < 15 % of the modeled signal for each overpass, whereas $XCO_2$ errors due to horizontal wind errors dominate the overall $XCO_2$ transport error (Table 1). The random uncertainties due to the choices of column receptors/parcels are negligible, < 1 % of the latitude-integrated modeled $XCO_2$ signal per track. The 68 % confidence limits of $XCO_2$ uncertainties due to errors in prior emission and transport (i.e., horizontal wind fields and vertical mixing) are 0.32 ppm-degree and 0.52 ppm-deg., which is ~20 % and 33 % of the mean modeled urban signal over 5 tracks, respectively (Table 1). The integrated $XCO_2$ transport error per track reflects the aggregate effect of several factors which interact, given how we propagate wind errors into $XCO_2$ space (Sect. 2.6):

1) The magnitude of the modeled urban $XCO_2$ enhancements. In general, air parcels that are very far away from potential upstream emitters may hardly "hit" the emission sources or gain their enhancements, even after the wind perturbation. If the estimated signal is large (e.g., 3.04 ppm-deg. on 20151216 in Table 1), its resultant integrated transport error can also be fairly large (1.83 ppm-deg. in Table 1).

2) The RMSE of u- and v-component winds. In general, larger wind errors will lead to larger changes in model trajectories and larger possibilities for perturbed trajectories in intersecting an emission source.

3) How air parcels interact with surface emissions, i.e., the geometry/angle between the model footprint (or the wind direction) and satellite swaths. Changes in this angle may fluctuate the width of enhanced latitudinal band along with the final integration latitudinal ranges (i.e., 1.10°–2.25°). If the back-trajectory or backward wind direction is more parallel to the OCO-2 swath (events on 20141227, 20151216 and 20160216 in Fig. S10), the integration range and error covariance among soundings are usually larger, which yields larger integrated $XCO_2$ errors (e.g., 1.22, 1.83, and 1.05 ppm-degree in Table 1). The averaged latitudinal range for integration is about 1.66° (~189 km) over 5 tracks.

Retrieval errors between OCO-2 soundings are found to be correlated in both space and time, with correlation coefficients (for land nadir) of 0.45 and 0.31 as a function of satellite footprint and time, respectively (Worden et al., 2017). Uncertainties of bin-averaged observed $XCO_2$ share similar source as the background uncertainties, both of which rely on spatial variation in noisy observations (in each bin or over background region). Different types of observed uncertainties are assumed to be uncorrelated. Because observations along with their uncertainties have been binned up and the satellite footprints for bin-averaged observed uncertainties are hard to track, we only account for the temporal correlation of retrieval errors between every two soundings. As a result, total observed uncertainty per track vary from 0.33–0.50 ppm-degree and the 68 % confidence limit of observed error is 0.19 ppm-deg., i.e., ~ 11 % of the mean observed signals (1.65 ppm-deg.) over total 5 overpasses.

**4 Discussions**

**4.1 Model capabilities and performances**

In this study, we demonstrate the coupling of forward- and backward-time Lagrangian particle dispersion model simulations within X-STILT and model applications in locating the urban plume, determining background $XCO_2$, identifying upwind sources, and
estimating enhanced $XCO_2$ caused by sources/sinks (Fig. 1). Specifically, backward-time simulations over an atmospheric column connect upwind emission sources with downwind atmospheric columns and generate spatial maps of this connection with additional information from satellite retrieval profiles. Although forward-time simulations from an urban box are an alternative and optional portion of X-STILT, these simulations help gain information regarding the location and size of the time-varying urban plume (Fig. 5a) and locate downwind polluted range on a satellite overpass.

Model sensitivity tests suggest two implications on simulating urban $XCO_2$ enhancements using LPDMs: 1) Receptor levels need to reach levels exceeding a typical mean PBL height to fully capture influences from surface emissions. 2) The model may capture a larger urban signal as number of levels increases. But, to minimize computational costs, one may try sparser and denser levels above and within a representative mean PBL height (the cutoff level) over upwind regions. Users can adopt their own setup of receptors in X-STILT according to combined results from sensitivity tests (Fig. 6d).

Additionally, X-STILT offers alternative solutions in dealing with errors in the meteorological fields, including regional random wind error perturbations and potential near-site wind bias corrections on model trajectories (Appendix C). For several satellite overpasses over Riyadh, models using WRF and GDAS are capable of capturing $XCO_2$ enhancements due to urban emissions, even though there remains small mismatch in the locations of model-data $XCO_2$ peaks. Model-to-model discrepancy between GDAS and WRF in latitudinally-integrated urban signals is not large, benefiting from relatively flat terrain and similar
interpolated terrain heights around Riyadh. No noticeable difference in overall RMSE in u- and v- component winds derived from radiosonde comparisons with WRF versus GDAS is reported in this case. Thus, global meteorological fields such as 0.5° GDAS can be used for studying "flat cities" like Riyadh.

When dealing with enhancements in column concentration with small signal-to-noise ratio, careful examination to modeled background $XCO_2$ should be taken care of. Although one can possibly "eyeball" the city plume from observed $XCO_2$ (especially
when a signal $XCO_2$ peak is visually distinctive), forward-time simulations with additional accounts for transport errors implemented in X-STILT may provide a more objective and efficient way (in that valuable human time is unnecessary) in figuring out the potential downwind sections along track that are affected by the city plume and extrapolating background region and its value. These advantages of overpass-specific background will become more important as more satellite tracks are incorporated within the analyses and future flux inversions.

**4.2 Implications on error analysis and future inversion using LPDMs**

Column transport uncertainties have not been rigorously examined for studies employing LPDMs like STILT and column measurements like OCO-2. In this work, we conducted comprehensive analysis towards observed errors incorporating natural and spatial $XCO_2$ variabilities, background and retrieval uncertainties; simulated errors including errors resulted from model configurations, horizontal and vertical atmospheric transport and prior emissions. On average, column transport errors (with 68 %
confidence limit) contribute to 33 % of the mean modeled urban signal over 5 overpasses, whereas the horizontal transport error on a per track basis are still substantial. We also accounted for horizontal transport error correlations among X-STILT release levels and among multiple soundings. For instance, the horizontal transport error covariance between soundings is responsible for

about 67 % of the latitude integrated errors, which emphasizes the importance of error covariance on model evaluations (e.g., Lin and Gerbig, 2005).

Estimated background uncertainty is represented by the spatial variation and retrieval errors of background observations and may be reduced given large sampling size. To further demonstrate X-STILT's potential role in inverse modeling and the potential background "bias" via different background methods on inversed results, we conducted a simple scaling factor inversion (Rodgers, 2000), based on 5 pairs of model-data latitudinally-integrated urban signals. Even though our sampling may seem to be small and the gridded urban source emissions are treated as a whole (i.e., no adjustments for emissions for each gridcell), these integrated signals and errors are chosen to reduce the impact of potential near-field wind bias on model evaluations. Also, we are partially limited by the overpasses over Riyadh (black bars in Fig. S1). The prior emissions from ODIAC are assumed to be "unbiased", which yields a prior scaling factor of unity ($\lambda_a = 1$). The prior error ($S_a$) represents the overall uncertainties of the sum and spatial spread of ODIAC emissions around Riyadh (further calculated from the inter-comparisons against FFDAS and EDGAR). Observational error covariance matrix ($S_\lambda$) contains error variances related to observation and horizontal and vertical transport errors (Table 1). Errors between every two overpasses are assumed to be independent.

Our conservative results based on GDAS suggest that the posterior scaling factor ($\hat{\lambda}$; of mean XCO$_2$ signal) and its posterior uncertainty (of the scaling factor) is $1.14 \pm 0.31$ using background from M3. However, potential errors in background XCO$_2$ defined by other methods may affect resultant observed signals and posterior scaling factors. Since the M2H- and M1-derived background values are generally lower and higher than M3 background, M2H- and M1-derived background values result in a higher and lower mean observed signal (2.30 and 0.88 ppm-degree in Table 1) than that based on M3 (1.65 ppm-degree). Furthermore, $\hat{\lambda}$ based on M2H is about 2.30, larger than that using M3 by 40 %. The $\hat{\lambda}$ derived from M3 background (1.14) can be more comparable to the WRF-Chem-based emission estimate in Ye et al. (2017). These results again emphasize the significant role of background definitions played in estimated observed signals and emission estimates. In particular, simple statistical approaches without considering the atmospheric transport may lead to erroneous conclusions (previously discussed in Sect. 3.3).

### 4.3 X-STILT's potential for broader applications

In theory, X-STILT can be applied to other column measurements and other species. The underlying Lagrangian atmospheric model (STILT) has been applied to simulate other atmospheric species, such as CO, CH$_4$ and N$_2$O (Mallia et al., 2015; Kort et al., 2008). One of the key modifications to X-STILT from STILT is the column weighting of STILT footprint values (Sect. 2.1.2). Specifically, X-STILT interpolates the OCO-2 *AK* and *PW* onto each modeled level and then applies weighting of the trajectory-level footprints before generating a horizontal footprint map. The X-STILT code can be easily modified to apply sensor-specific vertical profiles of *AK* and *PW* from other satellites or ground-based column measurements.

Lastly and more importantly, background may need to be derived differently according to different applications, e.g., local urban emissions versus regional fluxes. The overpass specific background (M3) aims at isolating the citywide emissions, so it makes use of the measurements outside the city, but still are quite closed to the city (within the few degrees latitude). However, if the study focus is to look at emissions over a much broader region (e.g., statewide emissions), background region should be defined farther away from the target region, e.g., taking the advantages of measurements from available upstream overpasses.

### 4.4 Limitations and future plans

Robust constraints on urban emission can be hampered due to their alternating-sign nature and signals potentially comparable to anthropogenic emissions (Shiga et al., 2014; Ye et al., 2017), which are also inferred from tracks we modeled over Cairo with non-

negligible biomass (results not shown in this paper). When examining summertime tracks or tracks over some other cities, potential local gradients in biospheric fluxes should be considered as those gradients can affect our overpass-specific background. Although biospheric fluxes or their resultant changes in $XCO_2$ concentrations are beyond the scope of this work, many studies have been working to address this challenge. Ye et al. (2017) incorporated biospheric fluxes from the North American Carbon Program (NACP) Multi-scale Synthesis and Terrestrial Model Intercomparison Project (MsTMIP; Fisher et al., 2016; Huntzinger et al., 2013) and performed downscaling on biospheric fluxes using MODIS-derived Green Vegetation Fraction (GVF), to provide high-resolution biospheric flux fields and estimated background $XCO_2$ by modeling. Besides, radiocarbon and terrestrial solar-induced chlorophyll fluorescence (SIF) data are helpful to isolate fossil fuel $CO_2$ and biospheric $CO_2$ (Fischer et al., 2017; Levin et al., 2003; Sun et al., 2017). In particular, recent studies have identified SIF as a better indicator/proxy of gross or net primary production than some other greenness indices over several different vegetation types (Shiga et al., 2018; Sun et al., 2017; Zuromski et al., 2018), which improves biospheric flux estimation in ecosystem models and benefits the interpretation of OCO-2/OCO-3 retrievals (Luus et al., 2017).

X-STILT extends its way to account for transport errors, background uncertainties and particle statistics in a column sense within LPDMs. Admittedly, the transport error analysis and near-field correction may work the best with the assistance of denser meteorological observing networks to characterize the error structures of transport errors. Increasing the density of surface networks may modify the wind error statistics including the wind error variances and horizontal correlated length-scale, and further impact the model transport uncertainties and inversed fluxes. Yet, this shortcoming is not inherent to X-STILT and applies to other means of quantifying the transport errors based on real data as well. The trade-off of choosing a city in the Middle East like Riyadh to minimize cloud and vegetation influences is the relatively sparse observations of surface meteorological network or aircraft. The most recent OCO-2 b8 Lite files include retrieved surface winds for each sounding. Unfortunately, most of those surface wind retrievals are not available over Riyadh, but the retrieved surface winds for other urban areas, if available, may be used for assimilation and assisting X-STILT error analysis.

Emission evaluations for different regions can be different and affected by different observational constraints. Even changes in different versions of the retrieval (Lite b7 vs. b8) may slightly affect the model-data comparisons and simple inversion results in this work. Modeled $XCO_2$ enhancements using the newer b8 differ slightly from those using b7 (purple dots in Fig. S8 vs. in Fig. S13) due to changes in the locations of the receptors, column averaging kernels, and data filtering (QF) for measurements around Riyadh. Specifically, observations from b8 may yield more overpasses with sufficient screened soundings than those from b7 (black and red bars in Fig. S1). However, much larger differences in observed enhancements are found and caused by the changes in total observed $XCO_2$ and estimated background values. Specifically, background uncertainty decreases by up to 0.1 ppm primarily attributed to smaller spread (smaller SD) of the observed $XCO_2$. Positive shifts in the total observed $XCO_2$ for b8 from b7 are found over most overpasses (Fig. S11). The M3-derived observed enhancements may be less affected by positive shifts in total observations, given similar positive shift associated with the overpass-specific background near the target urban region (dark green dashed lines in Fig. 6e vs. Fig. S12).

OCO-2 observations have been utilized in several recent studies along with this work with a particular look into relatively small areas, e.g., individual power plants (Nasser et al., 2017) and megacities (Ye et al., 2017). Even though the $XCO_2$ urban signal over Riyadh may be in general smaller than those over other large cities, both model and observation successfully detect the urban signal. Still, no summertime $XCO_2$ signal has been derived, due to the lack of screened observations (QF = 0) reported in OCO-2 Lite b7 file over most summertime tracks (black bars in Fig. S1). No diurnal variation, revisit time of 16 days and relatively narrow swath of OCO-2 may still pose challenges to urban emission estimates. We expect the inclusion of more column observations in

stationary (target) modes, e.g., by scanning over megacities by OCO-3 (Eldering et al., 2016), which may offer more concrete spatial and diurnal variabilities that benefits urban flux inversions. Many nations are devoting considerable resources in launching carbon-observing satellites that can potentially be coordinated in a larger monitoring system (Tollefson, 2016). Given that X-STILT can potentially work with most satellites (given their sensor-specific vertical profiles), we expect enhanced capability in emission constraints of urban emissions by combining column measurements with X-STILT.

*Code availability.* X-STILT is built on STILT (Lin et al., 2003) and STILT-R version 2 (Fasoli et al., 2018), which can be downloaded from GitHub repository (https://github.com/wde0924/X-STILT). The version of the X-STILT code coinciding with the work described in this manuscript is on Zenodo (http://doi.org/10.5281/zenodo.1432528). Model developments are still ongoing.

**Appendices**

**Appendix A: Four conservative criteria to select overpasses over Riyadh**

We accounted for four factors, including 1) the prevailing wind directions and downwind regions; 2) the portion of soundings with QF = 0; 3) the distance between satellite track and the city center, and 4) regional wind errors in modeled meteorological fields. In the end, we selected 5 overpasses via manual check.

**I.** First of all, we defined a spatial domain (2° latitudes by 3° longitudes) centered around Riyadh (i.e., 24.71° N, 46.74° E) and counted the total sounding numbers that fall into this domain for each overpass. This spatial domain can be determined by examining prevailing wind directions and locating downwind regions based on wind rose plot from radiosonde stations at the city center and the airport of Riyadh (with 4-character international ID of OERK and OERY) during each overpass date. Alternatively, forward-time model runs starting from a box around the center of Riyadh allows us to determine polluted latitudinal ranges on satellite overpass (Fig. 5). Detailed demonstrations about the forward-time runs are in Sect. 2.3.3. Total 43 overpasses with at least one measurement fall into this designed spatial domain for Riyadh (gray bars in Fig. S1). **II.** Next, we ensured the amount of screened observed data using warn levels/quality flags (QF). Because high warn level is associated with high total aerosol optical depth inferred from soundings (Lite b7) near Riyadh, we only used quality flag to control data quality in this study. After selecting overpasses with > 100 soundings with QF = 0, 11 overpasses remain. Most spring- and summer- time tracks (during Mar–Aug) fail to satisfy this criterion (black bars in Fig. S1). Further, we ensured enough amounts of screened observations are falling within a prescribed urban domain (1° x 1° box) around the city center (red bars in Fig. S1). Only 8 overpasses have > 50 screened soundings (red dashed line in Fig. S1). **III.** Overpasses with distinct enhancements in retrieved $XCO_2$ due to urban emissions are preferred. Near-field domain affected by PBL processes may extend over 100–1000 km based on the globally averaged ventilation time for PBL (Lin et al., 2003). We made a conservative assumption on the impacted near-field domain being a circle with a radius of 50 km around the city center. Thus, we calculated the smallest distance between soundings and city center (orange dots in Fig. S1) and most pass this filter given our examined spatial domain. **IV.** As a final step, since model results can potentially be affected by meteorological fields, regional u- and v- wind RMS errors below 3 km (derived from comparison against radiosonde stations, white crosses in Fig. 4) are calculated (numbers in brown in Fig. S1). Details on the wind error calculation are in Appendix B.

**Appendix B: Wind error calculation and regression-based transport error method in X-STILT**

In terms of the wind error component ($\boldsymbol{u_\varepsilon}$) mentioned in Sect. 2.6, two sets of parameters are used to describe, 1) $\sigma_{uverr}$, the standard deviation of horizontal wind errors (RMSE) describing to what extent should we randomly perturb air parcels; and 2) horizontal and vertical length-scales and time-scales (Lx, Lz, and Lt) determining how wind errors are correlated and decayed in space and time. We calculated different sets of wind error statistics over 3 vertical bins, i.e., 0–3 km, 3–6 km and 6–10 km, for randomizing air parcels. To obtain $\sigma_{uverr}$, observed winds at mandatory levels (i.e., 925, 850, 700, 500, 400, 300 mb) from surrounding radiosonde sites (Fig. 4) are compared against WRF- or GDAS-interpolated winds. Then, we averaged wind errors at different mandatory levels over aforementioned three vertical bins. In addition, wind errors are considered to be spatiotemporally correlated. To determine error correlation scales, differences in the wind errors are calculated and wind errors at different radiosonde stations or different reported hours (00UTC or 12UTC) are paired up based on their separation length- or time-scales. An exponential variogram is then applied to estimate the horizontal, vertical and temporal correlation scales, which are the separation scales when errors become statistically uncorrelated.

*Solution of negative transport errors:* The $CO_2$ variance derived from model trajectories after the randomizations ($\sigma^2_{\varepsilon+u'}$) can occasionally be smaller than that before the randomization ($\sigma^2_{u'}$) for a few levels, due to insufficient parcel numbers (green dots in Fig. S5). Instead of abandoning these data, we developed a regression-based method to deal with the reduction in $CO_2$ variances. Specifically, we applied linear regression lines to the two sets of $CO_2$ variances before and after the randomizations, with weights of $1/\sigma^2_{\varepsilon+u'}$. That means larger variances are weighted lesser. When we used several other ways (without the weights) to apply linear regression, extremely large regression slopes and negative y-intercept occur, which potentially leads to unreasonable large transport errors (in ppm) at lower levels within the PBL and negative transport errors aloft. Then, we scaled and recalculated $\sigma^2_{\varepsilon+u'}$ based on weighted regression slope $S_{WLR}$ and $\sigma^2_{u'}$. The regression line indicates the overall increase in $CO_2$ variance that serve as transport error in ppm:

$$\sigma^2_\varepsilon (CO_{2.sim.ak.n}) = (S_{WLR} - 1)\, \sigma^2_{u'} (CO_{2.sim.ak.n}), \tag{A1}$$

where the weighted linear regression is fitted for variances with versus without wind error component (dashed blue line in Fig. S5). Extremely large anthropogenic enhancement (e.g., >1000 ppm) for a given parcel may exist for a few cases. Thus, outliers (i.e., the upper 1st percentile of both parcel distributions before and after the randomizations) are removed for each level, before calculating variances in both $CO_2$ distributions.

**Appendix C: Correcting for wind biases within X-STILT**

While we did not apply the wind bias correction for the overpasses analyzed in this paper due to the biases being generally small (previously explained in Sect. 2.3.3), X-STILT has the capability to account for biases, if necessary. The basic idea is to correct the near-field wind biases in both forward- and backward- time trajectories. Because wind error at each observed pressure level can be quite different, vertically-weighted u- and v- wind biases were calculated by fitting logarithmic mean wind profiles based on available near-fields observed and simulated wind speeds and directions. We then calculated the deviations in latitude and longitude directions (dx, dy, with conversion from distance to degrees) given estimated u- and v- wind biases. These deviations accumulate as air parcels travel further backward or forward in time and are used to correct the location of each particle. After fixing the particle locations, Fig. S6b shows the general distribution of backward trajectory being clockwise rotated, compared to initial trajectory distribution in Fig. S6b. Air parcels in Fig. S6b appear to be "noisier" than those in Fig. S6a, due to inclusion of the random wind error component. Then the new bias-corrected set of column trajectory is used to generate spatial footprint.

This correction can also be performed to forward-time trajectory to reduce wind bias impact on best-estimated background value using the M3 method.

**Appendix D: The determinations of MAXAGL and cutoff level**

MAXAGL and a cutoff level (below which more model levels are placed) are the most important factors in determining modeled urban signals and can be determined based on few model trajectories starting from few satellite soundings for each overpass. Modeled mixing height $h$ reported for an individual air parcel at a timestamp, $h(p, t_m)$, can be very high over the upwind desert region near Riyadh. We determine MAXAGL to be the maximum mixing height for each individual air parcel. To determine a cutoff level, we calculate the averaged $h$ over all parcels as a function of backward time, as follows

$$\bar{h}(t_m) = \frac{1}{N_{tot}}\sum_{p=1}^{N_{tot}} h(p, t_m), \tag{A2}$$

where $t_m$ represents the backward timestamp, ranging from 0 to 72 hours back. The mean modeled mixing heights among air parcels at each timestamp $h(t_m)$ exhibit a diurnal cycle, where expected high values present during the daytime. Also, $\bar{h}(t_m)$ typically display relatively high values where parcels are more concentrated within a day backward, and low values as parcels disperse outwards few days back. We ended up using the maximum value of mean mixing heights over parcels and over time as a representative cutoff level.

The maximum $h(p, t_m)$ and maximum $\bar{h}(t_m)$ are 5816 m and 2420 m, for the specific sounding we showed (Sect. 3.1, Fig. 6). Considering potential uncertainties in modeled PBL or mixing heights, these two numbers are rounded to 6 km and 3 km for a representative MAXAGL and cutoff level. In addition, we generalize the rules for placing column receptors to other seasons, based on aforementioned calculations. Maximum $h(p, t_m)$ and maximum $\bar{h}(t_m)$ over the upwind region vary slightly among different soundings during different seasons. Typically, maximum $h(p, t_m)$ are mostly under 6 km for wintertime soundings (Dec, Jan, and Feb), but can reach ~7 km and 10 km for soundings in spring/fall and summer. Maximum $\bar{h}(t_m)$ are < 3 km for wintertime tracks and ~4 km and 6 km for tracks in spring/fall and summer. Therefore, column receptors are placed from the surface to 3 km with 100 m spacing and 3–6 km with 500 m spacing for wintertime overpasses with 100 parcels per level (Fig. 3e). For other seasons such as the summertime, additional receptors are placed from 6–10 km with a spacing of 1 km, to ensure the model captures entire contributions from surface emissions. Although we expect relatively similar MAXAGLs and cutoff levels for most soundings over the Middle East, due to overlaps in upwind regions, these values should be recalculated when other cities are examined (Eq. A2).

**Appendix E: Factors that may influence observed or modeled enhancements/signals**

In Section 3.5, we integrated $XCO_2$ enhancements along latitudes to estimate modeled and observed signals within a certain latitudinal band for each overpass. This latitudinal band starts with enhanced latitudinal range, then gets corrected based on model-data latitudinal shift in $XCO_2$ peaks, and finally extends by 20% of its length. Also, we tested the impact of different percentages other than 20 % on latitudinally-integrated signals. Because of relatively small $XCO_2$ enhancements over background range, the impact due to different percentages (i.e., 10 %, 15 %, 20 %, 25 %) are relatively small—i.e., with changes of 0.03 ppm and 0.06 ppm in averaged modeled and observed signals, respectively. These small changes show that our latitude integration band is representative as it does not include a second peak or miss large $XCO_2$ enhancements.

**E1 Influences on observed signals (bin-widths, warn levels)**

These modeled and observed signals reported in Sect. 3.5.1 are calculated based on the uneven sampling choice for model receptor lat/lon described in Sect. 2.1.1; i.e., with smaller bin widths of 0.025° and larger bin widths of 0.05° over which urban influences

are stronger and weaker. In addition, we tested the impact on observed signals resulted from different bin widths with constant values starting from 0.01° to 0.5°. Both the latitudinal variation and the overall observed signal for an overpass generally decrease as bin widths increase, because bin-averaged observed $XCO_2$ enhancements get smoothed out, especially over latitudes with strong urban influences. Some information is lost in latitude-integrated observed signals based on our sampling choices when comparing against the signals calculated using constant bin widths such as 0.02°. Yet, binning observations based on the lat/lon of model receptors ensures a fair comparison with the model and our uneven sampling choices may better resolve $XCO_2$ enhancements within much finer grid spacing (particularly under urban influences) in the premise of limited computational resources. In addition, warn levels (WLs) may impact the filtering of observed data, bin-averaged observed $XCO_2$, defined background and conclusion regarding the model-data comparisons. Based on 3 simple tests by selecting measurements with QF = 0 and additional WL filters (WL < 10, 12, and 15), observed signals slightly increase, as more conservative WL filtering is applied. Changes in linear regression slopes and correlation between best-estimated modeled and observed signals due to sample choices and WL filtering are small.

**E2 Influences on modeled signals (hourly vs. monthly emissions, nhrs, averaging kernel)**

An additional set of hourly scaling factors (Nassar et al., 2013) can be applied to ODIAC to downscale the monthly mean emissions down to hourly values. In this study, we use monthly mean $FFCO_2$ emissions from ODIAC and apply TIMES to only 1 of the total 5 overpasses. Simulations including TIMES are slightly larger than those without the hourly scaling factors. Also, numbers of hours may impact the modeled enhancements at each sounding/receptor. We also conducted another simulation for 12/27/2014 event using model trajectories with only 24 hours back (different from 72 hours used in main text). The decrease in anthropogenic enhancements is < 0.05 ppm per sounding, which is small due to very small surface influence from far-away emission sources. Lastly, we report overall discrepancy in the modeled anthropogenic enhancements with or without weighting by OCO-2 prior profiles to be small. The difference is about 1–2 % of the weighted modeled anthropogenic enhancements, which is much smaller than impact caused by uncertainties in transport, emissions, or different setups. Note that $XCO_2$ portion from OCO-2's prior profile is zero and averaging kernel is simply unity everywhere for non-AK weighted simulations.

*Acknowledgements.* This work is based upon work supported by the National Aeronautics and Space Administration funding under Grant No. NNX15AI41G and the National Science Foundation Graduate Research Fellowship under Grant No. DGE 1256260. We gratefully acknowledge Thomas Nehrkorn for providing modifications code to facilitate the forward-time box runs and thank Derek Mallia, Feng Deng, Arlyn Andrews, Andy Jacobson for their valuable advice on STILT-based modeling. The OCO-2 data were produced by the OCO-2 project at the Jet Propulsion Laboratory, California Institute of Technology, and obtained from the OCO-2 data archive maintained at the NASA Goddard Earth Science Data and Information Services Center. The authors acknowledge the NOAA Air Resources Laboratory (ARL) for the provision of the HYSPLIT transport and dispersion model and/or READY website (http://www.ready.noaa.gov) used in this publication. CarbonTracker CT-NRT.v2017 results provided by NOAA ESRL, Boulder, Colorado, USA from the website at http://carbontracker.noaa.gov. The support and resources from the Center for High Performance Computing (CHPC) at the University of Utah are gratefully acknowledged.

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

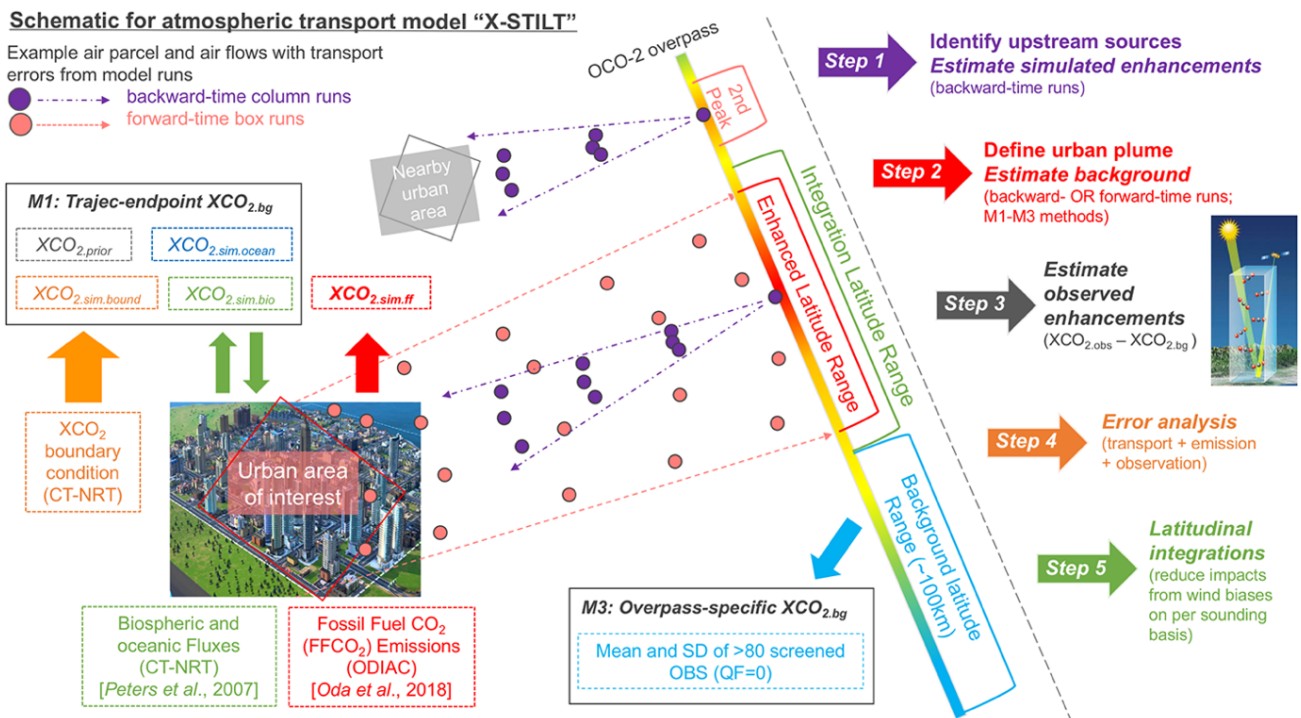

**Figure 1.** A schematic of X-STILT in 5 steps (with arrows on the right). Pink and purple dots and arrows represent the air parcels and overall air flows based on forward-time box runs and backward-time column runs with wind error component accounted for. Rainbow band is an example of one OCO-2 overpass with warmer color indicating higher observed $XCO_2$. M1 include modeled-derived biospheric, oceanic $XCO_2$ changes, $CO_2$ boundary conditions, and prior $CO_2$ portion from OCO-2. M3 requires enhanced latitude range based on either backward-time $XCO_2$ enhancements or forward-time urban plume.

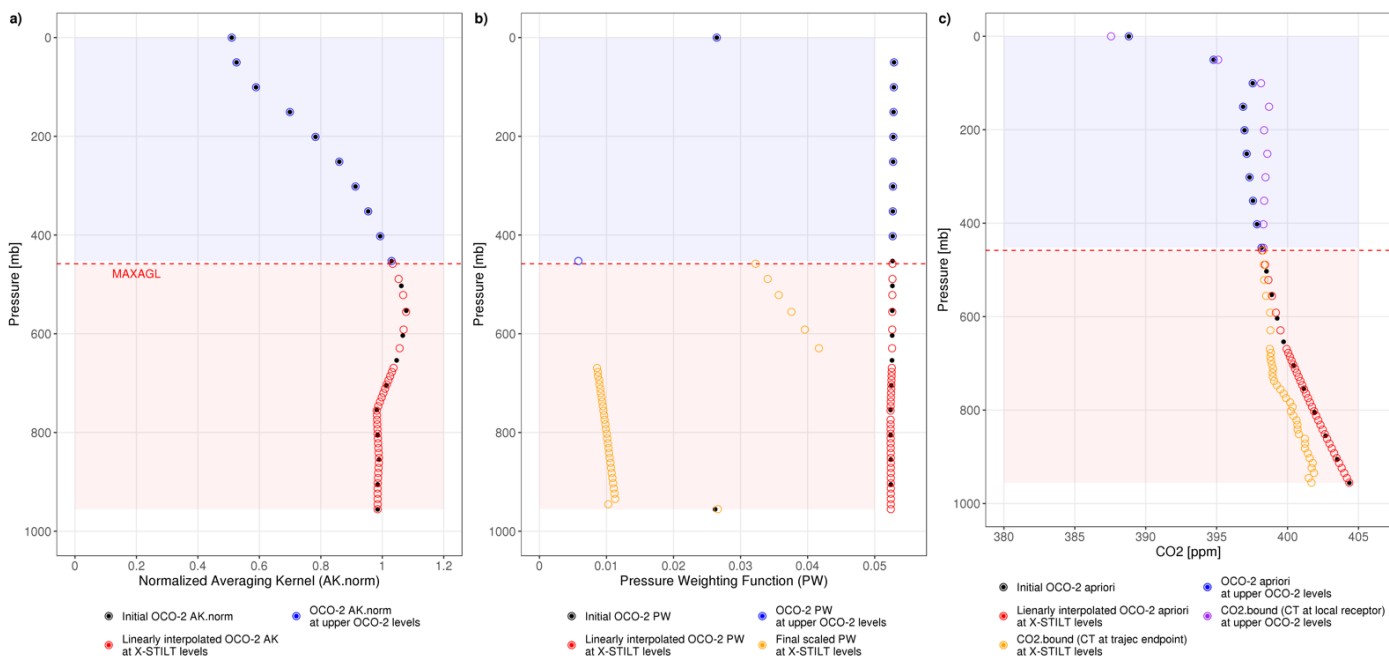

**Figure 2.** Demonstrations of interpolations on **a)** normalized averaging kernel profile, **b)** pressure weighting function and **c)** $CO_2$ boundary conditions (derived from CT-NRT) and OCO-2 a priori profile, given one sounding (lat/lon same as column receptors). Red and blue shadings denote the X-STILT release levels from the surface up to MAXAGL and upper OCO-2 levels.

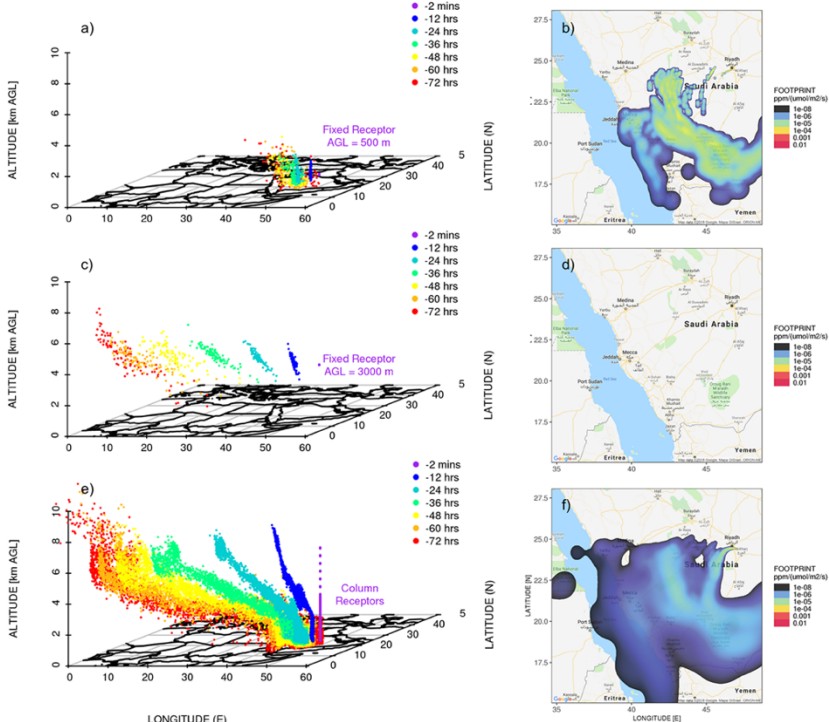

**Figure 3.** Left panels (**a, c, e**): 3D scatter plot of STILT ensembles that are initially released from a fixed receptor of 500 m, 3 km and column receptors for Riyadh on 10UTC 12/29/2014. Colors differentiate hours backwards (-2 mins, -12, -24, -36, -48, -60, and -72 hours) for each trajectory. Column receptors (e) are placed every 100 m within 3 km and every 500 m from 3–6 km. Right panels (**b, d, f**): Modeled fixed footprints vs. column footprints are plotted in blue to red gradient. Column footprints are weighted by pressure weighting functions. Only footprints values >1E-8 ppm/(μmol m$^{-2}$ s$^{-1}$) are displayed.

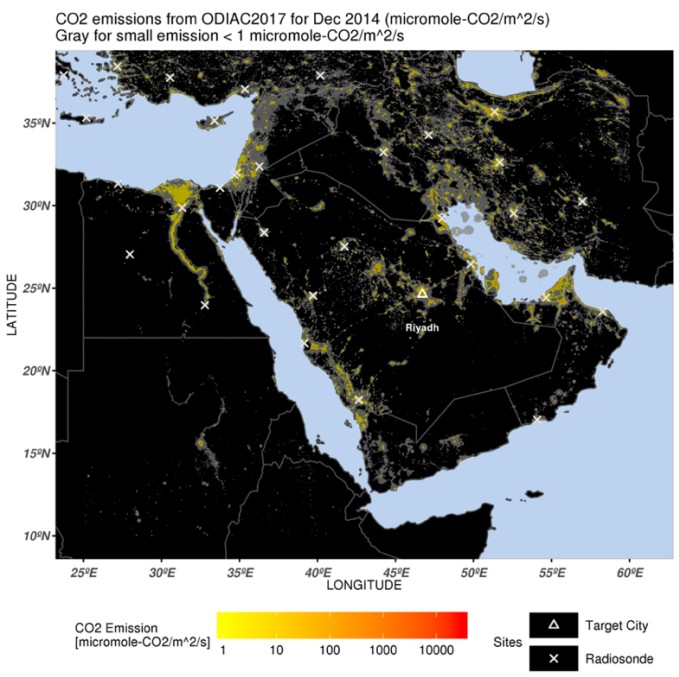

**Figure 4.** Monthly ODIAC emissions (yellow to orange) in log-scale at 1 km×1 km grid spacing for Dec 2014. White crosses and triangle denote the radiosonde networks used to evaluate provide wind error statistics and our study site of Riyadh. Small emissions (< 1μmole m$^{-2}$ s$^{-1}$) are shaded in gray.

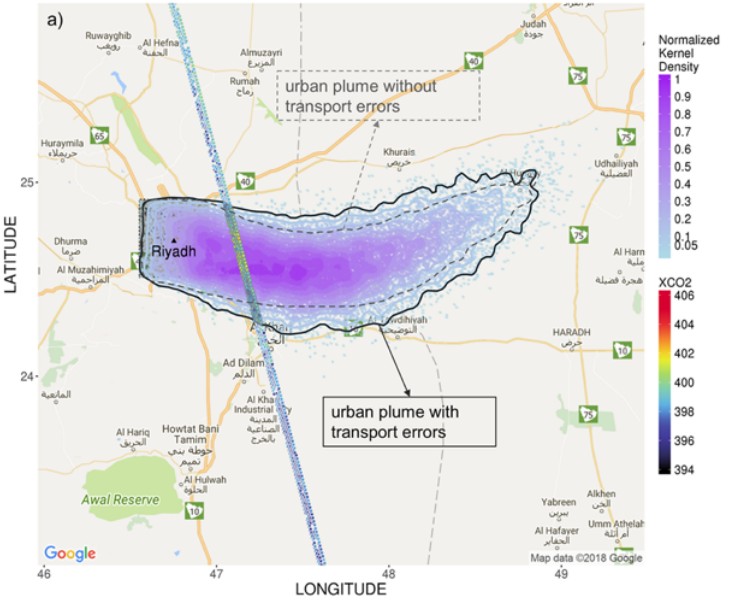

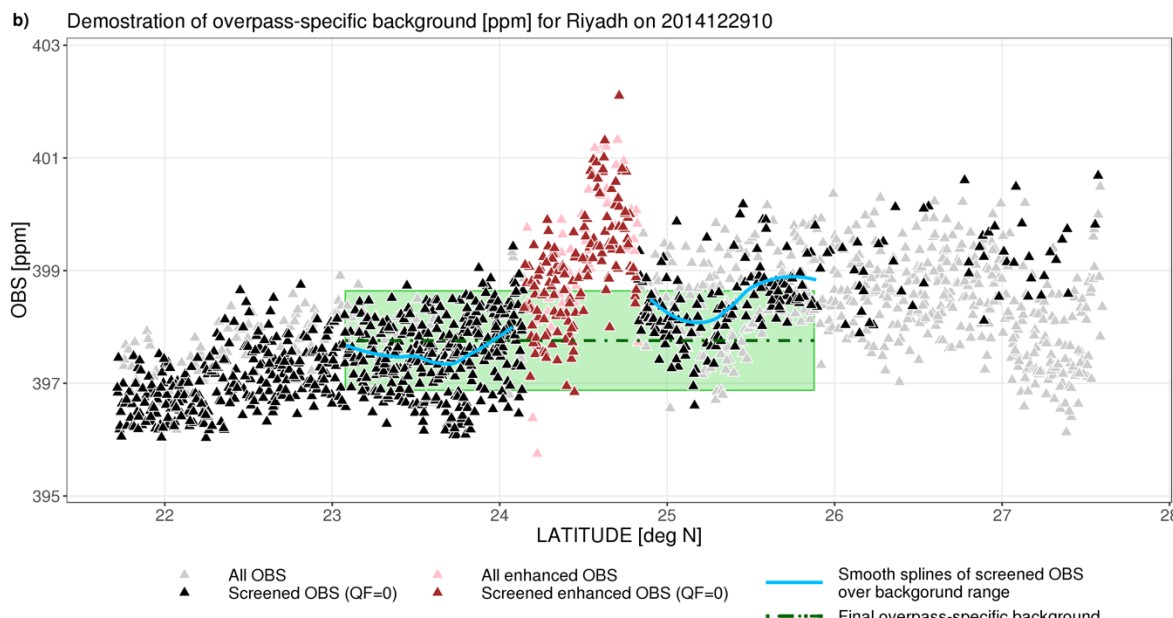

**Figure 5.** Demonstrations of overpass-specific background with an example of 12/29/2014 overpass for Riyadh. **a)** Forward particle distributions with random transport error included (blue and purple dots) and their derived normalized kernel density (solid purple contours) during OCO-2 overpass time (~3 mins) with observed $XCO_2$ (blue to red dots). Urban plumes defined based on 5 % of the max 2D kernel density estimated from parcels' distributions without (grey dashed line) and with (black solid line) transport errors. **b)** Latitude-series of observed $XCO_2$ with demonstration of background estimates. Smooth splines (solid blue lines) are drawn to visually reveal the variation of observed $XCO_2$ over background latitudinal band. Background uncertainty (green ribbon) includes both spatial uncertainty and retrieval uncertainty of observations over the background latitude range.

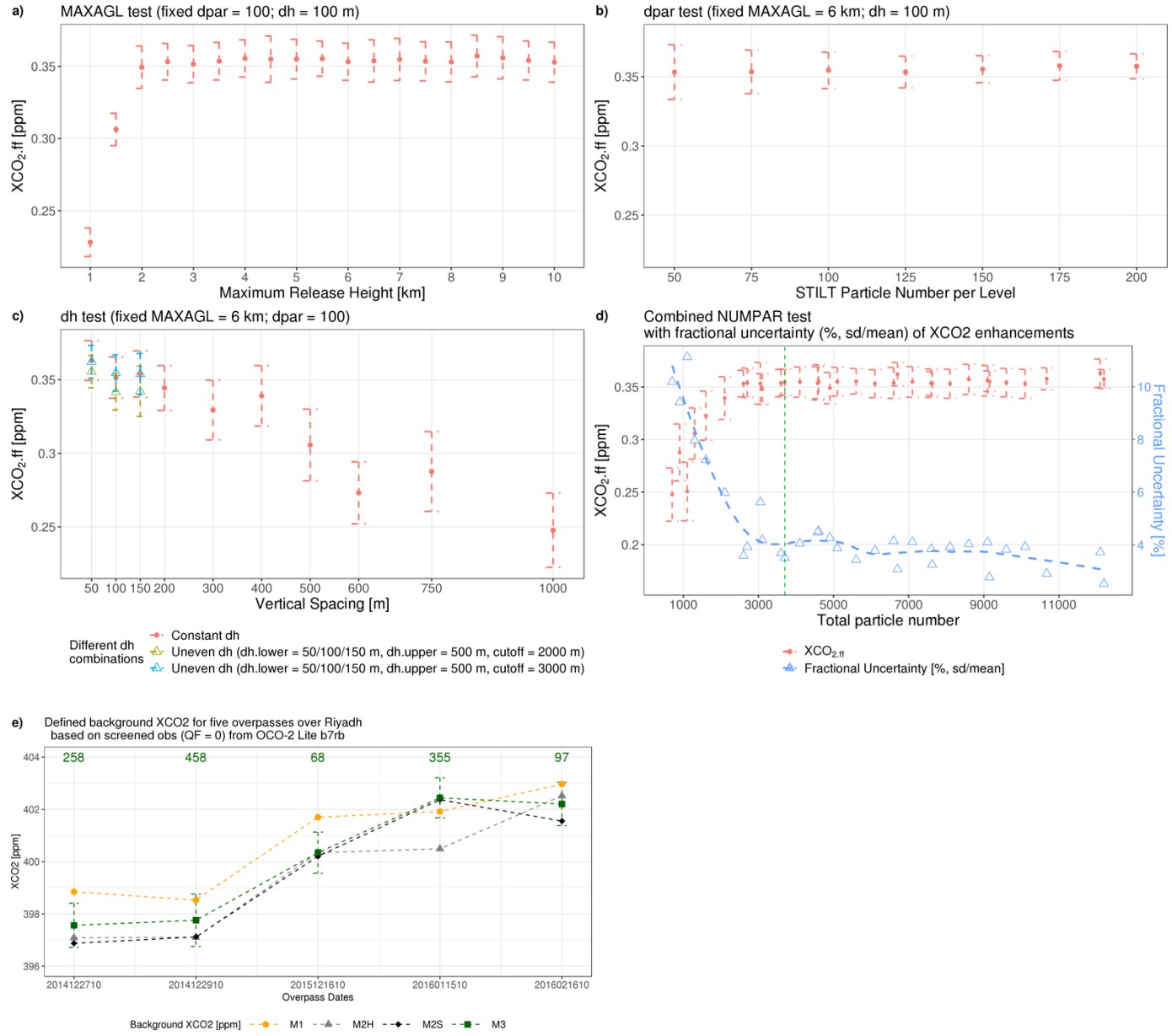

**Figure 6.** Results of sensitivity tests (**a-d**) shown for the one sounding with the largest retrieved $XCO_2$ for Riyadh and background comparisons for 5 tracks (**e**). Random error for each simulation is indicated as dashed red error bars (**a-d**); and potential biases are shown as the trend of the mean $XCO_2$ enhancements (red dots; **a-d**) derived from 100 times of bootstrap. **c)** For vertical spacing test, besides tests with constant *dh* (red dots and error bars), two other cases with uneven *dh* above and below a cutoff level are carried out. Case 1) tested 3 different lower *dh* (50, 100, and 150 m) with a fixed upper *dh* = 500 m and a cutoff level of 2000 km (yellow triangles and dashed error bars); Case 2 used the same upper and lower spacings as Case 1) but with a different cutoff level of 3000 km (blue triangles and dashed error bars). **d)** A summary plot of mean and SD of $XCO_2$ enhancements (red dots and dashed red error bars) and fractional uncertainties (%, blue triangles and dashed line) as functions of total particle number (NUMPAR). Green dashed vertical line denotes the configuration used in this study. **e)** Background comparisons using different methods (M1, M2H, M2S, and M3) for 5 tracks. The amounts of screened observations used for M3 background are labeled in dark green. M3 background errors (including spatial variation and retrieval errors over the background region) are indicated as dashed green error bars.

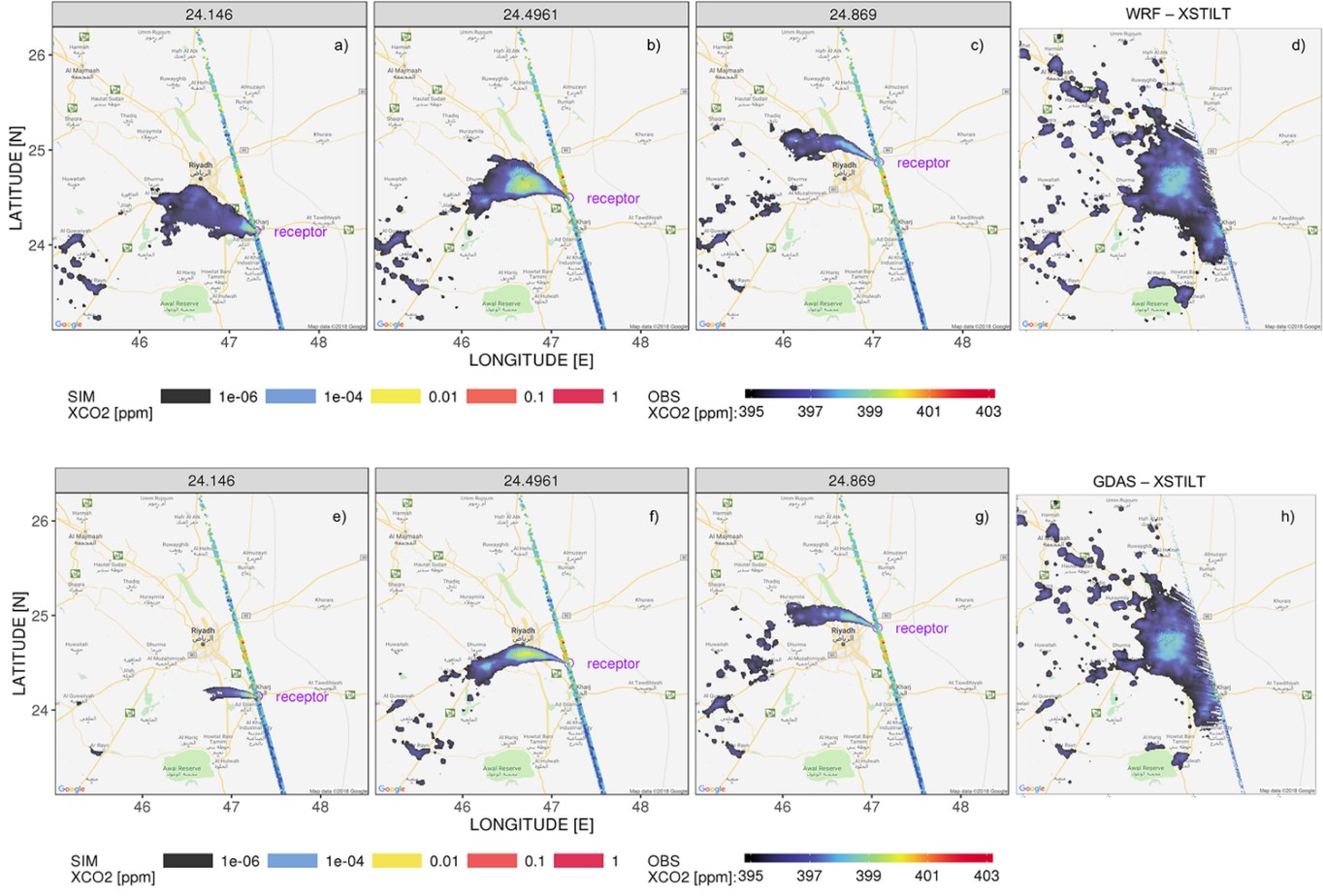

**Figure 7.** Spatial maps of 1 km × 1 km modeled $XCO_2$ contributions (ppm; log-scale) from 3 selected soundings along with screened observations (QF = 0) on 1000UTC 12/29/2014 over Riyadh, with meteorological fields driven by WRF (a-d) and GDAS (e-h). Panel d and h denote the latitude-integrated $XCO_{2,ff}$ contributions (with weights of receptor spacings, e.g., 0.02°) using WRF and GDAS, derived from spatial $XCO_2$ enhancements for over 60 column receptors along each overpass. The sum of the latitude-weighted spatial $XCO_2$ enhancements over all gridcells (in panel d or h) equals to the latitude-integrated $XCO_{2,ff}$ signal (ppm-degree) reported in Sect. 3.5. Only large enhancements $> 10^{-6}$ ppm are plotted.

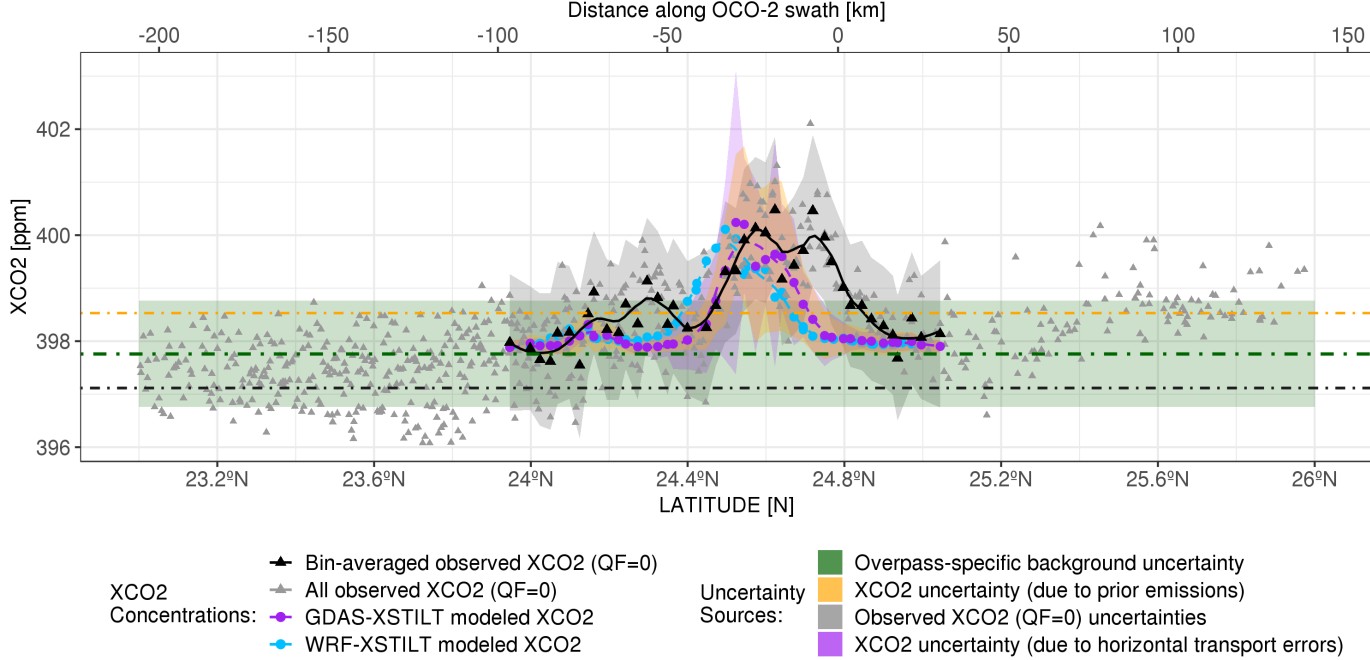

**Figure 8.** Latitude-series of sounding-level signal comparisons and error estimates for Riyadh. Screened observations with QF=0 and bin-averaged observed $XCO_2$ are shown in grey and black triangles. GDAS- and WRF- derived $XCO_2$ are displayed in purple and light blue dots, with smooth splines applied to visually reveal the main variations (purple and blue dashed lines). $XCO_2$ errors due to errors in emissions, transport and observation are drawn as yellow, purple, and light grey ribbons. Overpass-specific (M3) background $XCO_2$ is drawn as dark green dotted dashed line with its background uncertainty in light green ribbon. Background values using M1, M2H and M2S are drawn as orange, gray and black dotted dashed lines, respectively. The latitudinal range for integrating $XCO_2$ enhancements and associated various uncertainties is ~24° N–25° N in this case. The top x-axis is the distance (in km) along the OCO-2 swath from a "minimum distance sounding" that has the smallest distance from the city center.

| Overpass Dates | Lat-integrated Observed XCO₂ signal [ppm-deg.] | | | | Lat-int. Sim. XCO₂ signal [ppm-deg.] | GDAS u, v-wind RMSE [m/s] | WRF u, v-wind RMSE [m/s] | Lat-integrated Modeled XCO₂ Errors [ppm-deg.] | | | | Lat-integrated Observed XCO₂ Errors [ppm-deg.] | | | |
|---|---|---|---|---|---|---|---|---|---|---|---|---|---|---|---|
| | M1 | M2H | M2S | M3 | | | | Emiss | U, V- | PBL | Tot | Bg. | Bin | Retrieval | Tot |
| *20141227* | 0.25 | 2.41 | 2.74 | 1.62 | 1.76 | 1.94 (2.06) | 2.15 (1.85) | 0.73 | 1.22 | 0.23 | 1.44 | 0.24 | 0.14 | 0.32 | 0.42 |
| *20141229* | 0.47 | 1.75 | 1.74 | 1.09 | 0.64 | 1.81 (2.23) | 1.75 (2.03) | 0.34 | 0.41 | 0.06 | 0.54 | 0.18 | 0.11 | 0.28 | 0.35 |
| *20151216* | 1.01 | 2.92 | 3.20 | 2.92 | 3.04 | 1.74 (2.03) | | 1.04 | 1.83 | 0.36 | 2.14 | 0.26 | 0.21 | 0.39 | 0.51 |
| *20160115* | 2.04 | 3.63 | 1.54 | 1.47 | 1.06 | 1.81 (1.77) | | 0.56 | 0.60 | 0.17 | 0.84 | 0.15 | 0.10 | 0.27 | 0.33 |
| *20160216* | 0.65 | 0.77 | 2.11 | 1.17 | 1.37 | 1.78 (1.90) | | 0.70 | 1.05 | 0.16 | 1.27 | 0.24 | 0.15 | 0.33 | 0.43 |
| Mean signal *or* **SDOM** [ppm-deg.] | 0.88 | 2.30 | 2.27 | 1.65 | 1.57 | | | *0.32 (20 % of sim)* | *0.52 (hor. & ver.; 33% of sim)* | | *0.61 (39 % of sim)* | | | | *0.19 (11 % of obs)* |

$\hat{\lambda}$ [unitless]    0.75 (M1 obs vs. sim); 1.78 (M2H obs vs. sim); 1.52 (M2S obs vs. sim); 1.14 (M3 obs vs. sim)

**Table 1.** Results of signal and errors calculations for the examined five overpasses, including latitude-integrated observed versus modeled XCO₂ enhancements and errors (ppm-degree), regional wind RMSE (m/s), standard deviation of mean (SDOM) for various errors and posterior scaling factors (unitless) of the mean modeled XCO₂ signal. The GDAS (purple) and WRF (light blue) regional wind RMSEs from 0-3 km or 3-6 km are shown within or outside the bracket.

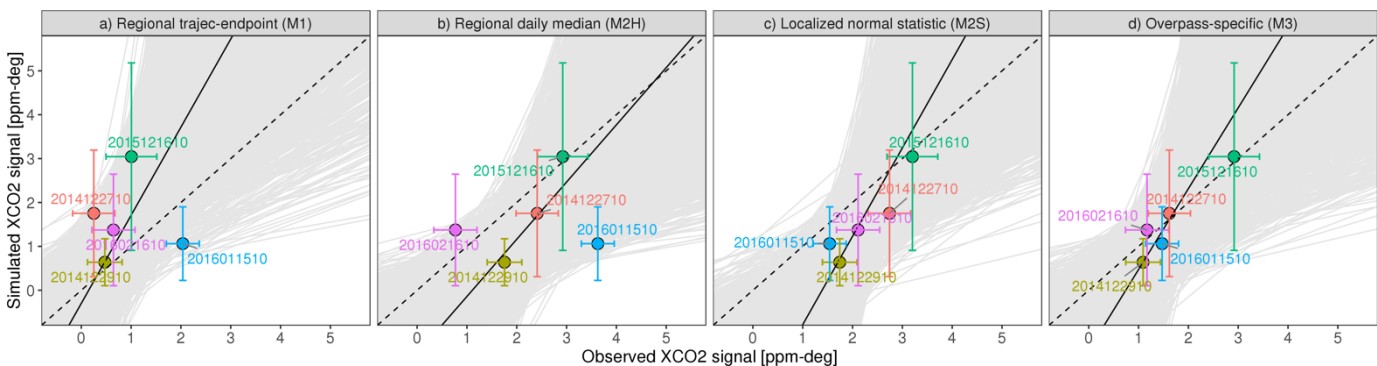

**Figure 9.** Correlation between observed and simulated anthropogenic XCO₂ signals for 5 overpasses. Colors differentiate different satellite overpass dates. Model-data comparisons using GDAS-derived XCO₂ signals and observed signals based on different background methods. Error bars along x-axis and y-axis represent the overall observed uncertainty (represented as 1-$\sigma$, including XCO₂ spatial variability, background uncertainty and retrieval errors) associated with observed signals and the overall modeled uncertainty ($\sigma$, including emission uncertainty and transport uncertainty) around modeled signals. Dashed line represents the 1:1 line. Monte Carlo experiments are performed to fit linear regression lines based on sampled model-data signals and associated errors. Regression lines with positive slopes are shaded in light grey. Median values of slopes and y-intercepts from those multiple regression lines (with positive slopes) are used to draw a linear regression (black solid line).