# Peer review of "A Lagrangian Approach Towards Extracting Signals of Urban CO2 Emissions from Satellite Observations of Atmospheric Column CO2 (XCO2): X-Stochastic Time-Inverted Lagrangian Transport model ("X-STILT v1")"

_Geoscientific Model Development, 2018_

## Short Comment (SC1) · 6 Jul 2018

It would be helpful for the reader is the link to the exact code version https://github.com/wde0924/X-STILT/releases/tag/v1.1 is references in the paper. As explained in https://www.geoscientific-model-development.net/about/manuscript_types.html the preferred reference to this release is through the use of a DOI which is then cited in the paper. For projects in GitHub a DOI for a released code version can easily be

created using Zenodo, see https://guides.github.com/activities/citable-code/ for details. Please note that in the Code Availability section you can still point the reader to the GitHub repository for the newest version even if you use a DOI for the relevant release.

Lutz Gross GMD Executive Editor

---

## Referee Comment (RC1) · Anonymous Referee #1 · 16 Jul 2018

This paper describes a new modification and application of the STILT model for total column measurements. This will allow X-STILT to be used to interpret satellite (and ground-based) total column abundances, and is a timely contribution, given the rapidly increasing number of satellite greenhouse gas total column measurements. The manuscript is thorough and technical, generally clear and well written, and suitable for this journal. I would recommend its publication after the following comments are addressed.

[Figure]

General Comments

Is X-STILT restricted to XCO2, or could it be applied to any total column tracer (e.g., XCH4, XCO, XN2O, etc.)? Could it also be applied to ground-based total column measurements? My understanding from reading this paper is that X-STILT could be applied more generally, and that you were showing a rigorous example of its functionality with OCO-2 XCO2. If this is true, its generality should be made more clear – perhaps with a more general title and introduction.

Specific Comments

P2L30-36. I don't think the current suite of CO2 satellites will completely fill in the gaps of the surface in situ networks, especially over specific locations such as cities. Certainly the future looks bright with OCO-3's "city mode", and the geostationary missions on the horizon, but I think you are overstating the impacts over cities as our satellite observing system currently stands.

Section 2.1.

I'm having trouble with your definition of the sensitivity of the satellite sensor: it seems incomplete. The column averaging kernel represents the change in the retrieved total column with respect to a perturbation in the abundance at a particular altitude. When the column averaging kernel is 0, the measurement is insensitive to changes at that altitude, and thus relies completely on the information in the a priori profile to construct the column. (Ref: OCO-2 ATBD, P58: https://co2.jpl.nasa.gov/static/docs/OCO-2%20ATBD_140530%20with%20ASD.pdf)

Weighting functions, which you mention on L33, at least to the retrieval community, refer to the Jacobian matrices, and while these are related to the averaging kernel matrices, they are not the same as the column averaging kernels (see P54 on the ATBD document above). I'd ask that this section is clarified further.

On P5, you mention the interpolation of the measurement onto the model levels. Why

wouldn't you do the reverse: interpolate the model onto the retrieval grid? Your method requires that you make several assumptions that seem to complicate your analyses (i.e., these "scaling factors" you mention). Is there a compelling reason not to interpolate the model instead of the measurements? Please explain these "scaling factors" in more detail to walk the reader through Fig. 2.

Section 2.4

A recent paper by Nassar et al. (2017) would be relevant to cite in Section 2.4. They use OCO-2 data to quantify power plant emissions, and they choose an overpass-dependent background that would be interesting to compare with your method.

Ref: Nassar, R., T. G. Hill, C. A. McLinden, D. Wunch, D. B. A. Jones, and D. Crisp (2017), Quantifying CO2 emissions from individual power plants from space, Geophys. Res. Lett., doi:10.1002/2017GL074702.

Section 2.6

My (admittedly simplistic) understanding of the transport error issue is that it is still important for models to get the vertical transport right when assimilating or inverting total column measurements, because the vertical transport sets the altitude at which advection occurs, and thus the distribution of the gas around the planet. So while the total column measurements themselves are insensitive to the altitude of the molecule within the column, it is not necessarily the case that the models are better able to reproduce the column. Indeed, the abstract of Lauvaux and Davis cited in this paragraph seems to confirm that vertical transport errors are very important for calculating fluxes from column-integrated measurements. Please address this issue further.

Furthermore, there is little discussion about the atmosphere above MAXAGL ($\sim$450 hPa). While this is unlikely to be important for CO2 emissions over regional or smaller scales, it may be important for other tracers (e.g., CH4). Can you comment on how important the tropopause altitude, for example, might impact this work? Over what

spatial and temporal scales would X-STILT properly represent the total column?

OCO-2 v7

You are using v7 of the OCO-2 data in this paper, but v8 is available and v9 will be available soon. V8 has significant improvements in the treatment of aerosols and throughput, which may be important for work over polluted urban regions. V9 will have improved pointing, especially important over topography. Please comment on whether your results will be robust against these changes to the OCO-2 data.

Technical Comments

In Section 2.1.1, I found myself wondering which version of OCO-2 data you were using, given the discussion about quality flags and albedo cutoffs, which is version-specific. I realize this is answered later in Section 2.2. I'd suggest either not mentioning the specifics of the quality filtering in section 2.1.1, or mentioning the data version in 2.1.1.

P10L14-7: I'm having trouble understanding these sentences, and I believe this wind correction may be an important step in X-STILT. Please explain in more detail.

P13L3-6: I believe you are saying that M2H is in general lower than M3. But can you say definitively that M3 is correct, and thus M2H has the bias? Also, 0.56 ppm is not small! This can be 25-50% of the enhancement.

There are several instances of omitted definite and indefinite articles, and a few typos here and there, but I assume that once this paper has been accepted, the copy editor will find and correct them more thoroughly than I have. However, I will list the ones I caught here. Between ** are the edits I suggest.

P2L17: top-down constrain*ts

P3L6-7: shed light** on CO2 emission** monitoring network*s*.

P4L18: Riyadh*,* with *a* population

P4L20: Saudi Arabia has the largest CO2 emission*s* among...

P4L32: "apple*s*-to-apple*s*"

P4L33: weighted using *the* satellite's *column averaging kernels*...

P5L17: at the same lat/lon as *the* satellite...

P5L21: compare ** overall modeled

P5L28: *The l*onger the time an air parcel..., *the* higher its footprint value...

P6L5: FFCO2 *are* derive*d* from...

P6L13: we binned ** the observed...

P6L16: estimate *the* increase in observed...

P6L22: 1x1 km resolution on ** monthly scale*s*... emission estimates by fuel type**
from the...

specific ODIAC emission categories on *a* monthly basis...

P6L31: line sources and diffuse** sources...

P8L32: which are more straightforward** and efficient** than solely *relying* on...

P9L7: boundary of *the* city...

P10L19: more suitable sites *to* retrieve...

P10L25: get around ** the impact on...

P11L6: we fit *an* exponential variogram...

P12L11: which results in *an* overall smaller footprint... Yet, column footprint*s*
cover**...

P12L14: an air column can be one or *a* few orders...

P12L18: regardless *of the* adopted meteorological fields.

P12L33: Here we emphasi*ze*...

P13L29: sensible → sensitive

P14L21: according to *the* OCO-2 Lite file...

P14L24: scatter*ed*

P15L36: latitudinal** integration ←this happens in other locations as well

P16L30: exceeding *a* certain averaged...

P17L8: emissions of *a* target city

P18L2: even large impact*s* on *the* posterior... can be caused by using *a* background derived from simplistic statistic*s*

P18L13: hampered ** due to...

P18L24: improves biospheric flux** estimation...

P20L10: for *a* few levels...

P21L17: These small changes *show* that our *latitude band integration*... a second peak or miss** large XCO2 enhancements.

P21L21: *widths*

P21L27: The word "benefit" seems out of place here.

P21L30: Based on three simpl*e* tests...

---

## Referee Comment (RC2) · Anonymous Referee #2 · 24 Aug 2018

This study is timely as the OCO2 satellite has begun producing data and relevant analyses are being conducted. I think the manuscript can contribute to the OCO2 community and, in general, the GHG community as well. The author did a lot of work including different sensitivity tests, and I think this work deserves publication after addressing issues I raise below.

Main comments:

The paper covers a lot of aspects of comparing modeled column simulations and observations. The main manuscript is long and sometimes deviates from the main story to tell; even boring although this paper is technical by nature and the information can be useful. I recommend that the authors remove some sections and technical details to the Supplement and consolidate the main text for a coherent story. Another issue is that the authors do not link the text with figures well; some of the figure captions are enormously long. In many places, the authors finish the sentences with "see Fig. X" without explaining the content of the figure well enough. I strongly recommend that the authors identify more important results (even move some figures to the Supplement, e.g., Figure 4 or 5, 7) and convey those main results with more care and clarity; please explain the figures! For example, Section 3.2 is useful (I am glad that the authors did this), but not essential for the main story given the length of the manuscript. The authors can spend the space (after moving some details) in explaining figures associated with the main results. Third, I am not quite satisfied with the transport error analysis. The problem is that the errors (mostly winds) for WRF and GDAS are not clearly defined, so it is hard to understand how good or bad the transport is and how the error can be related to signals (e.g., low winds to high signals or the impact of wrong wind directions – not presented clearly). The authors spend a lot of space to explain transport but it needs some improvement. Referring to the unpublished paper too much is not a good idea. Last, I would like the authors to comment on the utility of OCO2 for urban studies based on this work, because there is some skepticism about OCO2's capability for estimating urban emissions with relatively small areas.

Detailed comments:

P1, L19. Global assimilation data seems to be too coarse for the urban scale CO2 simulation. Why use GDAS?

P1, L21. "68 % in posterior scaling factor" should be "68 % in posterior signal" because here the bias in background is in the units of signal. Also, it is not clear what 68 % in posterior scaling factor means. Posterior uncertainty in 1-sigma? Or Does it mean the

bias in background resulted in 68% higher or lower bias in the posterior scaling factor?

P1, L22. It seems to me that the authors are referring to signal calculation, and the impact of uncertainty and bias on the urban signals by "Based on these results". I wonder if the authors can add a couple sentences that are more significant than these. If I put it differently, are these results the most important results we take home from this study?

P3, L31. Please add references related to "minimal guidance". The authors can simply add few references on uncertainties associated with atmospheric column simulations.

P3, L33 – 34: The authors underestimate recent developments in inverse modeling. There are several atmospheric inverse studies that consider transport errors and use full error matrix (not just diagonal), in particular non-CO2 studies (e.g., regional methane studies). The references there are old and does not support the statement. The authors need to be specific. I may agree that there are not many studies to incorporate full error characterizations for column-observation inversion studies, but there are now many studies to consider errors more carefully. The authors should be careful in this statement and need corrections.

Also, I am surprised that the authors use a very simple inversion – later in the section I find they are not well formulated but rudimentary – I don't see the benefit of including the inversion result in the study. Please note that there are many sophisticated inversion methods that are much more amenable to error characterizations – please do some literature review.

P4, L8-9: I don't quite understand "Most of these studies aim at extracting relatively large CO2 changes at a fixed level within the PBL or due to large emissions such as of wildfire". Which studies are the authors referring to? The point is tower vs. column or large signal vs. small? Are the authors suggesting that the study site in this work has very little CO2 changes (exchanges?)? The study areas in this study are different from other urban areas in previous studies in terms of CO2 variations or signal-to-noise

ratio? Also, related to this, why did the authors choose this study area instead of some US large cities?

P4, L13: It is not clear why the authors introduce a new background estimation method. I guess this has to do with column simulations, but please state the reason more clearly.

P4, L28: Please define "prior profile" since many "priors" are used in this paper.

P5, L4: It seems "ratios of the pressure difference between adjacent model levels over that between adjacent retrieval levels" needs more clarification. Once PW is interpolated to model levels, then the pressure difference between model levels (as the scaling factor) should be enough? Please clarify.

P5, L12: I wonder what "When WRF fields were available" means. WRF is not used for all days/hours? For the comment on the abstract, I added that GDAS alone is not sufficient for the urban scale.

Also, more importantly, the authors must add the minimum description of the WRF model, e.g., vertical and horizonal resolutions unless stated somewhere later in the sections. It is not appropriate to toss everything to another unpublished reference.

P5, L15: Is GDAS the primary choice?

P5, L19: Remove "a certain height", but directly use an explicit one, e.g., "the maximum release height" - unnecessary vagueness. I see a few places in this paper that use such a vague expression.

P5, L20: Please state what constitutes "different setups" so that the reader has a clear sense of the setups that might differ. As written, it is not clear.

P6, L26: Define "BP".

P7, L3: Please say so, if 0.1 degree is the final resolution for signal calculation, which could be coarse for a urban region.

[Figure]

P7, L10: Please comment on the 1-degree bio flux relative to the size of the study area and its potential impact (due to coarse resolution) on the inversion.

P8, L22: Please add comments on the potential impact of transport over the city when using Method 3. I note that the authors discussed the potential transport error for Method 1 (i.e., endpoint method). Wind direction could be a serious problem for Method 3. Enough overpasses (both up- and down-wind) are available for Method 3.

P9, L20: I agree with the authors that STILT configurations can affect the results. But I don't understand the use of bootstrapping here. The original sample here is from the 401 levels (too many in my opinion). However, what we are interested in is the results from different set-ups, e.g., 20, 40, levels, which can be different from the original samples of the 401 levels. In practice, 401 levels are unrealistic, e.g., for annual analysis.

P11, L15: It is surprising that MAXAGL < 2.5 km did not fully capture $CO_2$ enhancements. I would expect that there is not much surface influence above 2 km. Is it because the study region is associated with really high PBLH? As the authors stated in L30-32, the lower portion of the column should matter most. Then why would MAXAGL of ∼ 2.5 km not capture the full enhancement of $CO_2$? Please add sentences that discuss the reason for this. Actually, looking at Figure 8(a), I realize that there are only two cases below 2.5 km. So, 2.5 km itself looks fine. My guess is that even 2 km should be fine. I think the authors give the reader somewhat wrong information here, cosidering the fact that using a higher altitude for MAXAGL increases the computational cost significantly. My understanding from this is: 1) use 100 – 200 m vertical resolutions between 0 – 2 km and 2) above 2 km, use 500 m. If the authors can show even MAXAGL of 2 km is comparable to 2.5 or 3 km, this will reduce the computational cost significantly. I don't understand why the authors use 100 m for up to 3 km given the result shown in Figure 8(a), which in my opinion is too much without good reasoning. I think that some other studies will easily show denser vertical resolutions between 0 – 2 km is good enough.

P11, L34: Please clarify what the fractional uncertainty means here. How did the total particle number become >12500 with 100 particle every 100 m within 3 km?

P12, L32: "incorporates both" to "both incorporates"

P12, L37: I wonder what "we added a wind error component to broaden the urban plume (Sect. 2.4.3 and Sect. 2.6) that helps reduce the inclusion of enhanced values in the background region" means. I can understand this could help reduce strong local sources under the assumption that broadening plumes with additional errors reproduces the reality more accurately. But broadened background does not necessarily solve the bias in the wind direction that is directly related to the enhancement in the background region.

P13, L3-6: How did the authors judge which one is the more accurate background that is assumed to be close to the (unknown) true background? The impact of the background bias (0.56 ppm here) on the emission estimation depends on the magnitude of the observation; it can have only a small impact when the local observations are large.

P13, Section 3.4.1 Comparisons against OCO-2 XCO2 at selected soundings: What is the small conclusion here? After all the analysis, the authors state "we suspect that mismatch in the model-data enhancement widths is primarily due to errors in wind speeds". I expected that the authors state, e.g., "model X is better or worse than model Y in terms of wind' simulations compared to observations, and we also see better or worse in model X or Y for 'signal' comparison between model simulations and observations". Any advantage of WRF due to higher resolutions?

P13, L34: It depends on which wind observations are used. The number of sites for wind obs. in this study is too small to make a statement as shown here.

P17, L30: How large was the random error (S_lambda) relative to the background-subtracted enhancements? The 5 x 5 error matrix (if this is the model-data mismatch error covariacne, i.e., the irreducible error component in the linear model) suggests that

only 5 obs were used? If it is true, that seems to be too small, even for a simple linear regression. The scaling factor suggests the prior emissions are consistent with the observation. Is this the conclusion and what the authors expect from the comparison between modeled XCO2 and obs? The description for this simple inversion doesn't sound good at all.

P18, L4-6: I wonder if the background estimation for column CO2 from OCO2 can be improved. Somewhat disappointing.

I hope to see some discussions (a few sentences) on the utility of OCO2 for urban studies including the retrieval error (this urban region has relatively low enhancements, difficult for OCO2 to tell something), not only for this study area, but for future other regions, more generally.

Figure 7. The trajectories seem to be stratified, with each streak (looks like thick streak) somewhat disconnected from each other, which looks strange. Any explanation? Is it because of different levels?

Figure 8-e: Please use the same labels for the legend, e.g., M3.

---

## Author Comment (AC1) · 22 Sep 2018

**Response to Reviewer 1**

We thank the two reviewers for their efforts and constructive comments (https://www.geosci-model-dev-discuss.net/gmd-2018-123/#discussion). Each reviewer's comments are shown below in *italics*, followed by our point-by-point responses in blue.

**Anonymous Referee #1**

*This paper describes a new modification and application of the STILT model for total column measurements. This will allow X-STILT to be used to interpret satellite (and ground-based) total column abundances, and is a timely contribution, given the rapidly increasing number of satellite greenhouse gas total column measurements. The manuscript is thorough and technical, generally clear and well written, and suitable for this journal. I would recommend its publication after the following comments are addressed.*

We thank anonymous referee #1 for the positive feedback and have attempted to address these comments and made several clarifications and changes to the manuscript. Here are major changes made during the review process including

1) the quantification of $XCO_2$ errors due to vertical mixing errors within X-STILT (**Sect. 2.6.2**);

2) discussions on results using observations from b8 Lite files (**Sect. 4.4**) and X-STILT's potential for broader applications and expected changes (**Sect. 4.3**);

3) details on the wind bias correction within the model (**Appendix C**); and

4) updated simulations and codes built on the STILT-R version 2 (Fasoli et al., 2018) with updated description in **Sect. 2.1.2**.

**General Comments**

*"Is X-STILT restricted to XCO2, or could it be applied to any total column tracer (e.g., XCH4, XCO, XN2O, etc.)? Could it also be applied to ground-based total column measurements? My understanding from reading this paper is that X-STILT could be applied more generally, and that you were showing a rigorous example of its functionality with OCO-2 XCO2. If this is true, its generality should be made more clear – perhaps with a more general title and introduction."*

In general, we agree with the reviewer on X-STILT's potentials for wider applicability. We did not study other species than $CO_2$ or incorporate profiles from other sensors. The codes we currently modified only aim at $XCO_2$ and incorporate OCO-2 satellite profiles. However, we do foresee that X-STILT could be applied more generally. Still, we are inclined to retain the current title and majority of the manuscript, as our methods (in particular our definition of background) are built on our particular focus over urban areas. Continuous work is ongoing towards a more flexible model framework that can be more easily applied to other column measurements.

We have now clarified the model's generality and expected changes for other tracers within a new subsection (Sect. 4.3) of the main text:

> **"4.3 X-STILT's potential for broader applications**
> In theory, X-STILT can be applied to other column measurements and other species. The underlying Lagrangian atmospheric model (STILT) has been applied to simulate other atmospheric species, such as CO, $CH_4$ and $N_2O$ (Mallia et al., 2015; Kort et al., 2008). One of the key modifications to X-STILT from STILT is the column weighting of STILT footprint values (Sect. 2.1.2). Specifically, X-STILT interpolates the OCO-2 *AK* and *PW* onto each modeled level and then applies weighting of the trajectory-level footprints before generating a horizontal footprint map. The X-STILT code can be easily modified to apply sensor-specific vertical profiles of *AK* and *PW* from other satellites or ground-based column measurements.
> Lastly and more importantly, background may need to be derived differently according to different applications, e.g., local urban emissions versus regional fluxes. The overpass specific background (M3) aims at isolating the citywide emissions, so it makes use of the measurements outside the city, but still are quite closed to the city (within the few

degrees latitude). However, if the study focus is to look at emissions over a much broader region (e.g., statewide emissions), background region should be defined farther away from the target region, e.g., taking the advantages of measurements from available upstream overpasses."

**Specific Comments**

**1. P2L30-36.**

*"I don't think the current suite of CO2 satellites will completely fill in the gaps of the surface in situ networks, especially over specific locations such as cities. Certainly the future looks bright with OCO-3's "city mode", and the geostationary missions on the horizon, but I think you are overstating the impacts over cities as our satellite observing system currently stands."*

We agree with the reviewer that the surface in situ networks have their vital and irreplaceable role in the $CO_2$ measurement system. We might overstate the impacts over cities and be over-optimistic about the current suite of $CO_2$ satellites. Thus, we reworded the previous text (**P2L34-36**) as:

"Although most carbon-observing satellites have revisit times of multiple days (e.g., 3 days for GOSAT and 16 days for OCO-2), their global coverage, large number of retrievals and multi-year observations may further complement the current surface observing networks. Space-borne $CO_2$ measurements, in combination with surface $CO_2$ networks, may help reduce emission uncertainties and benefit urban emissions analysis, especially over regions with no surface observations (Duren and Miller, 2012; Houweling et al., 2004; Rayner and O'Brien, 2001)"

**2. Section 2.1.**

*"I'm having trouble with your definition of the sensitivity of the satellite sensor: it seems incomplete. The column averaging kernel represents the change in the retrieved total column with respect to a perturbation in the abundance at a particular altitude. When the column averaging kernel is 0, the measurement is insensitive to changes at that altitude, and thus relies completely on the information in the a priori profile to construct the column. (Ref: OCO-2 ATBD, P58: https://co2.jpl.nasa.gov/static/docs/OCO-2%20ATBD_140530%20with%20ASD.pdf)*
*Weighting functions, which you mention on L33, at least to the retrieval community, refer to the Jacobian matrices, and while these are related to the averaging kernel matrices, they are not the same as the column averaging kernels (see P54 on the ATBD document above). I'd ask that this section is clarified further.*

We apologize for the many confusions caused by the lack of clarity and a few mistakes made in Sect. 2.1 and Eq. (1). We modified the definition of AK as: "the sensitivity of the change in retrieved $XCO_2$ due to $CO_2$ anomaly at each retrieved grid" (**P5L3**). We reframed the entire Sect. 2.1 to clear up these confusions. And, here are some responses and clarifications:

1) In terms of representing the atmospheric column, we mislabeled $n$ in Eq. (1) in the previous version, and corrections have been made to **P5L10-13**:

   "$XCO_{2.sim.ak} = \sum_{n=1}^{nlevel}(AK_{norm,n}PW_nCO_{2.sim.n} + (I - AK_{norm,n})PW_nCO_{2.prior,n})$         (1)
   *I* is the identity vector and *n* stands for the combined vertical levels of X-STILT plus OCO-2. Specifically, we replaced OCO-2 levels with denser model release levels for the lower part of the troposphere (red circles from the surface in Fig. 2), while kept OCO-2 levels for upper part (blue circles in Fig. 2). To reduce computational cost, the air column is only simulated up to the maximum release height (MAXAGL in meters above the ground level, mAGL; Fig. 2)."

2) Background definitions can be quite different among studies depending on their applications (e.g., examined spatiotemporal scales). In this study, because our focus is about the urban emissions from a target city, we defined the background $XCO_2$ as the portion that is not affected by urban emissions and naturally broke the total modeled $XCO_2$ down to two components – $XCO_{2.ff}$ and $XCO_{2.bg}$.

- When we solely rely on model to estimate $XCO_{2.bg}$ (the trajectory-endpoint method, M1), the background $XCO_2$ is the sum of *AK*-weighted modeled biospheric and oceanic perturbations on top of $CO_2$ boundary conditions using CarbonTracker and a portion from OCO-2 a priori profiles (Eq. (2)).

- When the other background methods (M2H, M2S, M3) are being examined, the background $XCO_2$ per overpass is simple one number derived from statistical methods or the forward plume. Thus, no simulations on biospheric or oceanic anomalies involving OCO-2 prior or CarbonTracker is used in these cases.

- Clarifications have been made to the relevant text on **P5L23-28**:

  "Eq. (1) can further be rewritten as Eq. (2), since the simulated $CO_2$ profiles in Eq. (1) is comprised of $CO_2$ boundary condition plus $CO_2$ anomalies due to sources/sinks (FFCO₂, biospheric and oceanic fluxes):

  $$XCO_{2.sim.ak} = XCO_{2.sim.ff} + XCO_{2.sim.bio} + XCO_{2.sim.ocean} + XCO_{2.sim.bound} + XCO_{2.prior} = XCO_{2.sim.ff} + XCO_{2.bg}. \qquad (2)$$

  Given our focus, we defined background value as the $XCO_2$ portion not "contaminated" by urban emissions. Thus, $XCO_{2.sim.ak}$ is the sum of the $XCO_2$ enhancement due to FFCO₂ ($XCO_{2.sim.ff}$) and estimated background value ($XCO_{2.bg}$). Estimates of $XCO_2$ anomalies are further explained in Sect. 2.2.2 and four ways to estimate background values ($XCO_{2.bg}$) are proposed in Sect. 2.3."

3) Yes—the "weighting functions" we referred to in Sect. 2.1 is the column averaging kernel and pressure weighing functions, instead of the Jacobian matrices in the retrieval. We have corrected our word choices (**P5L6-7**) to "the satellite's column averaging kernels".

*On P5, you mention the interpolation of the measurement onto the model levels. Why wouldn't you do the reverse: interpolate the model onto the retrieval grid? Your method requires that you make several assumptions that seem to complicate your analyses (i.e., these "scaling factors" you mention). Is there a compelling reason not to interpolate the model instead of the measurements? Please explain these "scaling factors" in more detail to walk the reader through Fig. 2."*

Here are explanations to the Reviewer's questions on the scaling factors and interpolation of measurements' profiles:

1) *PW* function is primarily estimated based on the air mass or pressure difference (*dp*) between two layers. Fig. 2b shows that OCO-2 levels have relatively **constant pressure difference** (*dp_oco2*) and *PW* values (in black dots, except for the very top and bottom level). On the contrary, model levels are finer (in orange circles in Fig. 2b) with two different vertical spacings in **altitudes** (100 m vs. 500 m) below and above 3 km. So, those scaling factors are to adjust the *PW* function according to difference in *dp_oco2* vs. *dp_stilt*.

   For example, from 0 to 3 km, *dp_stilt* ranges from ~8-10 mb, while *dp_oco2* are mostly ~52 mb (red circles vs. black dots in Fig. 2b). Thus, we further scaled down the interpolated *PW* (red circles in Fig. 2b) by the ratio of *dp_stilt* over *dp_oco2* (e.g., sf = 10 mb/52 mb), because of less air between model levels than initial retrieval grids. Thus, final *PW* after scaling has value of ~0.01 (orange circles in Fig. 2b).

   Comparing air below 3km, fewer model levels are placed from 3-6 km (~650 to 450 mb), which gives larger *dp_stilt*, larger *PW* scaling factors and resultant larger scaled *PW* of ~0.03 –0.04 (orange circles in Fig. 2b).

   Since no model level placed above MAXAGL, *PW* stay the same as the initial OCO-2 *PW* values (blue circles in Fig. 2b). Finally, we made sure that the sum of vertical PW profile ends up being 1 for each sounding/receptor.

2) We agree with the reviewer that we could construct the 'redistribution matrix' and interpolate the model values on retrieval grid, as done in Basu et al. (2013). However, we would like to keep the current method for the following reasons (with reasons explained on **P5L14-16** in the revised version as well):

- It seems that the construction of redistribution matrix may also need to deal with or resolve the mismatch between model levels versus retrieval grids, unless model levels perfectly agree with retrieval grids.
- Most importantly, as our intention is to preserve finer variations in modeled $CO_2$ benefit from placing denser model levels within the PBL, we foresee the reversed interpolation (from model levels back to retrieval grids) could potentially involve some averaging or smoothing.

3) Clarifications have been made to the main text in **Sect. 2.1 (P5L14-22)** to explain above comments:

"Interpolations are further needed to resolve the mismatch between prescribed OCO-2 retrieval grids and model levels for the lower part of the troposphere. Our intention is to preserve the finer modeled $CO_2$ variations by performing interpolations of satellite profiles from retrieval grids to model levels. Vertical profiles of $AK_{norm}$, $PW$ and $CO_{2,prior}$ are treated as continuous functions and interpolated linearly to model grids (red circles in Fig. 2). Note that the initial OCO-2 $PW$ functions have steady value of ~0.052 (except for the very bottom and top levels; black dots in Fig. 2b), which results from constant pressure spacings ($dp\_oco2$) between two adjacent OCO-2 levels. However, X-STILT levels are much denser with smaller pressure spacings ($dp\_stilt$) or less airmass between their two adjacent levels. Therefore, the linearly interpolated $PW$ (red circles in Fig. 2b) needs an additional scaling via a set of "scaling factors" representing the ratios of pressure spacings in STILT versus OCO-2 retrieval ($dp\_stilt/dp\_oco2$), to arrive at the correct $PW$ for each finer model grid (orange circles in Fig. 2b)."

**3. Section 2.4.**

*"A recent paper by Nassar et al. (2017) would be relevant to cite in Section 2.4. They use OCO-2 data to quantify power plant emissions, and they choose an overpass-dependent background that would be interesting to compare with your method. Ref: Nassar, R., T. G. Hill, C. A. McLinden, D. Wunch, D. B. A. Jones, and D. Crisp (2017), Quantifying CO2 emissions from individual power plants from space, Geophys. Res. Lett., doi:10.1002/2017GL074702."*

We appreciate the reviewer in pointing out this paper. We have added this relevant paper when discussing different ways to determine the background in **Sect. 2.3.3 (P8L27-29)**:

"Nassar et al. (2017) derived overpass-dependent background and its uncertainty based on the averaged OCO-2 observations within four different tested background latitudinal ranges."

as well as discussing the challenges in defining background and associated uncertainties when dealing with column measurements in **Sect. 1 (P4L17-19)**:

"Lastly but more importantly, recent column studies (Nassar et al., 2017; Fischer et al., 2017) studied the impact of potential errors/biases in background values on their emission or fluxes estimates."

In general, both studies derived the background from OCO-2 measurements over the "clean" region and accounted for background uncertainties. However, as Nassar et al. (2017) focuses more on emissions from individual power plants over different regions, we did not conduct direct comparisons against our background values.

**4. Section 2.6.**

*My (admittedly simplistic) understanding of the transport error issue is that it is still important for models to get the vertical transport right when assimilating or inverting total column measurements, because the vertical transport sets the altitude at which advection occurs, and thus the distribution of the gas around the planet. So while the total column measurements themselves are insensitive to the altitude of the molecule within the column, it is not necessarily the case that the models are better able to reproduce the column. Indeed, the abstract of Lauvaux and Davis cited in this paragraph seems to confirm that vertical transport errors are very important for calculating fluxes from column-integrated measurements. Please address this issue further.*

We agree with the reviewer that while likely small, we may not completely be able to neglect the impact of vertical transport on column simulations. To quantify this error, we have now conducted another set of transport analysis to quantify $XCO_2$ errors due to vertical mixing. Sect. 2.6 in previous version has now been divided into two subsections for horizontal (Sect. 2.6.1) and vertical transport errors (Sect. 2.6.2, newly added). Section 2.6.2 has been added as:

**"2.6.2 Vertical transport errors**
Vertical turbulent mixing dominants the vertical transport of air parcels and control the dilution of surface emissions within the PBL (e.g., Gerbig et al., 2008). Uncertainties in the vertical mixing or PBL height can affect both the footprint magnitude and the its spatial distribution via different horizontal advections at each altitude. Although column-integrated measurements may be less sensitive to vertical distribution of air particles than in situ measurements, vertical transport errors can have some impacts on column simulations nonetheless, due to wind shear and its interaction with vertical redistribution of air parcels (Lauvaux and Davis, 2014). Comprehensive quantifications of the vertical transport errors in a column sense are performed in Lauvaux and Davis (2014) using ensemble of surface and planetary BL parameterizations involving a regional inverse modeling framework.

Instead, we made use of the stochastic nature of STILT and propagated typical PBL height errors in the model. Changes in STILT-modeled mixed layer height modify the vertical profiles of turbulent statistics that directly control the stochastic motions of the Lagrangian air parcels (Lin et al., 2003). Thus, we obtained different air parcel trajectories with rescaled PBL heights. The resultant vertical transport error in $XCO_2$ space is calculated as the root-mean-squared errors (RMSEs) between two sets of $XCO_2$ enhancements among different receptors for each overpass. Due to this calculation, vertical transport errors are only provided at the overpass level (results in Sect. 3.5). Gerbig et al. (2008) reported typical relative PBL errors in the range of ± 20 %. Thus, we rescaled the PBL heights higher and lower by 20 % and evaluated the scaling's impact on $XCO_2$ enhancements. Because of our focus on the urban emissions and potential small $XCO_2$ enhancements contributions beyond one day backwards in time, we only rescaled PBL within the first 24 hours of transport before arrival of the air parcels at the column receptors."

We further briefly mentioned the $XCO_2$ errors due to vertical mixing error as in **Sect. 3.5**:

"$XCO_2$ errors solely resulted from vertical mixing errors are in general < 15 % of the modeled signal for each overpass, whereas $XCO_2$ errors due to horizontal wind errors dominate the overall $XCO_2$ transport error (Table 1)."

*Furthermore, there is little discussion about the atmosphere above MAXAGL (~450 hPa). While this is unlikely to be important for CO2 emissions over regional or smaller scales, it may be important for other tracers (e.g., CH4). Can you comment on how important the tropopause altitude, for example, might impact this work? Over what spatial and temporal scales would X-STILT properly represent the total column?*

We strongly agree with the reviewer that atmosphere above MAXAGL can be less important for $XCO_2$ emissions, but can be important for other tracers like methane, due to chemical productions and losses along the atmospheric transport. In theory, the vertical profile of methane can be sensitive to the tropopause altitude since it rapidly decreases in the stratosphere.

We admit that no chemical reaction is considered in X-STILT at this point and the model levels are only placed over the lower troposphere, given our focus on urban-scale $CO_2$ surface emissions (that only get mixed within the PBL). As particles released from higher levels can hardly get entrained and make contact with the near-field land surface, the X-STILT column particles may properly represent the total column over the urban scale for less than one day. Due to our specific focus on urban $CO_2$, we decided not to add related content in the main text, but to address the reviewer's question here.

However, we may expect small impact on our work with reasoning showed below.

- Although variations for atmosphere above MAXAGL are not explicitly modeled or represented by X-STILT particles given the way we release air parcels, those variations are part of the defined background.

- Take methane as an example. Recall that the background air defined by the overpass-specific method (M3) are atmospheric columns located outside (but not far away from) the city plume. The background value is derived from measured $XCH_4$ over those background atmospheric columns. In fact, those measured $XCH_4$ are the resultant $XCH_4$ after being produced or destructed along their way (backward in time) and have contained information about the upper tropospheric and stratospheric variations over upwind regions. Since the background air and the plume-elevated air are not far away from each other, we may assume no big difference in their tropopause altitudes. Thus, both background and plume-elevated $XCH_4$ contains information about the $XCH_4$ over atmosphere over MAXAGL. And, the difference between the two gives us the methane enhancements due to urban emissions.

- One note on the background definitions: When using the trajectory-endpoint method (M1), model trajectories are used to model the $XCO_2$ boundary condition. However, when using the other three background methods (M2H, M2S and M3), model trajectories are only used to estimate the $XCO_2$ anomalies due to different sources and sinks. The choice of max release height may impact the modeled $XCO_2$ enhancements. While as shown in the MAXAGL sensitivity test, almost no changes in $XCO_2$ enhancement due to increase in MAXAGL.

**5. OCO-2 v7.**

*You are using v7 of the OCO-2 data in this paper, but v8 is available and v9 will be available soon. V8 has significant improvements in the treatment of aerosols and throughput, which may be important for work over polluted urban regions. V9 will have improved pointing, especially important over topography. Please comment on whether your results will be robust against these changes to the OCO-2 data.*

We thank the reviewer in pointing out different OCO-2 versions and look forward to the upcoming b9 product to help future studies over cities with complex terrain. We have now performed few more simulations and analysis using version 8 Lite product and briefly depicted the changes in observed and simulated urban $XCO_2$ signals in **Sect. 4.4 (P19L23-33)**. As ongoing improvements made in OCO-2 retrievals are modifying the observations, it may not be that fair for us to comment the robustness in our results, especially given model evaluation against observations.

Text has been added:

"Emission evaluations for different regions can be different and affected by different observational constraints. Even changes in different versions of the retrieval (Lite b7 vs. b8) may slightly affect the model-data comparisons and simple inversion results in this work. Modeled $XCO_2$ enhancements using the newer b8 differ slightly from those using b7 (purple dots in Fig. S8 vs. in Fig. S13) due to changes in the locations of the receptors, column averaging kernels, and data filtering (QF) for measurements around Riyadh. Specifically, observations from b8 may yield more overpasses with sufficient screened soundings than those from b7 (black and red bars in Fig. S1). However, much larger differences in observed enhancements are found and caused by the changes in total observed $XCO_2$ and estimated background values. Specifically, background uncertainty decreases by up to 0.1 ppm primarily attributed to smaller spread (smaller SD) of the observed $XCO_2$. Positive shifts in the total observed $XCO_2$ for b8 from b7 are found over most overpasses (Fig. S11). The M3-derived observed enhancements may be less affected by positive shifts in total observations, given similar positive shift associated with the overpass-specific background near the target urban region (dark green dashed lines in Fig. 6e vs. Fig. S12)."

**Technical Comments**

**6. Section 2.1.1**

*I found myself wondering which version of OCO-2 data you were using, given the discussion about quality flags and albedo cutoffs, which is version-specific. I realize this is answered later in Section 2.2. I'd suggest either not mentioning the specifics of the quality filtering in section 2.1.1, or mentioning the data version in 2.1.1.*

> We realize the confusion caused by the introduction on quality flags in Sect. 2.1.1. We have removed this discussion (the data filtering on observations, i.e., QF and aerosols cutoffs) in Sect. 2.1.1.

**7. P10L14-7**

*I'm having trouble understanding these sentences, and I believe this wind correction may be an important step in X-STILT. Please explain in more detail.*

> We apologize for the confusion and clarify that this attempt to correct wind biases can be an important part in X-STILT, in particular over places with large systematic wind error, less complex terrain and denser wind observations. We briefly mentioned this correction and discussed its limitation in **Sect. 2.6.1 (P11L33- P12L9)** and added an **Appendix C** for details of this bias-correction:
>
>> **"Appendix C: Correcting for wind biases within X-STILT**
>> While we did not apply the wind bias correction for the overpasses analyzed in this paper due to the biases being generally small (previously explained in Sect. 2.3.3), X-STILT has the capability to account for biases, if necessary. The basic idea is to correct the near-field wind biases in both forward- and backward- time trajectories. Because wind error at each observed pressure level can be quite different, vertically-weighted u- and v- wind biases were calculated by fitting logarithmic mean wind profiles based on available near-fields observed and simulated wind speeds and directions. We then calculated the deviations in latitude and longitude directions (dx, dy, with conversion from distance to degrees) given estimated u- and v- wind biases. These deviations accumulate as air parcels travel further backward or forward in time and are used to correct the location of each particle. After fixing the particle locations, Fig. S6b shows the general distribution of backward trajectory being clockwise rotated, compared to initial trajectory distribution in Fig. S6b. Air parcels in Fig. S6b appear to be "noisier" than those in Fig. S6a, due to inclusion of the random wind error component. Then the new bias-corrected set of column trajectory is used to generate spatial footprint. This correction can also be performed to forward-time trajectory to reduce wind bias impact on best-estimated background value using the M3 method."
>
> Unfortunately, we ended up not performing bias correction to model particles, because the wind observation sites are not perfect around Riyadh to estimate a robust wind error to rotate model particles. We mentioned these limitations and other more comprehensive methods in **Sect. 2.6.1 (P12L1-9)**:
>
>> "Unfortunately, only 2 radiosonde stations around Riyadh with 3 vertical pressure levels within the PBL (and sometimes with missing data) may be insufficient to correctly interpolate the near-field vertical wind biases. However, cities with meteorological profiles sampling more levels within the PBL and higher temporal frequency in reporting observed vertical winds will be more suitable sites to retrieve the near-field wind errors. Other methods include rotation and stretching of urban plumes derived from WRF-Chem (Ye et al., 2017), similar to the rotation of X-STILT air parcels, to quantify errors in wind directions and speeds. Deng et al. (2017) sought correction of wind biases in a sophisticated manner via data assimilation. Yet, the near-field correction within X-STILT can be potentially utilized in the future as a quick bias correction to the near-field wind in LPDMs, given denser wind observations and relatively flat terrains. Therefore, we decided to reduce the potential impact of wind bias on model-data comparisons using a latitudinal integration (further in Sect. 3.5)."

**8. P13L3-6:**

*I believe you are saying that M2H is in general lower than M3. But can you say definitively that M3 is correct, and thus M2H has the bias? Also, 0.56 ppm is not small! This can be 25-50% of the enhancement.*

We agree that a mean difference of 0.56 ppm (between M2H- and M3-derived background values) is actually quite large, given small column enhancement. And, we cannot definitively say that our method is correct since the "true" background is more of an unknown value. So, we have reworded some relevant sentences (e.g., "bias" to "mean difference").

However, we would argue that M2H may not be suitable for local/urban studies and have rewritten the latter two paragraphs in **Sect. 3.3** (**P13L36-P14L17**) to objectively discuss the pros and cons of M2H and M3 methods (also pasted below):

"We now focus on the comparison between M3 and M2H with objectively analyzing their advantages and limitations. On average, M2H derived background is lower than our localized "overpass-specific" background by 0.55 ppm (Fig. 6e), which can primarily be attributed to different defined background regions. M3 defined the background region from the same track as the one over Riyadh, which guarantees that the background air contains variations due to long-term atmospheric transport, natural sources/sinks and FFCO$_2$ emissions except for local emissions (e.g., from Riyadh). Whereas the enhanced air contains the enhancements due to local emissions on top of all the information included in the background air. Therefore, the subtraction between M3-defined background and enhanced air correctly represent the XCO$_2$ portion enhanced by the local emissions. On the contrary, M2H use a fairly broad background region (0° N– 60° N, 15° W–60° E in Fig. S7) to estimate gridded anomalies over all places in Europe, Middle East and North Africa. Although may yield more data, this broad spatial region may misrepresent the correct upwind region, because the wind regime can be quite different among different overpass dates or seasons.

We admit M3-defined background range and background value can be affect by potential large wind bias over cities other than Riyadh. However, the impact on background may be small and is implicitly considered in the background uncertainty (previously discussed in the last paragraph of Sect. 2.3.3). As for M2H, all regional OCO-2 measurements are lumped into its background calculation. For example, some measurements on the east-most overpass in Fig. S7 are affected by Riyadh's emissions, whereas atmospheric columns at soundings along the west two overpasses in Fig. S7 may not necessarily be the background air that eventually arrives at region around Riyadh. Thus, the regional median of XCO$_2$ may not physically indicate the accurate background that is supposed to isolate local-scale fluxes. Therefore, our localized overpass-specific background is designed and more suitable for extracting local-scale XCO$_2$ anomalies. Given relatively small urban enhancements around our study site, this 0.55 ppm difference may lead to large differences in estimated observed urban signals and emission evaluations (Sect. 4.2)."

Lastly, we may admit that our definition (using one mean value to represent the background) is not perfect and are aware of the potential wind bias impact on background definition. So, we made several attempts in this work:
1) by widening the polluted latitude range by bringing a wind error component (Sect. 2.3.3);
2) by trying to correct the wind bias on forward plume via a near-field correction (Appendix C, better for regions with denser wind observations); and
3) by introducing background uncertainty as the combined error impacts from retrieval error and natural variation (SD) of observed XCO$_2$ over background latitude range (Sect. 2.3.3). Retrieval errors of measurements over the background range have now been included in the background error as well, based on a comment from the Reviewer #2.

Relevant discussions based on above point 1) to point 3) have been made to **Sect. 2.3.3** on **P9L17-28** (also pasted here):

"In addition to random errors (that are resolved by the inclusion of the aforementioned wind error component and broadening of the city plume), potential large bias in near-field wind direction may lead to mismatch in modeled and observed background regions and may bring relatively higher XCO$_2$ values into background XCO$_2$. However, we do not explicitly account for the potential near-field wind bias's impact on forward-trajectories defined urban plume with following considerations. Firstly, we attempted to propagate a near-field wind bias into the modeled plume by rotating

forward trajectories, whereas the robustness of this near-field bias can be affected by the very few wind measurements near Riyadh (further explained in Sect. 2.6.1). Secondly, the background latitude range defined by M3 with the broadening effect (blue lines in Fig. 5b) in general matches well with that observed from OCO-2 for most overpasses, which implies that the overall wind bias around our study site is not significant. Lastly, even if potential wind bias may result in less accurate background range and bring elevated $XCO_2$ into the background, the background uncertainty implicitly contains information about the spatial variation in background measurements (green ribbon in Fig. 5b). In addition, the M3-derived background is the mean value of mostly hundreds of background observations (numbers in Fig. 6e), which may not be greatly affected by a few potential urban-enhanced measurements."

**9.** *There are several instances of omitted definite and indefinite articles, and a few typos here and there, but I assume that once this paper has been accepted, the copy editor will find and correct them more thoroughly than I have. However, I will list the ones I caught here. Between \*\* are the edits I suggest.*

*P2L17: top-down constrain\*ts\* P3L6-7: shed light\*\* on CO2 emission\*\* monitoring network\*s\*. P4L18: Riyadh\*,\* with \*a\* population P4L20: Saudi Arabia has the largest CO2 emission\*s\* among... P4L32: "apple\*s\*-to-apple\*s\*" P4L33: weighted using \*the\* satellite's \*column averaging kernels\*... P5L17: at the same lat/lon as \*the\* satellite... P5L21: compare \*\* overall modeled P5L28: \*The l\*onger the time an air parcel..., \*the\* higher its footprint value... P6L5: FFCO2 \*are\* derive\*d\* from... P6L13: we binned \*\* the observed... P6L16: estimate \*the\* increase in observed... P6L22: 1x1 km resolution on \*\* monthly scale\*s\*... emission estimates by fuel type\*\* from the... specific ODIAC emission categories on \*a\* monthly basis... P6L31: line sources and diffuse\*\* sources... P8L32: which are more straightforward\*\* and efficient\*\* than solely \*relying\* on... P9L7: boundary of \*the\* city... P10L19: more suitable sites \*to\* retrieve... P10L25: get around \*\* the impact on... P11L6: we fit \*an\* exponential variogram... P12L11: which results in \*an\* overall smaller footprint... Yet, column foot- print\*s\* cover\*\*... P12L14: an air column can be one or \*a\* few orders... P12L18: regardless \*of the\* adopted meteorological fields. P12L33: Here we emphasi\*ze\*... P13L29: sensible → sensitive P14L21: according to \*the\* OCO-2 Lite file... P14L24: scatter\*ed\* P15L36: latitudinal\*\* integration ←this happens in other locations as well P16L30: exceeding \*a\* certain averaged... P17L8: emissions of \*a\* target city P18L2: even large impact\*s\* on \*the\* posterior... can be caused by using \*a\* background derived from simplistic statistic\*s\* P18L13: hampered \*\* due to... P18L24: improves biospheric flux\*\* estimation... P20L10: for \*a\* few levels... P21L17: These small changes \*show\* that our \*latitude band integration\*... a second peak or miss\*\* large XCO2 enhancements. P21L21: \*widths\* P21L27: The word "benefit" seems out of place here. P21L30: Based on three simpl\*e\* tests...*

We sincerely thank the reviewer in thoroughly reading the manuscript and pointing out these issues listed above. We have corrected them all in the relevant text.

---

## Author Comment (AC2) · 22 Sep 2018

**Response to Reviewer 2**

We thank the two reviewers for their efforts and constructive comments (https://www.geosci-model-dev-discuss.net/gmd-2018-123/#discussion). Each reviewer's comments are shown below in *italics*, followed by our point-by-point responses in blue.

**Anonymous Referee #2**

*This study is timely as the OCO2 satellite has begun producing data and relevant analyses are being conducted. I think the manuscript can contribute to the OCO2 community and, in general, the GHG community as well. The author did a lot of work including different sensitivity tests, and I think this work deserves publication after addressing issues I raise below.*

We thank Reviewer #2 for the positive feedback and constructive comments, which help improve both the scientific contents and the flow of this manuscript. In general, we identified several main concerns raised by Reviewer #2, in terms of 1) the flow of the manuscript, 2) transport error analysis, 3) background estimates, and 4) bias in wind direction and its impact on background estimates.

We have tried to address each comment and make clarifications/modifications to the manuscript accordingly. Also, we recently merged the X-STILT model codes with the newer version of STILT (i.e., STILT-R version 2 by Fasoli et al., 2018) and updated figures and results.

**Main Comments**

***1.*** *The paper covers a lot of aspects of comparing modeled column simulations and observations. The main manuscript is long and sometimes deviates from the main story to tell; even boring although this paper is technical by nature and the information can be useful. I recommend that the authors remove some sections and technical details to the Supplement and consolidate the main text for a coherent story. Another issue is that the authors do not link the text with figures well; some of the figure captions are enormously long. In many places, the authors finish the sentences with "see Fig. X" without explaining the content of the figure well enough. I strongly recommend that the authors identify more important results (even move some figures to the Supplement, e.g., Figure 4 or 5, 7) and convey those main results with more care and clarity; please explain the figures! For example, Section 3.2 is useful (I am glad that the authors did this), but not essential for the main story given the length of the manuscript. The authors can spend the space (after moving some details) in explaining figures associated with the main results.*

We thank the reviewer for these valuable comments and suggestions that help better re-organize our manuscript. We have moved several figures from the main text to the supplement and modified the legend of almost every figure by removing redundant sentences. More explanations in the main text have now been added when explaining a figure (e.g., "red circles in Fig. X"). We removed some less important results and replaced with more important analysis and discussions suggested by both reviewers. **Table 1** is now added to summarize main results from signal calculations and error quantifications.

Still, we would like to keep some content in the main text, e.g., Section 3.2 and Fig. 3, as they visually show the modifications of X-STILT from STILT and may help readers easily understand the upwind surface influence onto a downwind atmospheric column.

***2.*** *Third, I am not quite satisfied with the transport error analysis. The problem is that the errors (mostly winds) for WRF and GDAS are not clearly defined, so it is hard to understand how good or bad the transport is and how the error can be related to signals (e.g., low winds to high signals or the impact of wrong wind directions – not presented clearly). The authors spend a lot of space to explain transport but it needs some improvement. Referring to the unpublished paper too much is not a good idea.*

We apologize for the lack of clarity in the transport error analysis. The definition of horizontal wind errors of meteorological fields is similar to that in Lin and Gerbig (2005) and was provided in **Appendix B**:

"In terms of the wind error component ($u_\varepsilon$) mentioned in Sect. 2.6, two sets of parameters are used to describe, 1) $\sigma_{uverr}$, the standard deviation of horizontal wind errors (RMSE) describing to what extent should we randomly perturb air parcels; and 2) horizontal and vertical length-scales and time-scales (Lx, Lz, and Lt) determining how wind errors are correlated and decayed in space and time. We calculated different sets of wind error statistics over 3 vertical bins, i.e., 0–3 km, 3–6 km and 6–10 km, for randomizing air parcels. To obtain $\sigma_{uverr}$, observed winds at mandatory levels (i.e., 925, 850, 700, 500, 400, 300 mb) from surrounding radiosonde sites (Fig. 4) are compared against WRF- or GDAS-interpolated winds. Then, we averaged wind errors at different mandatory levels over aforementioned three vertical bins. In addition, wind errors are considered to be spatiotemporally correlated. To determine error correlation scales, differences in the wind errors are calculated and wind errors at different radiosonde stations or different reported hours (00UTC or 12UTC) are paired up based on their separation length- or time-scales. An exponential variogram is then applied to estimate the horizontal, vertical and temporal correlation scales, which are the separation scales when errors become statistically uncorrelated."

The wind error and transport error statistics over the five overpasses we focused are now summarized in **Table 1**. Wind error statistics (RMSE for lower atmosphere, 0-3km) for several overpasses for Riyadh are labeled as numbers in **Fig. S1**. Additionally, we now add a new set of analysis and subsection about the vertical transport errors, via propagating typical PBL errors into the model, as part of our response to Reviewer #1. Please refer to **Sect. 2.6.2** for the changes.

We agree with the reviewer that a lot of numbers/statistics were listed for the transport error analysis without explicitly discussing the linkage from errors in wind speed to XCO$_2$ signals in Sect. 3.4 and 3.5 (in the previous paper version). Now, we have added a paragraph in **Sect. 3.5 (P16L22-35)** to discuss this linkage and removed some sentences in simply listing numbers/statistics.

Here are some other main points about the XCO$_2$ transport errors:
- No large systematic errors in u- and v- component wind is discovered over dozens of overpasses (Fig. S1).
- For each sounding, XCO$_2$ errors due to the horizontal transport error are calculated from the CO$_2$ variance differences between the standard trajectory and the perturbed trajectory, for each level. More details regarding the transport error quantification at each model level and for each sounding can be found in **Appendix B**.

- For each overpass, the latitude-integrated XCO$_2$ error due to horizontal transport is a mixture of several factors. Relevant text has been added to Sect. 3.5:
    "The integrated XCO$_2$ transport error per track reflects the aggregate effect of several factors which interact, given how we propagate wind errors into XCO$_2$ space (Sect. 2.6):
    1) The magnitude of the modeled urban XCO$_2$ enhancements. In general, air parcels that are very far away from potential upstream emitters may hardly "hit" the emission sources or gain their enhancements, even after the wind perturbation. If the estimated signal is large (e.g., 3.04 ppm-deg. on 20151216 in Table 1), its resultant integrated transport error can also be fairly large (1.83 ppm-deg. in Table 1).
    2) The RMSE of u- and v-component winds. In general, larger wind errors will lead to larger changes in model trajectories and larger possibilities for perturbed trajectories in intersecting an emission source.
    3) How air parcels interact with surface emissions, i.e., the geometry/angle between the model footprint (or the wind direction) and satellite swaths. Changes in this angle may fluctuate the width of enhanced latitudinal band along with the final integration latitudinal ranges (i.e., 1.10°–2.25°). If the back-trajectory or backward wind direction is more parallel to the OCO-2 swath (events on 20141227, 20151216 and 20160216 in Fig. S10), the integration range and error covariance among soundings are usually larger, which yields larger integrated XCO$_2$ errors (e.g., 1.22, 1.83, and 1.05 ppm-degree in Table 1). The averaged latitudinal range for integration is about 1.66° (~189 km) over 5 tracks."

**3.** *Last, I would like the authors to comment on the utility of OCO2 for urban studies based on this work, because there is some skepticism about OCO2's capability for estimating urban emissions with relatively small areas.*

We addressed this concern in the last paragraph of **Sect. 4.4**. We briefly mentioned the limits of using OCO-2 on urban studies, such as limited temporal coverage or limited screened observations. We expect more data and diurnal variations after the launch of OCO-3 and its orbit on the International Space Station.

"OCO-2 observations have been utilized in several recent studies along with this work with a particular look into relatively small areas, e.g., individual power plants (Nasser et al., 2017) and megacities (Ye et al., 2017). Even though the XCO$_2$ urban signal over Riyadh may be in general smaller than those over other large cities, both model and observation successfully detect the urban signal. Still, no summertime XCO$_2$ signal has been derived, due to the lack of screened observations (QF = 0) reported in OCO-2 Lite b7 file over most summertime tracks (black bars in Fig. S1). No diurnal variation, revisit time of 16 days and relatively narrow swath of OCO-2 may still pose challenges to urban emission estimates. We expect the inclusion of more column observations in stationary (target) modes, e.g., by scanning over megacities by OCO-3 (Eldering et al., 2016), which may offer more concrete spatial and diurnal variabilities that benefits urban flux inversions. Many nations are devoting considerable resources in launching carbon-observing satellites that can potentially be coordinated in a larger monitoring system (Tollefson, 2016). Given that X-STILT can potentially work with most satellites (given their sensor-specific vertical profiles), we expect enhanced capability in emission constraints of urban emissions by combining column measurements with X-STILT."

**Detailed Comments**

**4. P1, L19.** *Global assimilation data seems to be too coarse for the urban scale CO2 simulation. Why use GDAS?*

The reviewer raised five detailed comments (including *comment **4, 13, 14, 27** and **28***) related to the meteorological field we used. We address them altogether here.

Yes—GDAS is the primary choice in this study. STILT trajectories over all five overpasses are guided by meteorological fields from GDAS. Although the spatial resolution of GDAS (0.5 degree) is coarser than WRF customized in this study, GDAS is the main choice in this work due to the following considerations:

1)  The surrounding terrain around Riyadh is relatively flat. For other cities with complex terrain, we may have two options. If we still use global assimilation data, the model may likely "return" larger wind errors and resultant XCO$_2$ errors around the best estimates. Alternatively, we always have the option to use higher resolution meteorological fields, e.g., customized WRF or HRRR, to better resolve the subgrid scale dynamics and terrain flows with more accurate estimates in ground heights.

2)  The regional wind error statistics (compared against observed winds from radiosonde stations) of GDAS is similar to that of WRF for the few cases we examined (Table 1). The reviewer or readers may be concerned about the wind error quantification, as the number of observation sites around the city may not be that large (e.g., comment #27). However, we discussed the pros (i.e., less cloud and vegetation coverage) and cons (i.e., sparser wind observation network) of choosing such city like Riyadh in **Sect. 1 (P4L24-27)**:

    "Riyadh, with a population of over 6 million by 2014 (WUP 2014), is chosen as the city of interest because of its low cloud interference, limited vegetation coverage, and isolated location in a barren area, which leads to higher data recovery rates and facilitates the background determination. Saudi Arabia has the largest CO$_2$ emissions among Middle Eastern countries and ranks eighth globally in 2016 (Boden et al., 2017; BP, 2017; UNFCCC, 2017)."

    and in **Sect. 4.4 (P19L15-24):**

    "Admittedly, the transport error analysis and near-field correction may work the best with the assistance of denser meteorological observing networks to characterize the error structures of transport errors. Increasing the density of surface networks may modify the wind error statistics including the wind error variances and horizontal

correlated length-scale, and further impact the model transport uncertainties and inversed fluxes. Yet, this shortcoming is not inherent to X-STILT and applies to other means of quantifying the transport errors based on real data as well. The trade-off of choosing a city in the Middle East like Riyadh to minimize cloud and vegetation influences is the relatively sparse observations of surface meteorological network or aircraft. The most recent OCO-2 b8 Lite files include retrieved surface winds for each sounding. Unfortunately, most of those surface wind retrievals are not available over Riyadh, but the retrieved surface winds for other urban areas, if available, may be used for assimilation and assisting X-STILT error analysis."

3) Our ultimate goal is to look at emissions from a couple of cities over the Middle East or even around the world with the assistance of X-STILT in future studies. Customized WRF field for many more cities and overpasses can be relatively computational more expensive and require careful evaluations on their configurations over different regions.

4) Lastly, the scope of this manuscript is to present a modified atmospheric transport model framework with an application over a relatively "simple" city. Our intention is not about evaluating the differences between two meteorological fields and making conclusions about which one is better (comment #28). And the STILT model itself is not fixated to a particular choice of meteorological field.

**5. P1, L21.**
*"68% in posterior scaling factor" should be "68% in posterior signal" because here the bias in background is in the units of signal. Also, it is not clear what 68 % in posterior scaling factor means. Posterior uncertainty in 1-sigma? Or Does it mean the bias in background resulted in 68% higher or lower bias in the posterior scaling factor?*

Clarifications: Our intention is to reveal or highlight the impact of different background methods on the posterior scaling factor ($\hat{\lambda}$ in Table 1). The posterior scaling factors (for mean $XCO_2$ signal) using M2H- and M3- derived observed signals are ~1.78 and ~1.14, as shown in Table 1. We now reworded **P1L21-23** as:
> "In addition, a sizeable mean difference of -0.55 ppm in background derived from a previous study employing simple statistics (regional daily median) leads to a higher mean observed urban signal by ~39 % and a larger posterior scaling factor."

**6. P1, L22.**
*It seems to me that the authors are referring to signal calculation, and the impact of uncertainty and bias on the urban signals by "Based on these results". I wonder if the authors can add a couple sentences that are more significant than these. If I put it differently, are these results the most important results we take home from this study?*

Yes—the goal of this study is to provide a modified version of STILT for column measurements and associated error quantifications (with a case study over a city in the Middle East). We have changed 'Based on these results' to 'Based on our signal estimates and associated error impacts' on **P1L23**.

**7. P3, L31.**
*Please add references related to "minimal guidance". The authors can simply add few references on uncertainties associated with atmospheric column simulations.*

We have reworded the sentence. Although the error impact from receptor setups can be small, most studies simply depicted their model setups without further explaining why they chose those setups or the error impact (due to model configurations) on modeling $XCO_2$. No study examined this error impact on column simulations, to our best knowledge. Text has been changed to as:
> "Previous studies reported negligible to ~20 % of the modeled enhancements are reported as the error impact due to STILT particle number (released from a fixed level), depending on adopted particle numbers, examined species and their components/sources (Zhao et al., 2009; Gerbig et al., 2003; Mallia et al., 2015). When it comes to representing

an atmospheric column using particle ensembles, many studies depicted their setups for receptors/particles without further explaining why they chose those setups or the error impact (due to model configurations) on modeling $XCO_2$. Although this error impact may be small, we still perform a set of sensitivity tests to provide more guidance on placing column receptors."

**8. P3, L33 – 34:**
*The authors underestimate recent developments in inverse modeling. There are several atmospheric inverse studies that consider transport errors and use full error matrix (not just diagonal), in particular non-CO2 studies (e.g., regional methane studies). The references there are old and does not support the statement. The authors need to be specific. I may agree that there are not many studies to incorporate full error characterizations for column-observation inversion studies, but there are now many studies to consider errors more carefully. The authors should be careful in this statement and need corrections.*

We now add a few more references on recent urban $CO_2$ and regional methane inverse studies, e.g., Jeong et al., 2013, Lauvaux et al., 2016 and Zhao et al., 2009. And, we have changed some of our statements regarding the full error characterizations for column inversion studies. Text in **Sect. 1 (P3L35-P4L2)** has been reworded as:
> "Approaches to quantify errors in horizontal wind fields and vertical mixing have been proposed followed by comprehensive error characterizations on atmospheric simulations (Gerbig et al., 2008; Jeong et al., 2013; Lauvaux et al., 2016; Lin and Gerbig, 2005; Zhao et al., 2009). Recent efforts (e.g., Lauvaux and Davis, 2014; Ye et al., 2017) have been made to rigorously examine the column transport errors."

*Also, I am surprised that the authors use a very simple inversion – later in the section I find they are not well formulated but rudimentary – I don't see the benefit of including the inversion result in the study. Please note that there are many sophisticated inversion methods that are much more amenable to error characterizations – please do some literature review.*

We appreciate the criticism on the simple inversion from the reviewer. However, we note that conducting a comprehensive inversion or making conclusions about inversed urban emissions may be out of the scope of this study. This study focuses on the model descriptions for $XCO_2$ signal extraction and error quantifications (that helps provide insights into future comprehensive inverse studies), with a case study over Riyadh.

Two reasons for including a simple scaling factor inversion in discussion section:
1) to follow the scaling factor analysis and compare the transport error results in Ye et al. (2017), even though our methods and adopted atmospheric transport models can be different; and
2) to address the importance of background estimates and provide STILT-based error impacts (e.g., posterior covariances).

In addition, we agree with the reviewer that this is a simple inversion, probably because we treated the gridded upwind urban emissions as a whole (i.e., no adjustments for emissions for each gridcell) and integrated latitude-dependent $XCO_2$ enhancements. More sophisticated inversions on the spatially distributed emissions, given more sampled satellite overpasses or more sampled cities over the Middle East will be considered in future studies. However, we may justify that these simplifications are made for the consideration of reducing error impact, in particular from potential near-fields wind biases.

Relevant text in Sect. 4.2 has been modified and added:
> "Estimated background uncertainty is represented by the spatial variation and retrieval errors of background observations and may be reduced given large sampling size. To further demonstrate X-STILT's potential role in inverse modeling and the potential background "bias" via different background methods on inversed results, we conducted a simple scaling factor inversion (Rodgers, 2000), based on 5 pairs of model-data latitudinally-integrated urban signals. Even though our sampling may seem to be small and the gridded urban source emissions are treated as a whole (i.e., no adjustments for emissions for each gridcell), these integrated signals and errors are chosen to reduce the impact of

potential near-field wind bias on model evaluations. Also, we are partially limited by the overpasses over Riyadh (black bars in Fig. S1)."

**9. P4, L8-9:**
*I don't quite understand "Most of these studies aim at extracting relatively large CO2 changes at a fixed level within the PBL or due to large emissions such as of wildfire". Which studies are the authors referring to? The point is tower vs. column or large signal vs. small? Are the authors suggesting that the study site in this work has very little CO2 changes (exchanges?)? The study areas in this study are different from other urban areas in previous studies in terms of CO2 variations or signal-to-noise ratio? Also, related to this, why did the authors choose this study area instead of some US large cities?*

We regret the confusions caused by these lines. The main point of this paragraph is to point out some common ways to define background e.g., the trajectory endpoint method, as well as the limitations of those modeled background, especially when trying to estimate background from column observations. Text has been changed to as:
"The aforementioned studies (adopting the trajectory-endpoint method) aim at extracting relatively large $CO_2$ anomalies (e.g., at a fixed level within the PBL or due to large emissions such as of wildfire) out of the total measured $CO_2$."

No — we are not suggesting the study site in this work is special or the $CO_2$ changes for this site is low. We just wanted to bring up the difference in extracting urban enhancements from PBL-based or column observations, where the enhancements are relatively larger and smaller by nature. Because the relatively small column enhancement and SNR when extracting the column signal, even a small error in background as low as 1 ppm can be "harmful" for interpreting $XCO_2$ variation.

For the reason of choosing Riyadh rather than other large cities in US, we explained in **P4L23-27**: "its low cloud interference, limited vegetation coverage, and isolated location in a barren area, which leads to higher data recovery rates and facilitates the background determination." And, we will expand our study area to examine more cities around the world in future work.

**10. P4, L13:**
*It is not clear why the authors introduce a new background estimation method. I guess this has to do with column simulations, but please state the reason more clearly.*

Yes—we introduce a new method because of the relatively small SNR in extracting urban enhancements (of few ppm) out of total $XCO_2$ concentration as well as limitations of some other methods. We added a very brief limitation of trajectory-endpoint method for column simulation on **P4L10-12:**
"However, for studying $XCO_2$ that is less variable than near-surface $CO_2$ (Olsen and Randerson, 2004), potential errors in modeled concentration fields and atmospheric transport may pose more significant adverse impact on derived urban signals."

and now add the limitation of simple statistics on **P4L14-15**:
"These simple statistical methods often neglect the transport and may use the less accurate spatial region to select measurements for deriving background values."

We further discussed these background methods in **Sect. 2.3 and 3.3**.

**11. P4, L28:**
*Please define "prior profile" since many "priors" are used in this paper.*
"Prior profile" stands for the "a priori $CO_2$ profile" from OCO-2 Lite product. Text has been made on **P5L5**.

**12. P5, L4:**

*It seems "ratios of the pressure difference between adjacent model levels over that between adjacent retrieval levels" needs more clarification. Once PW is interpolated to model levels, then the pressure difference between model levels (as the scaling factor) should be enough? Please clarify.*

*PW* function is primarily estimated based on the air mass or pressure difference (*dp*) between two layers. Fig. 2b shows that OCO-2 levels have relatively **constant pressure difference** (*dp_oco2*) and *PW* values (in black dots, except for the very top and bottom level). On the contrary, model levels are finer (in orange circles in Fig. 2b) with two different vertical spacings in **altitudes** (100 m vs. 500 m) below and above 3 km. So, those scaling factors are to adjust the *PW* function according to difference in *dp_oco2* vs. *dp_stilt*.

For example, from 0 to 3 km, *dp_stilt* ranges from ~8-10 mb, while *dp_oco2* are mostly ~52 mb (red circles vs. black dots in Fig. 2b). Thus, we further scaled down the interpolated *PW* (red circles in Fig. 2b) by the ratio of *dp_stilt* over *dp_oco2* (e.g., sf = 10 mb/52 mb), because of less air between model levels than initial retrieval grids. Thus, final *PW* after scaling has value of ~0.01 (orange circles in Fig. 2b).

Comparing air below 3km, fewer model levels are placed from 3-6 km (~650 to 450 mb), which gives larger *dp_stilt*, larger *PW* scaling factors and resultant larger scaled *PW* of ~0.03 –0.04 (orange circles in Fig. 2b).

Since no model level placed above MAXAGL, *PW* stay the same as the initial OCO-2 *PW* values (blue circles in Fig. 2b). Finally, we made sure that the sum of vertical PW profile ends up being 1 for each sounding/receptor.

Relevant text has been clarified (in Sect. 2.1 on **P5L14-22**):

"Interpolations are further needed to resolve the mismatch between prescribed OCO-2 retrieval grids and model levels for the lower part of the troposphere. Our intention is to preserve the finer modeled $CO_2$ variations by performing interpolations of satellite profiles from retrieval grids to model levels. Vertical profiles of $AK_{norm}$, *PW* and $CO_{2,prior}$ are treated as continuous functions and interpolated linearly to model grids (red circles in Fig. 2). Note that the initial OCO-2 *PW* functions have steady value of ~0.052 (except for the very bottom and top levels; black dots in Fig. 2b), which results from constant pressure spacings (*dp_oco2*) between two adjacent OCO-2 levels. However, X-STILT levels are much denser with smaller pressure spacings (*dp_stilt*) or less airmass between their two adjacent levels. Therefore, the linearly interpolated *PW* (red circles in Fig. 2b) needs an additional scaling via a set of "scaling factors" representing the ratios of pressure spacings in STILT versus OCO-2 retrieval (*dp_stilt/dp_oco2*), to arrive at the correct *PW* for each finer model grid (orange circles in Fig. 2b)."

**13. P5, L12:**

*I wonder what "When WRF fields were available" means. WRF is not used for all days/hours? For the comment on the abstract, I added that GDAS alone is not sufficient for the urban scale. Also, more importantly, the authors must add the minimum description of the WRF model, e.g., vertical and horizonal resolutions unless stated somewhere later in the sections. It is not appropriate to toss everything to another unpublished reference.*

STILT trajectories over all five overpasses are guided by meteorological fields from GDAS. These customized WRF fields can be computational expensive and require careful evaluations on its configurations. Thus, model trajectories for the first two overpasses (i.e., 12/27/2014 and 12/29/2014) are driven by nested WRF and GDAS fields.

We have added a brief description on WRF configurations in **Sect. 2.1.1 (P5L35-P6L1)**: "Hourly WRF fields contain 51 vertical levels with boundary conditions from 6-hourly 0.5°×0.5° NCEP FNL (Final) Operational Global Analysis data (Ye et al., 2017) are customized and utilized for the first 2 of the total 5 overpasses over Riyadh."

**14. P5, L15:** *Is GDAS the primary choice?*

Yes. For the reason of choosing GDAS, please refer to our response to comment #4.

**15. P5, L19:**

*Remove "a certain height", but directly use an explicit one, e.g., "the maximum release height" - unnecessary vagueness. I see a few places in this paper that use such a vague expression.*

We have replaced 'a certain height' with 'the maximum release height' or 'MAXAGL'.

**16. P5, L20:**

*Please state what constitutes "different setups" so that the reader has a clear sense of the setups that might differ. As written, it is not clear.*

Different setups comprise of the maximum release level (MAXAGL), the vertical spacing of release levels (dh), and the particle number per level (dpar), which is now clarified in the main text (**P6L8-9**).

**17. P6, L26:** *Define "BP".*

BP is the acronym for the British Petroleum Company plc and BP Amoco plc (an oil and gas company). We add it in the main text.

**18. P7, L3:**

*Please say so, if 0.1 degree is the final resolution for signal calculation, which could be coarse for a urban region.*

Clarifications: We still kept 1km x 1km anthropogenic emissions from ODIAC and generated 1x1km footprints to calculate the $XCO_2$ signal. The 1km x 1km should be fine for getting $XCO_2$ signals from the urban. We further clarified this point on P9L20 ("To calculate modeled XCO2 enhancements, we used the latest (year 2017) version of …")

However, when it comes to emission uncertainty calculations, emissions from ODIAC are aggregated to 0.1° (due to mismatches in the horizontal resolutions of emission grids). Thus, another set of footprints with 0.1° x 0.1° spatial resolution is generated to convolve with the 0.1° x 0.1° spatial emission uncertainty, which propagates the errors in prior emissions to the $XCO_2$ space (to the 1st order).

**19. P7, L10:**

*Please comment on the 1-degree bio flux relative to the size of the study area and its potential impact (due to coarse resolution) on the inversion.*

We agree with the reviewer the 1° x 1° CarbonTracker can be comparable to the size of the urban domain and too coarse to resolve the subgrid scale heterogeneity in biospheric fluxes. However, the potential impact on inversion or the signal calculation is small due to following reasons. And, we did not modify the main text.

1) For the inversion and signal calculations, we actually used the overpass-specific background from M3, instead of the trajectory-endpoint based background that relies on CarbonTracker biospheric fluxes.

2) The biospheric influence has been included over the background latitude range and then get subtracted from the total observed $XCO_2$. M3 may work fine, unless large gradient of biospheric fluxes exists around the urban area (mentioned as a potential limitation of M3 in **Sect. 4.4, P19L1-2**):

"When examining summertime tracks or tracks over some other cities, potential local gradients in biospheric fluxes should be considered as those gradients can affect our overpass-specific background."

3) Lastly, the land around Riyadh is relatively barren with minimal biomass coverage. For studying other cities, we can use biospheric fluxes with finer spatial resolution generated from other inventories/models, e.g., MsTMIP.

*20. P8, L22:*

*Please add comments on the potential impact of transport over the city when using Method 3. I note that the authors discussed the potential transport error for Method 1 (i.e., endpoint method). Wind direction could be a serious problem for Method 3. Enough overpasses (both up- and down-wind) are available for Method 3.*

We agree with the reviewer that It is possible that higher $XCO_2$ values may be included in the background ranges, due to mismatches between modeled and observed plumes. We have added a paragraph in **Sect. 2.3.3 (P9L17-28)** to discuss this impact on background value (also pasted here):

"In addition to random errors (that are resolved by the inclusion of the aforementioned wind error component and broadening of the city plume), potential large bias in near-field wind direction may lead to mismatch in modeled and observed background regions and may bring relatively higher $XCO_2$ values into background $XCO_2$. However, we do not explicitly account for the potential near-field wind bias's impact on forward-trajectories defined urban plume with following considerations. Firstly, we attempted to propagate a near-field wind bias into the modeled plume by rotating forward trajectories, whereas the robustness of this near-field bias can be affected by the very few wind measurements near Riyadh (further explained in Sect. 2.6.1). Secondly, the background latitude range defined by M3 with the broadening effect (blue lines in Fig. 5b) in general matches well with that observed from OCO-2 for most overpasses, which implies that the overall wind bias around our study site is not significant. Lastly, even if potential wind bias may result in less accurate background range and bring elevated $XCO_2$ into the background, the background uncertainty implicitly contains information about the spatial variation in background measurements (green ribbon in Fig. 5b). In addition, the M3-derived background is the mean value of mostly hundreds of background observations (numbers in Fig. 6e), which may not be greatly affected by a few potential urban-enhanced measurements."

*21. P9, L20:*

*I agree with the authors that STILT configurations can affect the results. But I don't understand the use of bootstrapping here. The original sample here is from the 401 levels (too many in my opinion). However, what we are interested in is the results from different set-ups, e.g., 20, 40, levels, which can be different from the original samples of the 401 levels. In practice, 401 levels are unrealistic, e.g., for annual analysis.*

Clarifications: Yes— what we are interested in is the results from different setups, e.g., whether 20 or 40 levels can be enough. The number of levels (nlevel) is further decomposed into the vertical spacing dh and the MAXAGL. By increasing dh from 50 m to 100 m, the number of levels reduces by half with fixed MAXAGL of 6 km.

Note that the original sample is release from 0 to 10 km with a spacing of 25 m (n = 1, 2, 3, …, 401). For example, if we wanted to test the one case with dh = 50m, we randomly resampled trajectories released from every other level from 0 to 6 km (n = 1, 3, 5, …, 241) for 100 times. In other words, we got 100 sets of resampled trajectories with the same combination (MAXAGL = 6km, dpar = 100 and dh = 50m). From those 100 new sets of trajectories and resultant 100 $XCO_2$ enhancements, we calculated mean and SD of those enhancements. SD is used to reveal the random uncertainty (error bar in Fig. 5c), while the mean for one case is compared with other means, to reveal any systematic bias (e.g., the decreasing trend of red dots in Fig. 5c). Therefore, we actually do not care about difference between the resampled trajectories against the original sample.

*22. P11, L15:*

*It is surprising that MAXAGL < 2.5 km did not fully capture CO2 enhancements. I would expect that there is not much surface influence above 2 km. Is it because the study region is associated with really high PBLH? As the authors stated in L30-32, the lower portion of the column should matter most. Then why would MAXAGL of ~2.5 km not capture the full enhancement of CO2? Please add sentences that dis- cuss the reason for this. Actually, looking at Figure 8(a), I realize that there are only two cases below 2.5 km. So, 2.5 km itself looks fine. My guess is that even 2 km should be fine. I think the authors give the reader somewhat wrong information here, considering the fact that using a higher altitude for MAXAGL increases the computational cost significantly. My understanding from this is: 1) use 100 − 200 m vertical resolutions be- tween 0 − 2 km and 2) above 2 km,*

*use 500 m. If the authors can show even MAXAGL of 2 km is comparable to 2.5 or 3 km, this will reduce the computational cost significantly. I don't understand why the authors use 100 m for up to 3 km given the result shown in Figure 8(a), which in my opinion is too much without good reasoning. I think that some other studies will easily show denser vertical resolutions between 0 – 2 km is good enough.*

We thank the reviewer in pointing out this detail and now add the one case with MAXAGL of 2 km in Sect. 3.1 and Fig. 6a. The one simulation using MAXAGL of 2km looks much better than the one with 1.5km MAXAGL, but can be slightly lower than simulation using even larger MAXAGL. We have further run another set of uneven vertical spacing test to see whether a cutoff level of 2 km is enough.

Results are added and explained in **Sect. 3.1 (P13L3-10**, also pasted as below):
"We further performed two cases with uneven vertical spacing below and above a "cutoff level". Both tested three different lower spacings (of 50, 100 or 150 m) with a fixed upper spacing of 500 m. Two cases differ only in their cutoff levels (2 or 3 km). The comparison of the uneven *dh* against the constant *dh* experiment shows that their results in $XCO_2$ enhancements are fairly similar, suggesting that the lower spacing below the cutoff level matters mostly to model results, because most anthropogenic $XCO_2$ enhancements are confined within the PBL. Also, results for uneven *dh* case with the cutoff level of 3 km (blue triangles in Fig. 6c) are more closed to the "truth" implied by the constant *dh* case (red dots in Fig. 6c). To be safe, column receptors are placed from 0–3 km with a spacing of 100 m and from 3–6 km with a spacing of 500 m."

Yes—the PBLHs or mixing height are generally high over the upwind region of our city. Information about model-interpolated mixing depths can be found in Appendix D.

**23. P11, L34:**
*Please clarify what the fractional uncertainty means here. How did the total particle number become >12500 with 100 particle every 100 m within 3 km?*

Clarifications: The fractional uncertainty is calculated as the ratio of random uncertainty (in ppm, error bar in Fig. 6a-c) over the averaged simulated enhancement (in ppm, red dots in Fig. 6a-c) of results by resampling trajectories for 100 times. We have now clarified the fractional error in **Sect. 2.5 (P10L31-34)** as well:
"100 urban enhancements are calculated from 100 new sets of trajectories for each test. Basic statistics—i.e., mean values and standard deviations (or fractional uncertainty, i.e., SD/mean) among these 100 enhancements—are used to infer systematic and random uncertainties in each test, respectively (with results showed in Sect. 3.1)."

For testing the sensitivity of $XCO_2$ due to changes in one receptor parameter, the other two parameters are fixed. Specifically, dpar = 100 and dh = 100m are used for testing different MAXAGLs from 1 to 10 km (Fig. 6a);
dh = 100m and MAXAGL = 6km are used for testing different dpar (Fig. 6b);
dpar = 100 and MAXAGL = 6km are used for testing different dh (Fig. 6c).
Thus, the one simulation (dpar = 100, dh = 50 m, MAXAGL = 6 km) has 12,000 particles.

We now clarify the use of fixed MAXAGL of 6km for dpar test in **Sect. 3.1 (P12L37)** and in Fig. 6a-c:
"In addition, we conducted two experiments using constant and uneven vertical spacings with the fixed MAXAGL of 6 km and *dpar* of 100."

**24. P12, L32:** *"incorporates both" to "both incorporates"*
Text changed.

***25. P12, L37:***

*I wonder what "we added a wind error component to broaden the urban plume (Sect. 2.4.3 and Sect. 2.6) that helps reduce the inclusion of enhanced values in the background region" means. I can understand this could help reduce strong local sources under the assumption that broadening plumes with additional errors reproduces the reality more accurately. But broadened background does not necessarily solve the bias in the wind direction that is directly related to the enhancement in the background region.*

We agree with the reviewer and are aware of the impact of this wind error component onto our defined background values. Please refer to our response for comment #20.

***26. P13, L3-6:***

*How did the authors judge which one is the more accurate background that is assumed to be close to the (unknown) true background? The impact of the background bias (0.56 ppm here) on the emission estimation depends on the magnitude of the observation; it can have only a small impact when the local observations are large.*

We agree with the reviewer that it can be different to judge which method is the "truth", since the background value is an unknown and our examined sample size could be small.

However, we would still argue that M2H may not be suitable for local/urban studies, like this study. We have now added two paragraphs to try to discussion the limitations and advantages of M2H and M3 in **Sect. 3.3 (P13L37 – P14L18**, pasted as below):

"We now focus on the comparison between M3 and M2H with objectively analyzing their advantages and limitations. On average, M2H derived background is lower than our localized "overpass-specific" background by 0.55 ppm (Fig. 6e), which can primarily be attributed to different defined background regions. M3 defined the background region from the same track as the one over Riyadh, which guarantees that the background air contains variations due to long-term atmospheric transport, natural sources/sinks and $FFCO_2$ emissions except for local emissions (e.g., from Riyadh). Whereas the enhanced air contains the enhancements due to local emissions on top of all the information included in the background air. Therefore, the subtraction between M3-defined background and enhanced air correctly represent the $XCO_2$ portion enhanced by the local emissions. On the contrary, M2H use a fairly broad background region (0° N–60° N, 15° W–60° E in Fig. S4) to estimate gridded anomalies over all places in Europe, Middle East and North Africa. Although may yield more data, this broad spatial region may misrepresent the correct upwind region, because the wind regime can be quite different among different overpass dates or seasons.

We admit M3-defined background range and background value can be affect by potential large wind bias over cities other than Riyadh. However, the impact on background may be small and is implicitly considered in the background uncertainty (previously discussed in the last paragraph in Sect. 2.3.3). As for M2H, all regional OCO-2 measurements are lumped into its background calculation. For example, some measurements on the east-most overpass in Fig. S4 are affected by Riyadh's emissions, whereas atmospheric columns at soundings along the west two overpasses in Fig. S4 may not necessarily be the background air that eventually arrives at region around Riyadh. Thus, the regional median of $XCO_2$ may not physically indicate the accurate background that is supposed to isolate local-scale fluxes. Therefore, our localized overpass-specific background is designed and more suitable for extracting local-scale $XCO_2$ anomalies. Given relatively small urban enhancements around our study site, this 0.55 ppm difference may lead to large differences in estimated observed urban signals and emission evaluations (Sect. 4.2)."

***27. P13, L34:***

*It depends on which wind observations are used. The number of sites for wind obs. in this study is too small to make a statement as shown here.*

We have changed the statement on **P14L34-36** to "Based on available radiosonde sites over the Middle East with relatively flat terrain (white crosses in Fig, 4)".

We are aware of the limitation of sparse wind observation network for this city and discussed the trade-off of choosing this city in **Sect. 4.4**:

> "Yet, this shortcoming is not inherent to X-STILT and applies to other means of quantifying the transport errors based on real data as well. The trade-off of choosing a city in the Middle East like Riyadh to minimize cloud and vegetation influences is the relatively sparse observations of surface meteorological network or aircraft."

Also, please refer to the relevant response to comment *#4*.

**28. P13, Section 3.4.1**

*Comparisons against OCO-2 XCO2 at selected soundings: What is the small conclusion here? After all the analysis, the authors state "we suspect that mismatch in the model-data enhancement widths is primarily due to errors in wind speeds". I expected that the authors state, e.g., "model X is better or worse than model Y in terms of wind' simulations compared to observations, and we also see better or worse in model X or Y for 'signal' comparison between model simulations and observations". Any advantage of WRF due to higher resolutions?*

> As stated in Sect. 2.1.1 (P6L2-3):
>
> > "We note that the primary focus is to assess the resulting errors given the choice of a particular wind field (i.e., GDAS 0.5°), rather than to carry out analyses of differences between WRF and GDAS."

Even though the shape of resultant $XCO_2$ contribution maps appear to be different between two models (e.g., Fig. 7b and 7f), the two latitude-integrated $XCO_2$ contributions (**Fig. 7d and 7h**) appear to be quite identical. The GDAS and WRF regional wind RMSEs are also listed in **Table 1**.

**29. P17, L30:**

*How large was the random error (S_lambda) relative to the background-subtracted enhancements? The 5 x 5 error matrix (if this is the model-data mismatch error covariance, i.e., the irreducible error component in the linear model) suggests that only 5 obs were used? If it is true, that seems to be too small, even for a simple linear regression. The scaling factor suggests the prior emissions are consistent with the observation. Is this the conclusion and what the authors expect from the comparison between modeled XCO2 and obs? The description for this simple inversion doesn't sound good at all.*

The random error (square root of the observational error variance; in ppm) are about 63 % to 85 % of background-subtracted enhancements for different overpasses for Riyadh. These random error per overpass are assumed to be independent (due to mostly long separation time) and reduced when aggregating over 5 overpasses. In this revised version, we further added 1) retrieval errors in the background error and 2) error in vertical mixing in the X-STILT transport error (based on a comment from reviewer 1). Thus, the random error is slightly higher than that previously reported.

Yes—we only use 5 pairs of latitude integrated observed and modeled $XCO_2$ signals. And various errors at each sounding have been properly aggregated to the overpass level to reduce impact from wind bias. We carefully examined every possible overpass based on number of soundings and screened soundings, wind errors and distance to the city (Fig. S1). Then, those overpasses are under manual check to see whether there's promising enhancements. Although we may be limited by our stringent criteria, we are inclined not to perform simulations or model evaluations over some other tracks with insufficient soundings.

We appreciate the criticism and agree with the reviewer that this is a simple inversion and have now commented the limitation of this simple inversion on its lack of consideration of the spatiotemporal structures in **Sect. 4.2**. We will perform more sophisticated column inversion urban studies and analysis on urban emissions in future studies.

The posterior scaling factor is about 1.14 given observed signals using M3 background, which does not suggest the prior emissions are consistent with the observations. No -- our intention is not to make conclusions about the urban emissions for Riyadh. We will perform more comprehensive inverse analysis over more cities in future studies. For reasons of conducting this simple inversion, please refer to our response for comment #8.

**30. P18, L4-6:**
*I wonder if the background estimation for column CO2 from OCO2 can be improved. Somewhat disappointing. I hope to see some discussions (a few sentences) on the utility of OCO2 for urban studies including the retrieval error (this urban region has relatively low enhancements, difficult for OCO2 to tell something), not only for this study area, but for future other regions, more generally.*

We appreciate the reviewer for this constructive comment on background estimates. We have now updated the background uncertainty by including the retrieval errors of observations over the background latitude range.

The reviewer is making a good point, and it will be great that we examined the background estimation over other urban regions. However, this may be beyond the scope of this model description paper (with application applied to a city). We will examine more urban regions given background from OCO-2 in future studies.

**31. Figure 7.**
*The trajectories seem to be stratified, with each streak (looks like thick streak) somewhat disconnected from each other, which looks strange. Any explanation? Is it because of different levels?*

Yes -- Figure 7 (now Fig S4) contains all air parcels released from different vertical levels. Air parcels at higher levels are driven by higher wind speed and different wind directions aloft than winds within the PBL. Those air parcels released from levels within the PBL are more concentrated near the receptor while parcels released from higher levels are displayed more to the west.

**32. Figure 8-e**: Please use the same labels for the legend, e.g., M3.
Have changed the label in panel e.

---

## Author Comment (AC3) · 22 Sep 2018

Thank you for the recommendation. I have created a release within the Github repository to mark the code version relevant to the manuscript and revised the code availability section with these updates. This is available at the following URL and Zenodo-registered DOI:

https://github.com/wde0924/X-STILT/releases/tag/v1.2

http://doi.org/10.5281/zenodo.1432528

---

## Author Response (AR2)

**Response to Topical Editor**

We thank the reviewers and topical editor for reviewing the revised manuscript. Even though no edits are required towards the final version of the manuscript based on editor's decision, we still made minor changes throughout the text with tracked changes in red. Few grammar mistakes have been corrected and few sentences have been reworded. We note that no change to the scientific content of this final version from the last revised manuscript has been made.

[revised manuscript text omitted]

**Figure 9.** Correlation between observed and simulated anthropogenic XCO$_2$ signals for 5 overpasses. Colors differentiate different satellite overpass dates. Model-data comparisons using GDAS-derived XCO$_2$ signals and observed signals based on different background methods. Error bars along x-axis and y-axis represent the overall observed uncertainty (represented as 1-$\sigma$, including XCO$_2$ spatial variability, background uncertainty and retrieval errors) associated with observed signals and the overall modeled uncertainty ($\sigma$, including emission uncertainty and transport uncertainty) around modeled signals. Dashed line represents the 1:1 line. Monte Carlo experiments are performed to fit linear regression lines based on sampled model-data signals and associated errors. Regression lines with positive slopes are shaded in light grey. Median values of slopes and y-intercepts from those multiple regression lines (with positive slopes) are used to draw a linear regression (black solid line).